# In vivo cell biological screening identifies an endocytic capture mechanism for T-tubule formation

Thomas E. Hall [1✉], Nick Martel[1], Nicholas Ariotti[1,2], Zherui Xiong [1], Harriet P. Lo[1], Charles Ferguson[1], James Rae[1], Ye-Wheen Lim[1] & Robert G. Parton [1,3✉]

The skeletal muscle T-tubule is a specialized membrane domain essential for coordinated muscle contraction. However, in the absence of genetically tractable systems the mechanisms involved in T-tubule formation are unknown. Here, we use the optically transparent and genetically tractable zebrafish system to probe T-tubule development in vivo. By combining live imaging of transgenic markers with three-dimensional electron microscopy, we derive a four-dimensional quantitative model for T-tubule formation. To elucidate the mechanisms involved in T-tubule formation in vivo, we develop a quantitative screen for proteins that associate with and modulate early T-tubule formation, including an overexpression screen of the entire zebrafish Rab protein family. We propose an endocytic capture model involving firstly, formation of dynamic endocytic tubules at transient nucleation sites on the sarcolemma, secondly, stabilization by myofibrils/sarcoplasmic reticulum and finally, delivery of membrane from the recycling endosome and Golgi complex.

[1] Institute for Molecular Bioscience, University of Queensland, Brisbane, QLD 4072, Australia. [2] Electron Microscope Unit, Mark Wainwright Analytical Centre, The University of New South Wales, Kensington, Australia. [3] Centre for Microscopy and Microanalysis, University of Queensland, Brisbane, QLD 4072, Australia. ✉email: thomas.hall@imb.uq.edu.au; r.parton@imb.uq.edu.au

The transverse tubule (T-tubule) system of skeletal muscle is an extensive membrane domain that allows action potentials from innervating neurons to reach the depths of muscle fibres thereby triggering calcium release in a synchronised fashion. The T-tubule system is a vital component of the skeletal muscle, frequently disrupted in muscle disease. This is exemplified by aberrant T-tubules and mutations in key T-tubule components in many human myopathies[1–6]. An understanding of the biogenesis of this enigmatic organelle is necessary for further insight into these diseases; however, progress has been compromised by the absence of adequate experimental models.

The T-tubule system develops within a defined, but extended period, in mammalian systems. In the mouse, the T-tubules are first observed as longitudinal elements at E16[7] which gradually become organised into transverse elements from around birth to post-natal day 10[8]. The mature T-tubule system is an elaborate system of surface-connected tubules permeating the entirety of the muscle fibre, and occupying a volume greater than that of the sarcolemma from which is it is thought to be derived[9]. The mature T-tubule system has a protein and lipid composition which is distinct from the sarcolemma, and an extremely precise organisation and morphology which includes linear elements and triad junctions at specific points, making contact with the sarcoplasmic reticulum. No cell culture systems faithfully reproduce the development of the highly organised T-tubule system characteristic of vertebrate muscle, and a tractable vertebrate animal model has been lacking until now.

Here, we establish an in vivo model to follow formation of T-tubules in real time. We combine live confocal imaging with state-of-the-art electron microscopic approaches, genome editing and quantitative overexpression screening to define the membrane pathways and molecular processes orchestrating T-tubule development. We derive a model for T-tubule formation involving the co-option of existing endocytic pathways and spatially restricted tubule capture to generate the organised T-tubule system.

## Results

**In vivo visualisation of T-tubule development.** In order to provide robust visualisation of the developing T-tubule system, we first tested several transgenic markers and vital dyes for co-labelling with the T-tubule specific protein Bin1[10–12] (Fig. 1a–e, Supplementary Fig. 1a, b). We found that fluorescent proteins tagged with the CaaX domains from either H-ras or K-ras were robustly localised to the T-tubules, as well as to the sarcolemma at 48 hpf (Fig. 1a, b). For this reason we obtained an existing transgenic line which strongly and robustly expresses GFP-CaaX in all cells from a ubiquitous promoter to use in our analyses[13]. This general membrane marker allowed us to visualise development of the T-tubule membrane system at an early stage without bias caused by focussing on specific proteins. To investigate the development of the T-system during the transition from nascent myotube to muscle fibre, we imaged transgenic embryos from immediately post-fusion of the fast muscle (16 hpf), until 48 hpf when the T-tubules were observed to penetrate the cell all the way from the sarcolemma to the midline of the fibre (Fig. 1f, Supplementary Movies 1 and 2). Direct measurements from still images showed that the T-tubules adopted a regular spacing at 1.9 µm intervals.

**A quantitative model of T-tubule development.** In order to understand the development of the T-system in a quantitative manner, and to probe more deeply our observations from light microscopy, we turned to electron microscopy. We used a potassium ferricyanide stain on thin transmission electron

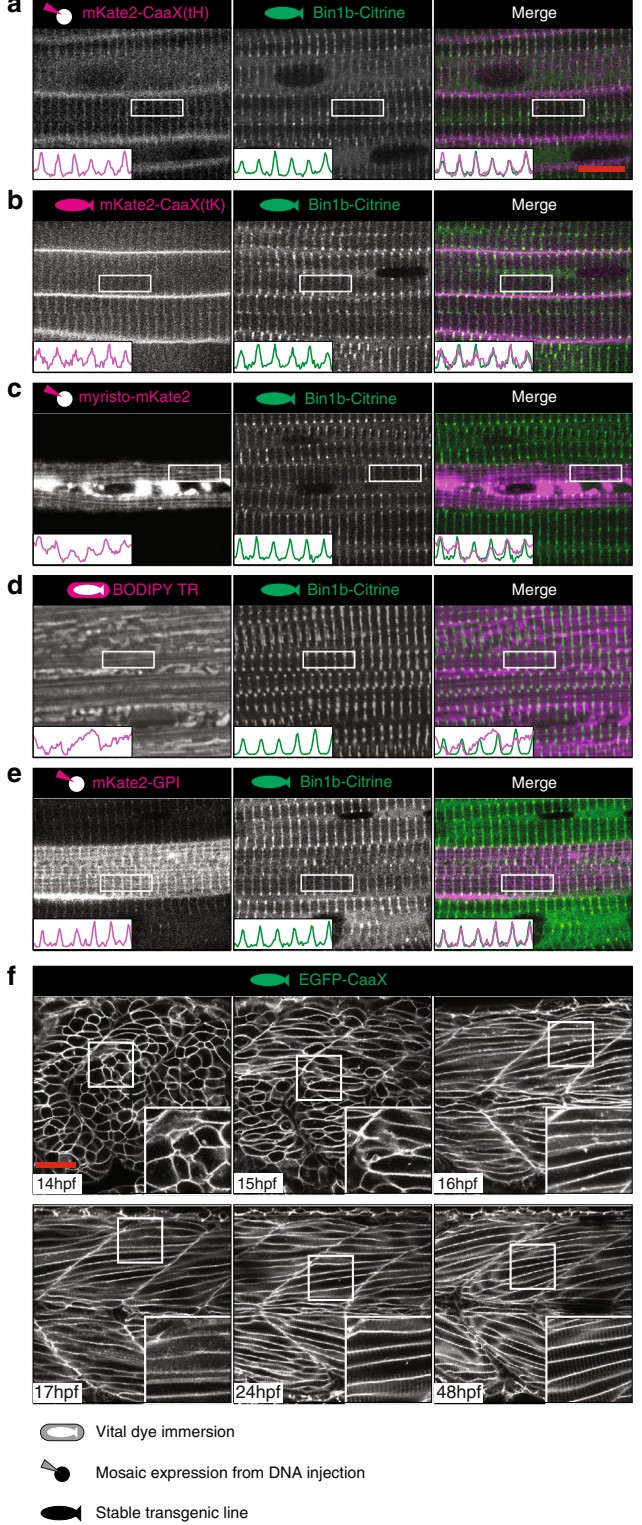

**Fig. 1 An in vivo system for T-tubule development. a–e** The CaaX domain from Hras marks both sarcolemma and T-tubule. Markers were screened at 48 hpf for T-tubule specificity on a stable transgenic background expressing Bin1b-Citrine from the endogenous locus. **a** CaaX domain from Hras (tH). **b** CaaX domain from Kras (tK). **c** Myristoylation domain from ARF5. **d** BODIPY-TR vital dye. **e** GPI anchor from zebrafish folate receptor gamma. **f** Time course of tubule formation using stable EGFP-CaaX background. All images are representative of 12 individual cells within different individual animals. See also Supplementary Movies 1 and 2. Scale bars; (**a–e**) 10 µm, (**f**) 50 µm.

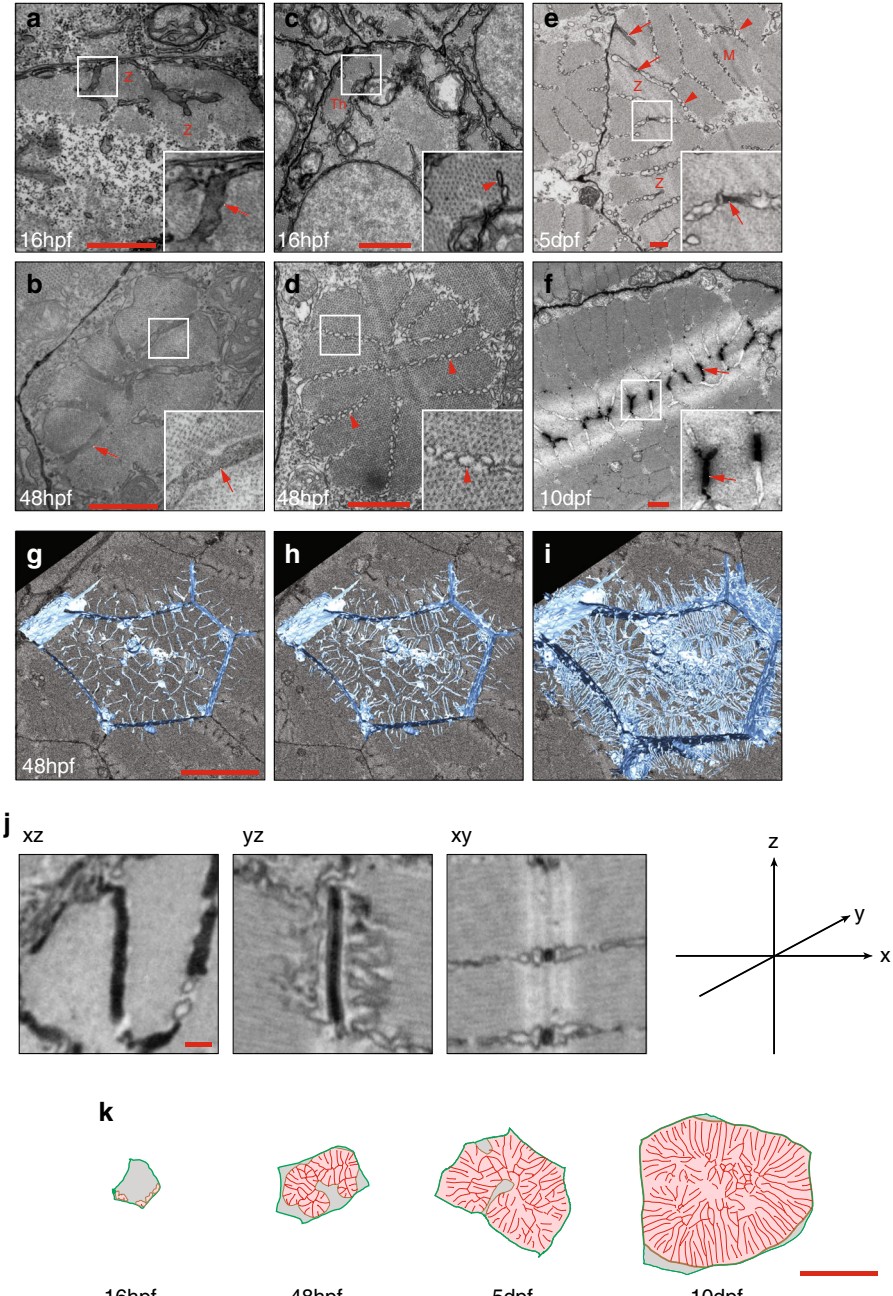

**Fig. 2 Ultrastructure of T-tubule development. a–f** Transverse orientated transmission electron microscope (TEM) sections showing early T-tubules and sarcoplasmic reticulum. **a, b** Transverse sections through the Z line (Z) at 16 and 48 hpf. Electron dense, ferricyanide stained tubules are denoted by arrows. **c, d** Equivalent transverse sections taken towards the centre of the sarcomere, some distance from the Z-line. Arrowheads denote sarcoplasmic reticulum, which has an unstained lumen and a beaded appearance. **e, f** Oblique sections at 5 and 10 dpf show both ferricyanide stained, electron dense tubules (arrows) and sarcoplasmic reticulum (arrowheads). M, M-line, Z, Z-line. See also Supplementary Movie 3 (**g–i**) TEM images were derived from two individual animals per time point and specific observations described were noted in >12 individual cells. Serial blockface electron microscopy and 3D reconstruction shows association of T-tubules with myofibril furrows or splits at level of the Z-line (see also Supplementary Movie 3). Images are derived from two individual animals and images shown are representative of both (**j**) Single planes from a focussed ion beam volume enables precise measurements of tubule morphometrics (quantitative data are shown in Fig. 3g). This analysis is derived from one individual. **k** Tracing of TEM images illustrates the architecture of myofibril furrows and associated tubules. Red, tubules/furrows; pink, myofibrils; green, sarcolemma. Full masks are shown in Fig. 3a–c. These analyses are derived from one individual per time point. Scale bars; (**a–f**) 1 μm, (**g, h**) 10 μm, (**j**) 200 nm, (**k**) 10 μm.

microscopy (TEM) sections to label the T-tubules[8] and distinguish them from sarcoplasmic reticulum (SR). Sections taken at 16 hpf showed electron dense, transverse oriented tubular membranes associated with the forming Z-lines (arrow Fig. 2a). By contrast, transverse sections taken towards the centre of the sarcomere some distance away from the Z-lines showed

association of longitudinal sarcoplasmic reticulum (identifiable by its beaded appearance and lower electron density) with the early myofibrillar material (arrowhead, Fig. 2c). By 48 hpf, the myofibrillar bundles, analogous to previously described nascent myofibrils[14] developed longitudinal furrows as they increased in diameter consistent with the Goldspink model of myofibrillar

splitting during growth[15] (Fig. 2b, d–f). Thin sections and serial blockface scanning electron microscopy with 3D reconstruction showed that these furrows contained SR and, at the level of the Z-line, T-tubules (Fig. 2b, d–j, Supplementary Movie 3). Tracing of thin sections taken between 16 hpf and 10 dpf showed that furrows develop in a spatially predictable manner, such that each contractile filament maintains close proximity to a furrow, and consequently a T-tubule (Fig. 2k). We reasoned that these reiterated spatial relationships would be amenable to mathematical modelling.

To produce a quantitative mathematical model of membrane sequestration into the forming tubules, we required more accurate empirical data from larger numbers of developing muscle fibres than is possible using serial blockface methods. Instead, we assembled large mosaics of tiled TEM micrographs covering entire transverse sections through the trunk musculature at 48 hpf, 5 dpf and 10 dpf. Using custom written image analysis macros we were able to produce masks defining muscle fibres, myofibrils and furrows within a half-section through an entire animal (Fig. 3a–c), and extract morphometric data (Fig. 3d, e, Supplementary Fig. 3a–c).

Using the functions derived from these empirical measurements, and from measurements of somite width (Fig. 3f; a proxy for fibre length) and T-tubule diameter (Fig. 3g), we derived a single equation to predict the total surface area of T-tubule membrane within muscle fibres of specified length and diameter (Supplementary Fig. 3d–g, full details are given in the Methods). With this model we calculated that the T-tubule system (only identifiable after 16 hpf), has developed into a complex organelle with a surface area comprising 35% of the total surface area of the cell (2579 : 4817 $\mu m^2$; T-system : sarcolemma) by 48 hpf (total surface area = T-system surface area + sarcolemma). By 5 dpf the relative surface area of the T-tubule domain gradually increases to 40% (4737 : 7092 $\mu m^2$) and after 10 dpf will overtake the surface area of the sarcolemma (13695 : 14456 $\mu m^2$; 49%). For the largest fibres in adult fish, we applied our model to fibres from a 12-month-old male from the zebrafish atlas (http://bio-atlas.psu.edu/zf/view.php?s=220&atlas=18; http://bio-atlas.psu.edu/zf/view.php?s=275&atlas=17), where the largest fibres are approximately 580 $\mu m$ in length and have diameters of up to 40 $\mu m$. In these fibres, our model predicts that the T-tubule system has approximately 1.5 times of the total surface area of the cell (Fig. 3h, i). The increase in surface area of the two membrane systems during the two early time periods correspond to an average of ~1.03 $\mu m^2\, min^{-1}$ of new membrane addition for the entire 48 h–5 dpf period and ~2.27 $\mu m^2\, min^{-1}$ for the 5 dpf to 10 dpf period. These results emphasise the magnitude of new membrane synthesis and, necessarily, flow through and delivery from the Golgi complex to allow this membrane expansion. Previous studies have suggested a maximal rate of approximately 2.7 $\mu m^2$ of membrane $min^{-1}$ for passage through the Golgi complex in rapidly dividing transformed cell lines[16] which would be adequate for the generation of the T-tubule system and for the expansion of the sarcolemma based on our estimates. However, additional membrane synthesis would obviously be required for the generation and expansion of the other crucial membrane systems during this period such as the SR. This implies that specific cellular machinery must be upregulated at the onset of T-tubule formation, including components required for membrane synthesis, for sorting, and for targeted delivery to specific domains including the T-tubule system during this growth period.

**Stability is dependent upon sarcomere integrity**. The strict association of the T-tubules with the sarcomeric Z-lines led us to question whether the contractile apparatus was involved in the scaffolding of T-tubules during development. To examine further the relationship of sarcomerogenesis to T-tubule development, we made a transgenic line expressing a red fluorescent protein fused to the lifeact tag[17], which labels F-actin, including sarcomeric actin. Live imaging of fish carrying both red Lifeact and green CaaX transgenes revealed transverse tubules invariably extended inwards towards the midline of the cell while maintaining a perpendicular association with the developing actin-containing myofibrils (Fig. 4a, b, Supplementary Movie 4), in agreement with our EM data. Examination of the sarcoplasmic reticulum, marked by another stable transgenic line expressing a red fluorescent protein tagged with an N-terminal calreticulin signal sequence and a C-terminal (KDEL) ER retention sequence, showed that the SR also adopts a regular, Z-line associated arrangement upon sarcomerogenesis (Fig. 4c, d). Titin cap (Supplementary Fig. 4a, b), alpha-actinin (c–d) and junctophilin 1a (e–f) were also spatially associated with T-tubule development, whereas SERCA1, a marker of the non-junctional SR was specifically excluded from the CaaX positive T-domain (g–h). Finally, in order to test directly whether tubule formation is directly dependent upon the sarcomere, we employed a loss of function approach using CRISPR-Cas9[18]. We generated embryos possessing null alleles of slow myosin heavy chain (the only myosin expressed in zebrafish slow muscle at early stages[19]). All slow muscle cells in these embryos showed a reduction in the distance between T-tubules, reflecting a concertina effect on the sarcomeres, where the Z-lines became closer together (Fig. 4e, g). Next, we generated a null allele of titin, the molecular scaffold which links myosin filaments to the Z-line[20]. In these mutants, the T-tubules completely lost their transverse structure and instead became a disorganised meshwork (Fig. 4f). TEM images showed unstructured myosin and actin filaments and narrow unstructured electron dense tubules (Fig. 4h–j). We noted however, the stochastic coalescence of junctional SR and electron dense tubule (arrows, Fig. 4i, j). Quantitation of the minimum diameters of SR-associated vs non-SR associated T-tubule (oblique cut tubes) in titin mutants showed a 90% increase in average diameter (43.5 vs 22.8 $\mu m$) in tubules associated with SR, suggesting that tubule morphology is influenced by interactions with SR or thin filament components (Fig. 4k).

**T-tubules are stabilised by endocytic capture**. The tight association of the forming T-tubules with the Z-line and junctional SR led us to investigate further the first interactions of the tubules with the forming contractile apparatus. Using live imaging, we observed at the earliest stages, both tubular and vesicular CaaX positive structures beneath the plasma membrane (Fig. 5a–c). These tubular structures appeared to then become stabilised into the familiar transverse-orientated tubules regularly spaced at 1.9 $\mu m$ intervals. We next asked whether, during formation, the T-tubules are in fact surface-connected, or whether the nascent tubule begins as an internal compartment which opens to the extracellular milieu later in development. To test this, we performed intramuscular injection of Alexa-647 labelled UTP, which has been used as a low molecular weight fluorescent tracer that is unable to cross an intact plasma membrane[21]. Injection of the tracer into 16, 24 or 48 hpf GFP-CaaX zebrafish resulted in immediate infiltration into the tubules, confirming surface connectivity (Fig. 5d–f). Similarly, uptake of the fluid-phase marker Alexa-647-dextran (10,000 MW), also resulted in immediate infiltration into the tubules, whereas uptake into putative endosomes only occurred after several minutes (Fig. 5g, Supplementary Movie 5). Serial blockface scanning electron microscopy and 3D reconstruction of 48 hpf fibres at high resolution showed

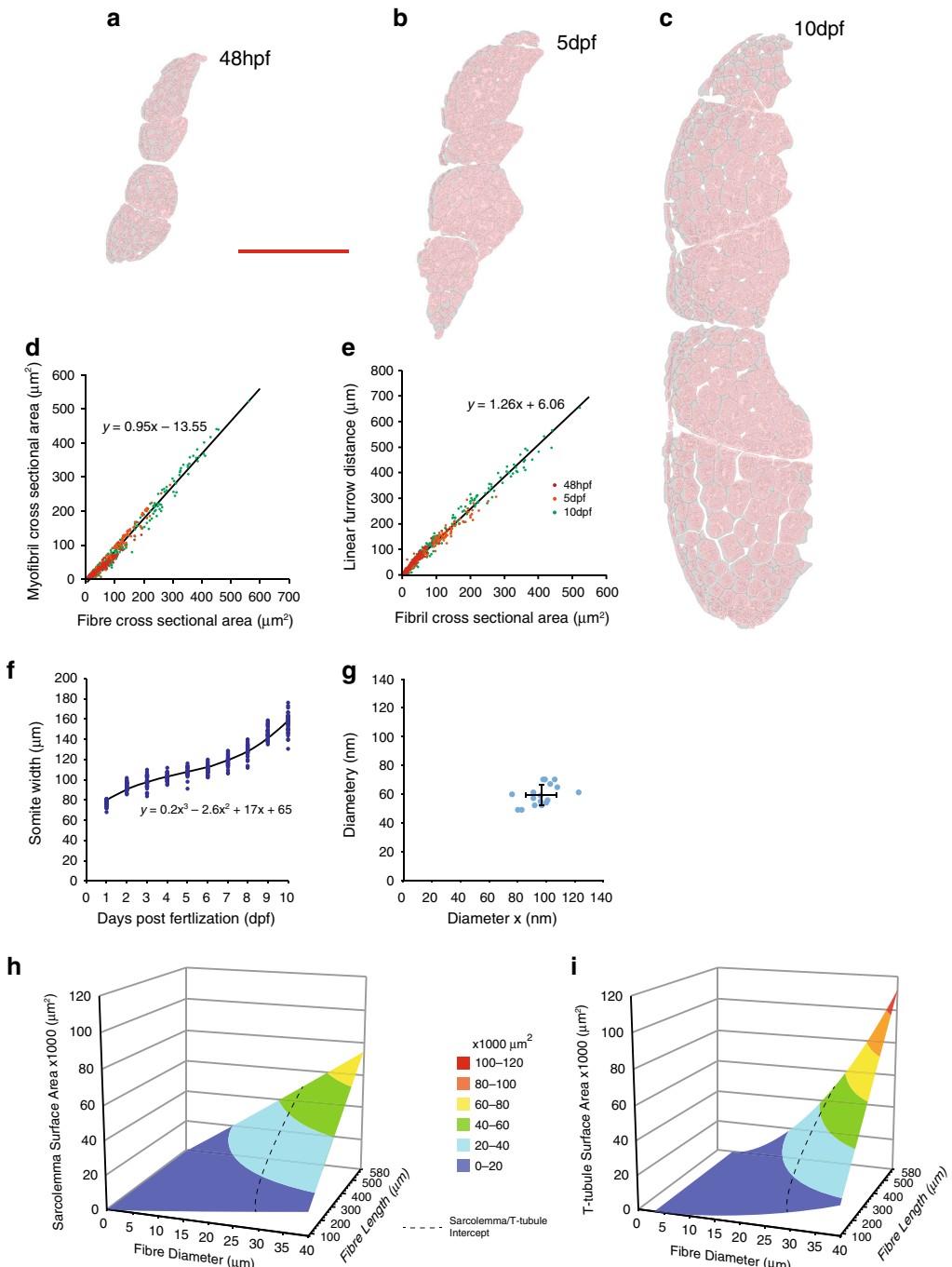

**Fig. 3 Mathematical model of T-tubule development. a–c** Tracing of tiled TEM montages allows extraction of quantitative data. **d** Relationship between fibre cross-sectional area and myofibril cross-sectional area. **e** Relationship of myofibril cross-sectional area to linear furrow distance. **f** Somite width during the first 10 days post fertilisation $n > 25$ per time point. **g** Precise measurements from focussed ion beam volumes shows that T-tubules are elliptical (see also Fig. 2j) $n = 17$ tubules in one fish. **h**, **i** Application of the model for sarcolemmal and T-tubule surface area for muscle fibre sizes up to 40 μm diameter × 580 μm length. This was the maximum measured from a single 1 year old adult male. Dotted line shows the intersect of the two planes. Scale bar; (**a–c**) 50 μm.

narrower tubular connections between the plasma membrane and the stabilised, sarcomere associated tubules (Fig. 5h). In areas where sarcomeres were separated from the plasma membrane by cytoplasm, these connecting tubules frequently looped and formed connections to the sarcolemma some distance from their origin (Supplementary Fig. 5a). Injection of unconjugated horse radish peroxidase into the circulation at 48 hpf also resulted in infiltration into the tubules (Supplementary Fig. 5b, c). Taken together with our analyses of *smyhc1* and *titin* mutants, these data

suggest that the earliest detectable tubules are surface-connected and are therefore analogous to endocytic tubules. In this model, tubules derived from the sarcolemma would be stabilised when they interact with the contractile apparatus either directly, or through specification of the junctional sarcoplasmic reticulum, a process which we term endocytic capture.

**Molecular composition of the early T-tubules.** In order to probe the composition of the early tubules in zebrafish muscle, and to

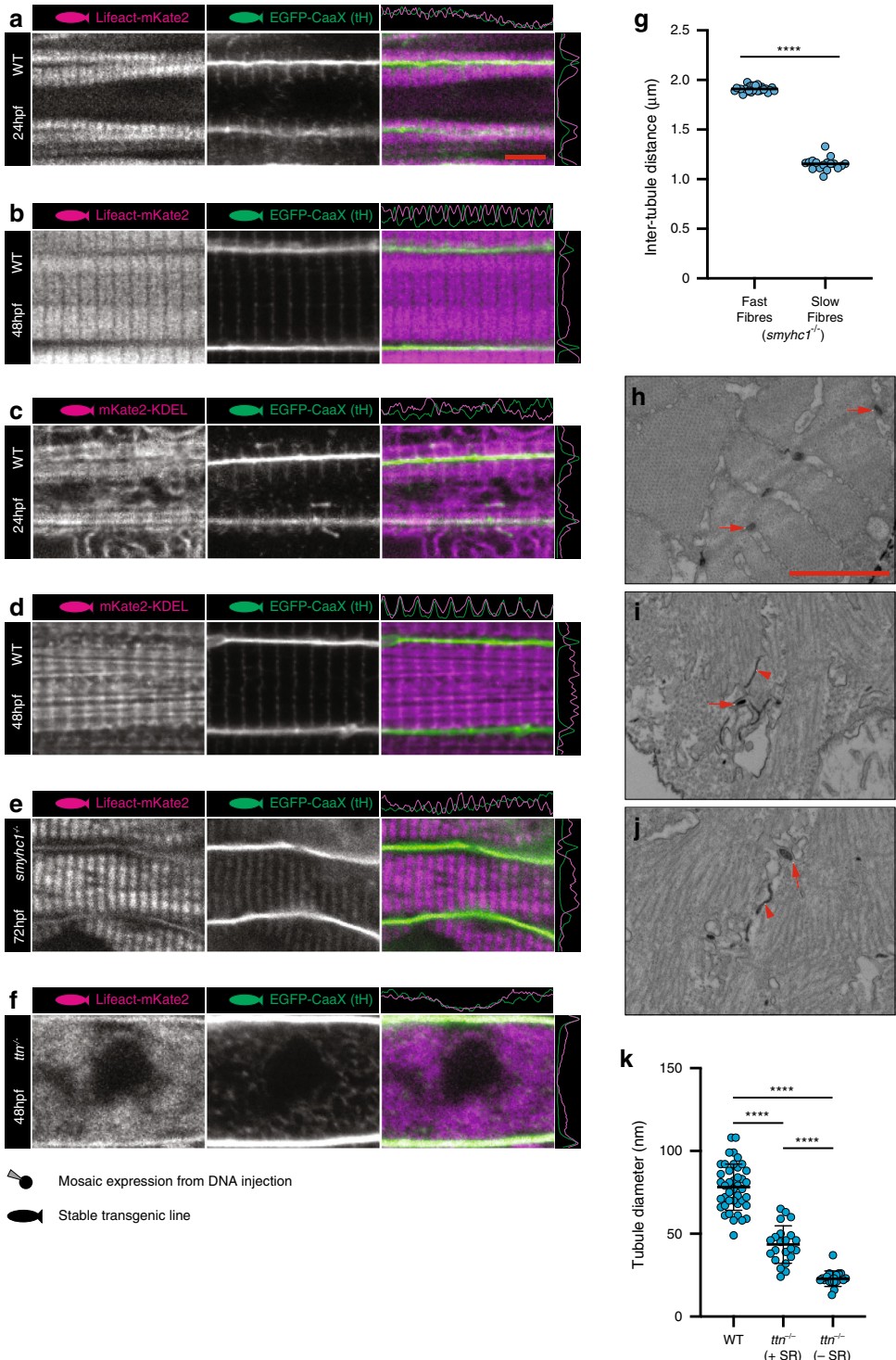

**Fig. 4 Sarcomere formation and sarcoplasmic reticulum. a**, **b** Lifeact (marks actin). See also Supplementary Movie 4. **c**, **d** KDEL (marks endoplasmic/ sarcoplasmic reticulum). **e** *smyhc1*$^{-/-}$ embryos show intact tubules with reduced but regular spacing in slow muscle fibres. **f** *titin*$^{-/-}$ embryos show dysregulated tubule structure. **g** Quantification of inter T-tubule distance in fast and slow fibres from *smyhc1*$^{-/-}$ mutant embryos. Fast fibres $n = 25$, slow fibres $n = 17$ biologically independent animals, one cell per animal over one independent experiment. Two-tailed T-test. Error bars show mean ± SD. **** $p < 0.0001$. **h** Transverse TEM section showing WT tubules (arrows). **i**, **j** Transverse TEM sections from *titin*$^{-/-}$ mutants showing difference in morphology between tubules associated with sarcoplasmic reticulum (arrows) and tubules not associated with sarcoplasmic reticulum (arrowheads). **k** Quantification of sarcoplasmic reticulum associated and non-associated tubules in *titin*$^{-/-}$ mutant embryos. WT $n = 42$, *ttn*$^{-/-}$ +SR $n = 22$, *ttn*$^{-/-}$ −SR $n = 19$ tubules from two biologically independent animals per group, over one independent experiment. One-way ANOVA followed by Tukey's multiple comparison test. Error bars show mean ± SD. ****$p < 0.0001$. (**a**, **c**) 24 hpf; (**b**, **d**, **f**, **k**), 48 hpf (**e g**, **h**, **i**, **j**), 72 hpf. Scale bars (**a–f**) 5 μm, (**h–j**) 1 μm. All images are representative of 12 individual cells within different individual animals.

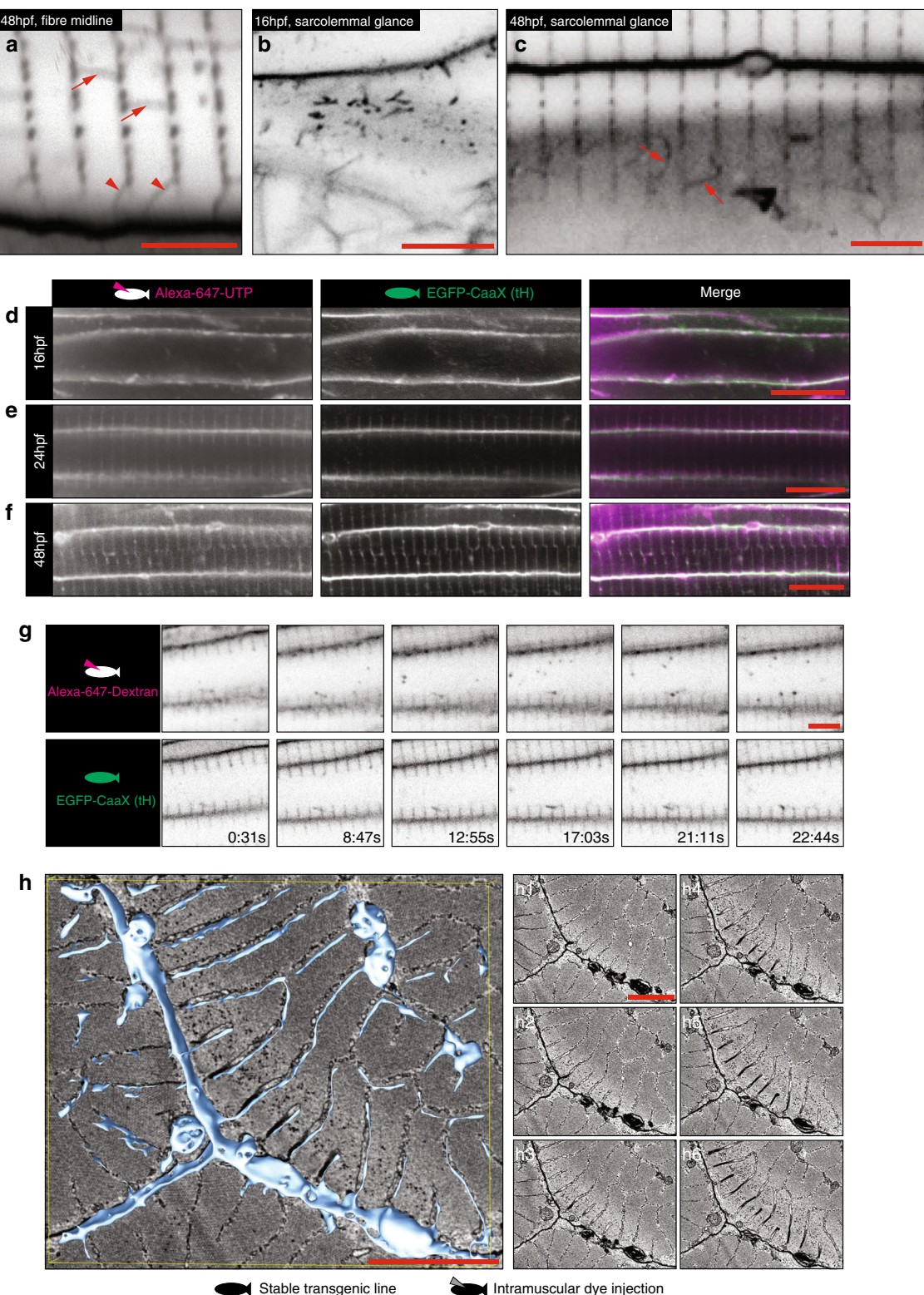

Stable transgenic line    Intramuscular dye injection

gain insight into their biogenesis, we sought to examine the localisation of the zebrafish orthologues of a suite of trafficking and T-tubule associated markers previously defined in mammalian systems (Fig. 6a–j). We developed a set of computational tools which allowed us to perform quantitative analyses of the association of transiently expressed fluorescently tagged proteins with the T-tubules in an unbiased fashion with reasonable throughput. Using this pipeline, we were able to calculate localisation values for defined domains, in this case T-tubule and

sarcolemma (Fig. 6i, j) from large numbers of replicated images using ImageJ (full code is given in Supplementary Software). We were also able to calculate the extent of T-tubule perturbation, by comparing the amplitude of CaaX intensity along the longitudinal axes of the cells (Fig. 6k–m). These analyses showed a striking and selective enrichment of a subset of these proteins; Bin1b, Cavin4a and Cavin4b, EHD1a, Cav3, Dynamin2b and CD44b to the area of the Z-line and junctional SR (Fig. 6a–j, Supplementary Fig. 6a–g). Localisation of these factors to this intracellular

**Fig. 5 Tubules are stabilised by endocytic capture. a–c** Different tubule morphologies by confocal microscopy (EGFP-CaaX, inverted). **a** Longitudinal elements are frequently seen between forming tubules (arrows) and additional elements appear to connect to the sarcolemma (arrowheads). **b** Glancing optical sections across the sarcolemma at 16 hpf immediately before stabilised transverse orientated tubules become visible, show CaaX-positive tubules directly beneath. **c** At 48 hpf putative surface-connected elements (arrows) are frequently seen in glancing optical sections across the sarcolemma. Images are representative of 12 individual cells within different individual animals. **d–f** Intramuscular injection of Alexa-647 conjugated UTP into EGFP-CaaX fish results in immediate infiltration into the developing T-system. (**d**) 16 hpf, (**e**) 24 hpf, (**f**) 48 hpf. Images are representative of three different individual animals. **g** Timelapse microscopy of 10,000 MW-dextran-Alexa-647 injected EGFP-CaaX fish shows immediate infiltration into the tubules, and uptake into intracellular vesicles within 10 min (see also Supplementary Movie 5). Images are representative of three different individual animals. **h** Serial blockface electron microscopy and 3D reconstruction shows tubules connecting stabilised, sarcomere associated tubules to the sarcolemma. Images were derived from two individual animals and images shown are representative of both. See also Supplementary Fig. 5a. Individual Z planes are shown in **h1–h6**. Scale bars; (**a–c**) 5 μm, (**d–f**) 10 μm, (**g**, **h**) 5 μm.

domain suggested an endocytic tubule formation mechanism with similarities to those described in non-muscle cells[22,23]. To test a functional role for these proteins, we generated dominant negative forms of these proteins where possible, and compared the amplitude plots of the CaaX marker as a measure of the extent of perturbation (Fig. 6l, m). We found a marked reduction in amplitude with expression of some of the dominant negative forms, in particular, EHD1a-dEH and EHD1b-dEH (Fig. 6d, m, Supplementary Fig. 6a, b). Notably, in the cases of EHD1a and EHD1b, tubule intensity was also increased when the wildtype proteins were expressed. These analyses demonstrate that endocytic protein function is necessary for the normal development of the T-system and suggests a model where formation occurs by co-option of existing endocytic machinery.

Next, we used a set of well-established probes to examine the phosphoinositide composition of the T-tubules, since different membrane compartments have particularly distinct phosphoinositide signatures. T-tubule phosphoinositide composition has not been directly studied in vivo, although enrichment of PtdIns(4,5)P$_2$ on early tubules in C2C12 cells, and a tenfold increase in total phosphoinositides upon differentiation into myotubes have been demonstrated[10]. Of all probes tested, LactC2/PS, had the highest T-tubule signal (Fig. 7a, h) consistent with a previous biochemical study[24]. We also found substantial levels of BTK/PtdIns(3,4,5)P$_3$ (Fig. 7b, h) and PLC-D/PtdIns(4,5)P$_2$ (Fig. 7c, h) in early T-tubule membranes. Of the other probes used, ING/ PtdIns(5)P showed a small but significant localisation to the tubules (Fig. 7d, h), FAPP1/ PtdIns(4)P was largely confined to the Golgi complex as expected (Fig. 7e), ATG18/ PtdIns(3,5)P$_2$ labelled an internal compartment consistent with lysosomes (Fig. 7f) and 2xFYVE$^{hrs}$/ PtdIns(3)P gave a punctate pattern consistent with early endosomes (Fig. 7g, see also GalT, lamp1 and EEA1, Supplementary Fig. 7a–c, respectively). These data suggest that a defining feature of the early tubules is a distinct lipid composition, which may form a platform on which other proteins are recruited to establish the distinct protein composition.

**Overexpression screening implicates endosomal Rab proteins.** In order to identify specific cellular machinery upregulated or deployed during formation of the T-tubule/SR junction, we performed an overexpression screen of the entire zebrafish Rab family. Rab proteins are key regulators of membrane trafficking, controlling almost every step of vesicle trafficking including formation, transport and fusion. Approximately 60 Rab proteins are involved in different aspects of membrane regulation in mammalian cells and the functions of many of these proteins are well described in non-muscle cells. Indeed, Rab protein overexpression has been shown to have specific functional consequences upon specific membrane trafficking processes, presumably by sequestering key regulators[25], a strategy validated

in other in vivo screening systems[26]. We generated a muscle-specific expression library by cloning each Rab open reading frame behind a muscle-specific promoter and a red fluorescent protein tag. Identified Rab proteins were further tested using specific dominant-acting mutants.

Using this systematic quantitative analysis, we found a subset of Rab proteins with specific localisation to the T-tubule/SR junction (Fig. 8a–e, i). This included Rab33bb, Rab32b and Rab40b which all showed strong and specific localisation to this region (Fig. 8a–c). Closely related paralogues frequently had markedly different profiles confirming the specificity of the screen (e.g. compare Rab33bb to Rab33ba, Fig. 8i). Rab8b specifically localised to the tubules (ranked 10th) but also frequently produced ectopic knots of CaaX-negative tubules extending beyond the leading front of tubule formation (Supplementary Fig. 8b). In this case, overexpression was also associated with a decrease, rather than an increase in T-tubule intensity (Fig. 8i). In contrast, Rab32b, ranked second for localisation, resulted in an approximately 80% increase in CaaX intensity, strongly suggestive of positive regulation of T-tubule formation (Fig. 8i). We found that the most perturbing construct was Rab6a (Fig. 8f, i). Rab6b was not localised to the T-tubules consistent with its role in vesicle fission from the Trans–Golgi Network (TGN)[27]. Strikingly, the next most perturbing group of Rab proteins all belong to a subset of endosomal Rabs, including Rab4a, Rab22a, Rab11ba, Rab5c which are all present on early (Rab22a, 5c) or recycling (Rab11ba, 4a, 22a) endosomes[28–31] (Fig. 8h, i, Supplementary Fig. 8a, Rab11ba and Rab5c images not shown). These effects were confirmed using the lipid marker BODIPY-FLC$_5$-Ceramide (Supplementary Fig. 8c–h) to avoid confounding effects specific to the Hras CaaX domain (note that this analysis showed that Rab23 caused redistribution of the CaaX signal, rather than a specific effect on the T-tubules themselves [compare Fig. 8g with Supplementary Fig. 8d] and was therefore omitted from further analysis). The specific role of the most perturbing Rab proteins (Rab6a, 11ba, 4a, 22a and 5c) on T-tubule development were confirmed by overexpression of the dominant negative and constitutively active variants (Supplementary Fig. 8i).

**An in vitro model of T-tubule formation.** Our in vivo approach allowed identification of rab proteins which localise to the triadic compartment of the sarcomere, where the T-tubules form at the interface of the junctional SR and Z-line. In order to probe further the role of these molecules in a systematic manner, and to validate their triad association, we developed a non-muscle in vitro system based on the readily transfectable baby hamster kidney (BHK) cell line. Upon transfection with zebrafish Bin1b, we found that BHK cells form an extensive plasma membrane-connected tubule system which was able to recruit both the CaaX surface marker, and caveolin3, but not the early endosomal marker snx8[32] (Supplementary Fig. 9e–g).

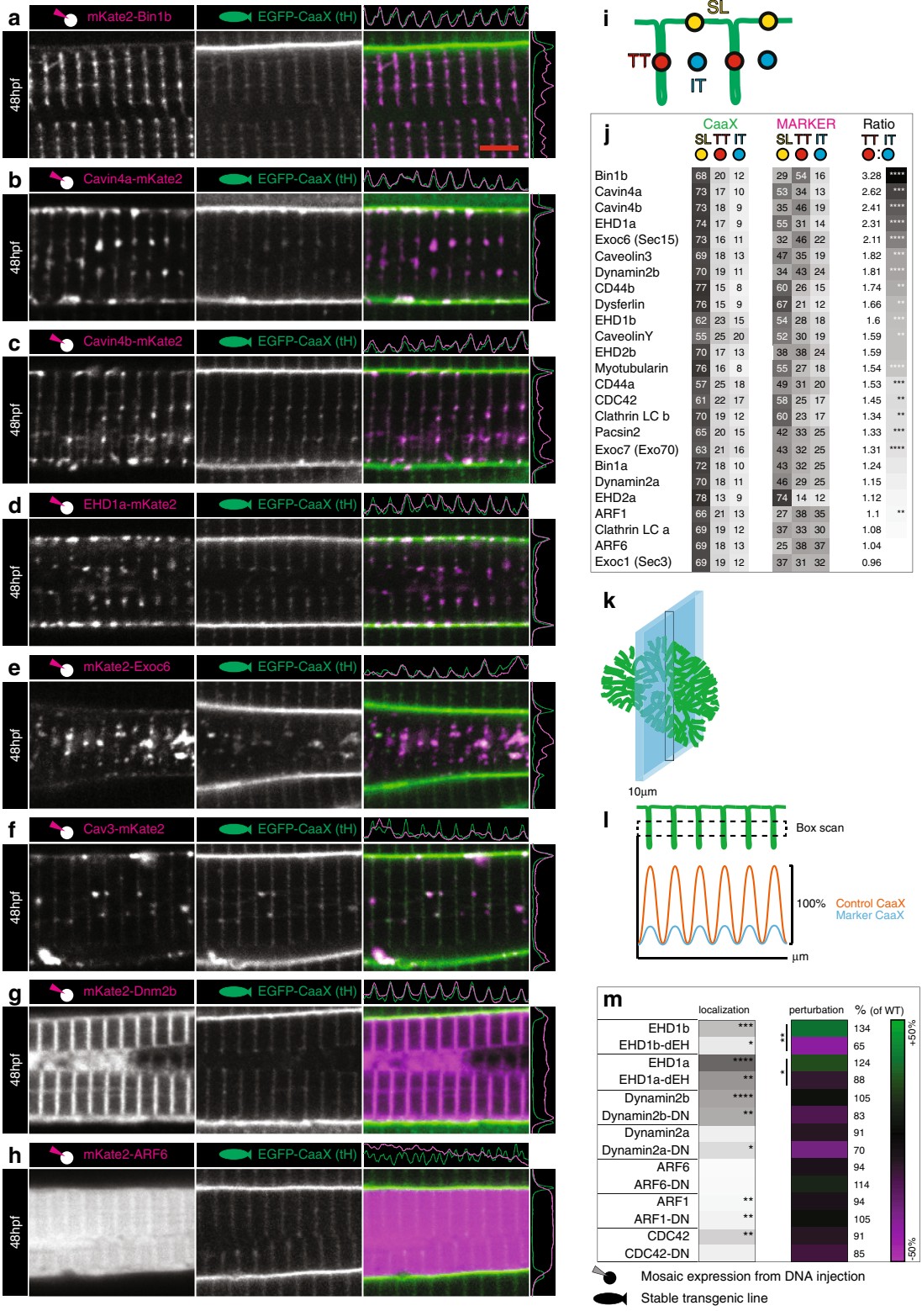

We took the top ten hits for localisation from our in vivo screen, and expressed them in this system. Remarkably, six out of the ten candidates (rab13, 8a, 8b, 33bb, 40b, 26; Fig. 9a–g) co-localised to the induced tubules suggesting that, even in the absence of both sarcomere and junctional SR they are able to be recruited to Bin1b-induced membrane tubules. Two factors (rab32b and 37; Supplementary Fig. 9a, b) marked puncta independent from the Bin1b positive tubules, and a further two did not localise to any discernible structure (rab38a and 19; Supplementary Fig. 9c, d). Combined with the in vivo screen, we can conclude that a subset of Rabs associate with the developing T-tubule/SR junction, and that the majority of these are recruited to the minimal Bin1 tubule in a model system.

**Fig. 6 Composition of early T-tubules.** Images of the top seven markers for T-tubule localisation, compared to a non-tubule marker (ARF6). **a** Bin1b. **b** Cavin4a. **c** Cavin4b. **d** EHD1a. **e** Exoc6. **f** Caveolin3. **g** Dynamin2b. **h** ARF6. Scale bar, 5 µm. **i** Schematic showing domains used for localisation analysis. TT, T-tubule domain; SL, sarcolemma; IT, inter-tubule domain. **j** Markers ranked for T-tubule localisation. **k** Schematic showing the information captured by lateral optical sectioning of T-tubules by confocal microscopy. The average diameter of the T-tubules in the transverse plane was 96 nm when measured precisely by electron microscopy. However these structures can appear to be up as large as 380 nm in the *xy* plane of confocal microscope images due to light scatter and can increase to 800 nm in the *z*-plane. In this schematic, the T-tubules are scaled to a 380 nm diameter, and the optical section is scaled to 800 nm. **l** Schematic showing analysis strategy for assessment of perturbation. A boxed region of interest was placed over an array of eight T-tubules to give the average pixel intensity in the *y* dimension. This method ensured that quantitative assessment of signal was possible even where T-tubules were fragmented or only partially present. For each sample, amplitude was expressed as a percentage of the wildtype (non-expressing) cell from within the same image. **m** Comparison of wildtype and dominant negative variants for proteins amenable to such changes. For perturbation and localisation, *n* = 6 fibres per marker, each from individual biologically independent animals per condition, measured over one independent experiment. Two-tailed T-test. *$p < 0.05$, **$p < 0.01$, ***$p < 0.001$, ****$p < 0.0001$. Exact *p* values are given in source data. All images are representative of 12 individual cells within different individual animals.

## Discussion

In this study, we have defined the molecular mechanisms underpinning vertebrate T-tubule development. Our unique system, utilising live imaging of transgenic zebrafish embryos, three-dimensional electron microscopy, mathematical modelling and semi-automated overexpression screening has allowed dissection of T-tubule formation in vivo, a process which has been refractory to study for decades. We propose that initial tubule formation, immediately following muscle fibre fusion, utilises the machinery of endocytosis. The T-tubule system further develops to become an extensive domain permeating the entire cell and eventually surpasses the sarcolemma in terms of surface area. Mathematical modelling of this process shows that its continuous growth requires rates of membrane synthesis comparable with those of rapidly dividing transformed cell lines. By combining overexpression screening of the entire zebrafish rab protein family with quantitative microscopic analysis we show a role for a Golgi complex small GTPase and a subset of rab proteins specific to the early and recycling endosomes in the expansion of the T-tubule system.

Based on our quantitative 4D analyses, utilising advanced light and electron microscopy, we put forward a testable model for skeletal muscle T-tubule formation comprising of three stages. Firstly, tubules derived from the sarcolemma nucleate on the inner leaflet of the plasma membrane. These early tubules possess markers of clathrin independent endocytosis, in particular CD44, EHD1[33] and dynamin2[34,35]. Furthermore, we were able to reciprocally regulate T-tubule integrity when we over-expressed EHD1a or b in wildtype or dominant negative forms confirming that these proteins play a key role in this process. Secondly, these tubules are stabilised by interactions with the sarcomere, either directly or indirectly via the sarcoplasmic reticulum, a process which we have termed 'endocytic capture'. Our three-dimensional electron microscopy revealed that the spatial positioning of the tubules was dependent upon the splitting of myofibrils to form 'furrows' during growth, a phenomenon described over 30 years ago but which has received little attention since[15]. We present evidence that the sarcomere forms the scaffold for the early T-system, since genetic disruption of the sarcomere has a downstream effect on T-tubule morphology. Scaffolding appears to occur independently of the thick filament, since genetic perturbation by deletion of myosin results only in a change in tubule spacing. Genetic perturbation of titin however, results in dysregulation of T-tubule development such that tubules lose completely any ordered orientation. They are narrower, and form only stochastic junctions with SR, where they are also significantly wider in diameter. This raises the intriguing possibility that T-tubule recruitment might actually be mediated through maturation of SR, and only indirectly by scaffolding from the sarcomere. In differentiating mouse fibres, the junctional SR

appears to form absolutely concomitantly with the transversely oriented tubules[8], and non-transverse tubules, such as longitudinal elements, are not associated with SR. Our data also show that the tubules extending from the sarcomere to the sarcolemma, remain narrow and tortuous, even after the mature architecture is fully developed, raising the possibility that they may be temporally and spatially dynamic, perhaps even remodelling between contraction cycles. Thirdly, because of the prodigious surface area described by our mathematical model, T-tubule development is sensitive to perturbation of trafficking pathways. It appears that vesicles containing fluid-phase dextran might interact with the forming tubules, and our data show that perturbation by overexpression of the endosomal rabs rab11ba, 4a, 22a or 5c, results in significant tubule loss, while the integrity of the sarcomere is preserved. Exactly how this directional membrane traffic into growing tubules is achieved is currently unclear, but a recent study using cultured fibroblasts from centronuclear myopathy patients carrying mutations in myotubularin, demonstrated that the myotubularin-dependent conversion of endosomal PI3P into PI4P is necessary for tethering and fusion of exocytic carriers at the plasma membrane[36]. A similar mechanism regulating vesicle fusion at the T-tubule would provide a common axis for the pathobiology of Bin1-, Dynamin2- or myotubularin-related centronuclear myopathies.

This study provides a paradigm for the formation and stabilisation of the T-tubule system during muscle development and establishes a framework in which to probe specific molecular events, such as causative mutations in human disease aetiology.

## Methods

**Zebrafish husbandry.** Fish were maintainedby University of Queensland Biological Resources[37]. All animals were maintained on the Tuebingen/AB (TAB) background.

**Animal ethics.** All experimental procedures were reviewed and approved by the Molecular Biosciences Animal Ethics Committee at the University of Queensland prior to commencement.

**Stable transgenic fish lines.** The new transgenic lines Bact2-Lifeact-mKate2 and Bact2-mKate2-KDEL were made using the Tol2 system[38,39]. Freshly laid embryos were microinjected with 50ng/ul plasmid vector with 25ng/ul transposase RNA. In order to make our transposase RNA, we generated a new Gateway compatible backbone for in vitro transcription of RNA (pT3TS-Dest_R1-R3) and a new entry clone containing a codon optimised transposase sequence fused to an SV40 nuclear localisation sequence on each end (pME-nls-transposase). These were recombined in an LR reaction with p3E-pA[40] to make pT3TS-nls-transposase. This plasmid was linearised with KpnI and XbaI before in vitro transcription with T3 polymerase[40,41]. All transgenics were crossed until Mendelian frequencies were reached, indicative of single genomic integrations. Stable lines were maintained as heterozygotes. Existing lines used in this study were acta1-EGFP-CaaX (Hras)[13], acta1-Lifeact-EGFP[42], acta1-mCherry-CaaX[42] (Supplementary Table 1).

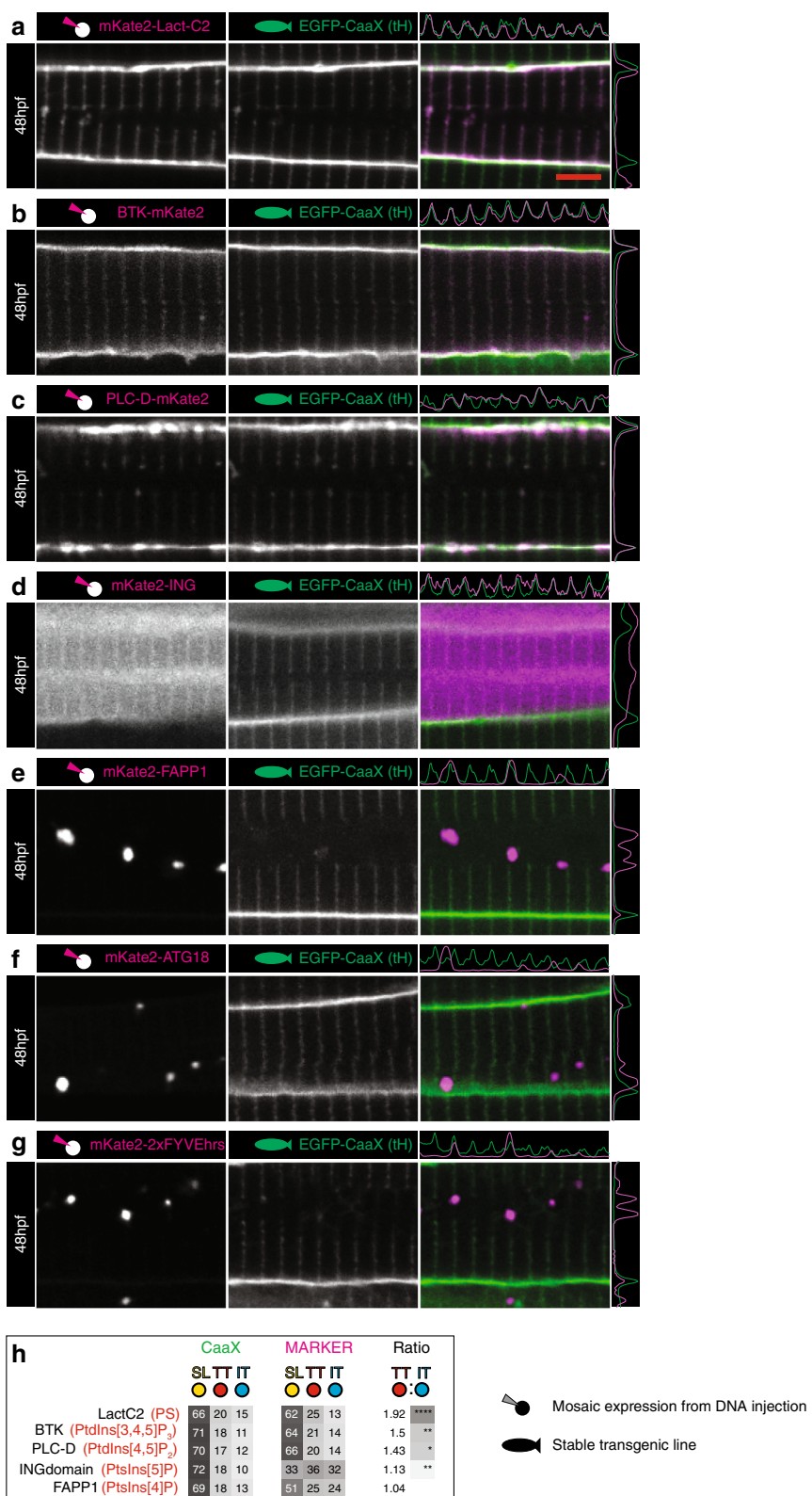

**Fig. 7 Phosphoinositide signature of early T-tubules. a** LactC2 probe for phosphatidylserine (PS). **b** BTK probe for PtdIns(3,4,5)P₂. **c** PLC-D probe for PtdIns(4,5)P₂. **d** ING probe for PtdIns(5)P. **e** FAPP1 probe for PtdIns(4)P. **f** ATG18 probe for PtdIns (3,5)P₂. **g** 2xFYVEhrs probe for PtdIns(3)P. **h** Ranked heatmap comparing average pixel intensity across each domain (T-tubule, sarcolemma and inter-T-tubule domain), for markers of interest and EGFP-CaaX. $n = 6$ fibres per marker, each from individual biologically independent animals per condition, measured over one independent experiment. Two-tailed T-test. $*p < 0.05$, $**p < 0.01$, $***p < 0.001$, $****p < 0.0001$. Exact $p$ values are given in source data. Scale bars, 5 µm. All images are representative of 12 individual cells within different individual animals.

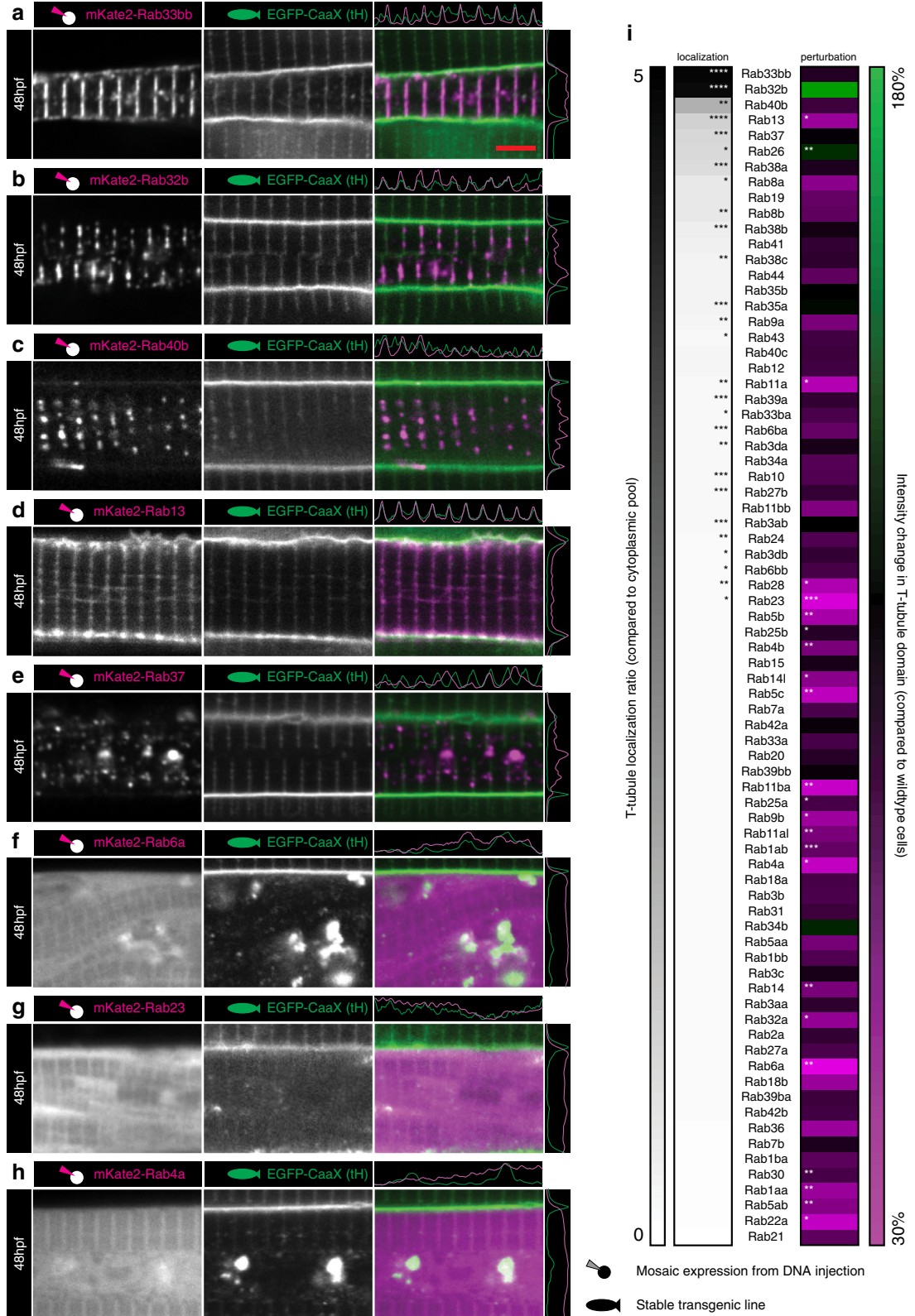

**Fig. 8 An overexpression screen of Rab proteins. a–e** Top five hits for T-tubule localisation. **a** Rab33bb. **b** Rab32b. **c** Rab40b. **d** Rab13. **e** Rab37. **f–h** Top three hits for perturbation of fluorescence amplitude. **f** Rab6a. **g** Rab23. **h** Rab4a. **i** Ranked heatmap showing specific localisation to the T-tubule domain, and the capacity of Rab overexpression to perturb T-tubule formation. $n = 6$ fibres per marker, each from individual biologically independent animals per condition, measured over one independent experiment. Two-tailed T-test. *$p < 0.05$, **$p < 0.01$, ***$p < 0.001$, ****$p < 0.0001$. Exact $p$ values are given in source data. Scale bars, 5 µm. See also Supplementary Fig. 8. All images are representative of 12 individual cells within different individual animals.

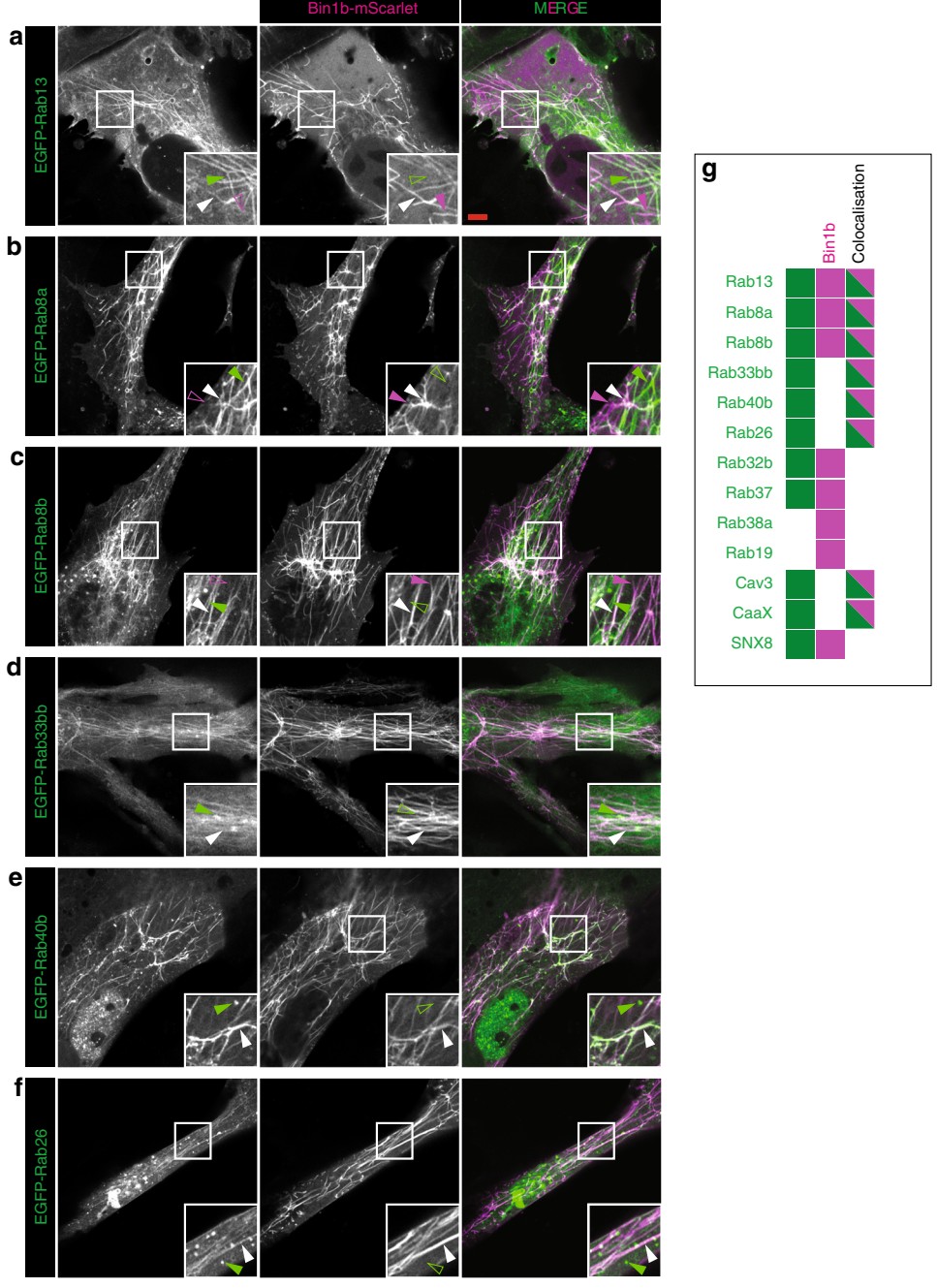

**Fig. 9 An in vitro model of T-tubule formation.** (**a**) Rab13, (**b**) Rab8a, (**c**) Rab8b, (**d**) Rab33bb, (**e**) Rab40b, (**f**) Rab26, (**g**) summary graphic showing cells that possess tubules with Bin1b only (magenta), marker only (green) and tubules where bin1b colocalizes with the marker (dual colour; (see also Supplementary Fig. 9). Images are representative of three individual cells imaged within the same experiment.

**BODIPY staining**. BODIPY vital dye staining was carried by overnight incubation in 5 μg/ml BODIPY-FLC$_5$ (Molecular Probes, catalogue number D3521) followed by a 1 h incubation in 50 μM BODIPY-TR CellTrace (Molecular Probes, catalogue number C34556)[12,21]. Embryos were washed for 5 min in standard embryo media before being mounted for microscopy.

**Intra-cardiac/intramuscular injection**. For fluid-phase marker tracing by electron microscopy, zebrafish embryos were placed in 3.5 cm culture dishes and anaesthetised in tricaine. Approximately 1 nl of 10 mg/ml unconjugated horse radish peroxidase (Sigma-Aldrich P8375) and 5 mg/ml 2000kDa FITC-conjugated Dextran (Thermo-Fisher D7137) dissolved in water was injected into the pre-cardiac sinus at 48 hpf[43]. After 45 min injected embryos were checked for fluorescence in the blood vasculature before being fixed and processed for TEM. For fluid-phase marker tracing by confocal microscopy, approximately 4 nl of 1 mM Alexa-647 conjugated UTP (Thermo-Fisher A32763) was injected into the trunk musculature of 16, 24 and 48 hpf zebrafish[44] and imaged immediately. For small molecular

weight marker tracing by confocal microscopy, approximately 4 nl of 1 mM Alexa-647 conjugated 10,000 MW dextran (Thermo-Fisher D22914) was injected into the trunk muscle of 24 hpf zebrafish and imaged immediately.

**Confocal imaging**. Confocal imaging of zebrafish was performed on a Ziess 710 meta confocal microscope[43]. Zebrafish were mounted in 1% low melting point agarose in embryo media[37], in either a MatTek dish (for inverted microscopes) or a standard 8 cm petri dish for upright microscopes. Objectives used were Zeiss water immersion ×40 N/A 1.0 (catalogue number 420762) on upright microscopes or oil immersion ×40 N/A 1.3 (catalogue number 421462) on inverted microscopes. Embryos were anaesthetized in 2.5 mM tricaine prior to imaging. Confocal imaging of BHK cells was carried out at 37 °C on a Zeiss Inverted LSM 880 with Airyscan using a ×40 Plan Apochromat objective (catalogue number 420762-9800-799). At 18 h post transfection, cell culture medium was replaced with phenol red free DMEM/F12 medium (Invitrogen Australia, 11039-021) containing 10% FBS 1 h prior to microscope imaging.

**Tissue culture**. BHK cells from a baby hamster kidney fibroblast cell line (ATCC CCL-10), were cultured in DMEM (Life Technologies Australia Pty Ltd Invitrogen, 11995073) supplemented with 10% (vol/vol) FBS (GE Healthcare Australia, SH30084.03) and 1% (vol/vol) L-glutamine (Life Technologies Australia, 25030-081)[45]. Cells were seeded onto glass bottom dishes and at 70% confluence were transfected with DNA constructs using Lipofectamine 3000 reagents (Life Technologies Australia Pty Ltd Invitrogen, L3000015). All cells used were between passage 9 and 13 and all cell lines in use in our laboratory are subject to a quarterly mycoplasma testing regime using the Lonza MycoAlert Mycoplasma detection kit (LT07-418).

**Mosaic transgenesis**. Plasmids were delivered to recently laid embryos by microinjection of approximately 4 nl into the yolk at 15 ug/ml without transposase RNA. Embryos were sorted for transgenesis prior to imaging.

**Constructs and cloning**. New ORFs were generated from cDNA and first cloned into pGemTeasy (Promega). Primers were designed using primer3[46] to the boundary between the ORF and the UTR. Expression plasmids were constructed using the Tol2 kit[40]. New entry clones were made by PCR subcloning from existing plasmids using primers containing gateway compatible sites. The multisite gateway system is designed so as to be able to stitch together three cassettes in the correct order and orientation within a backbone. In order to streamline the production of large numbers of constructs using many of the same cassettes, we used a strategy in which we were able to perform the gateway reaction in reverse. That is, to take one of our final expression clones and replace either the middle cassette, 3′ cassette, or both cassettes with the *ccdb* gene. This has the advantage that reactions are able to be performed with a smaller number of component vectors increasing the frequency of correct recombination, and that the largest and most difficult cassettes to recombine (such as promoters) do not need to be recombined in every reaction. In order to do this, we set up gateway reactions with standard volumes and vector concentrations, but only two components; the final expression clone, and the appropriate entry clone containing the *ccdb*/chloramphenicol cassette. In order to perform the LR reaction in reverse, we used BP clonase, transformed into *ccdb* resistant cells, and spread onto plates containing both ampicillin and chloramphicol. In order to substitute both middle and 3′ cassettes at the same time, used our new entry vector pDONR_p1-p3 (synthesised by Genscript, U.S.A.). All new constructs have been made available from Addgene and are listed in Supplementary Table 1.

**Image processing**. In order to avoid bias and ensure reproducibility, we developed a suite of custom macros for ImageJ which we were able to use for the majority of our image processing tasks (Supplementary Table 2). Full code for all macros is given in Supplementary Software. For all in vivo quantitative analyses of localisation or perturbation, six fibres from different individual fish were imaged per condition. For all confocal data sets, we first used Macro 1 "GEN_Split_to_tif_CZI_AND_LSM.ijm" to produce merged .tif files from the native Zeiss .lsm or .czi formats that could be quickly browsed. Where single images are shown, a representative image was selected and then the original confocal file was opened in Fiji (ImageJ), and the raw 16-bit data from each channel was copied and pasted directly into the RGB channels of Adobe Photoshop. Colour-blind compatible images were made by copying and duplicating the red channel into the blue to produce magenta. Brightness and contrast were adjusted as necessary. Intensity traces shown beside individual panels represent average pixel intensities across the entire panel in the $x$ and $y$ planes, with each channel scaled between the maximum and minimum for the channel. The traces are not intended to reflect ratios between channels, rather, these traces give an at a glance representation of the extent to which the two channels correlate, taking into account the limitation that they are averaged across the entire image. Methodology for quantitative comparisons is detailed below.

Fibre morphometrics from tiled transmission electron microscope images were captured by manual tracing into regions of interest (ROIs) using Macro 2 "GEN_Morphometrics.ijm". This macro prevents ROIs from overlapping in the case of fibres by trimming the edges of new ROIs to abut existing ROIs. Myofibrils are trimmed to be within existing fibres, and are automatically named after the parent fibre. Similarly myofibrillar furrow ROIs are trimmed to fall within myofibril ROIs and are named after the parent ROI. Data can then be quickly extracted from the ROI set using the standard measuring tools.

Analysis of perturbation in T-tubule intensity by amplitude was performed in a semi-automated manner. In vivo samples and images can be heterogeneous in intensity and quality. However, our mosaic analysis of single overexpressing fibres in an otherwise wildtype provided a robust an internal control, and allowed us to compare domains within individual images, before comparing between images. First, Macro 3 "GEN_Split_to_green_and_blue_highlight_saturated.ijm" was used to generate .tif images from the original data with enhanced brightness and contrast, and any saturated pixels highlighted. Next, Macro 4 was used to generate sets of box ROIs covering arrays of eight T-tubules from adjacent expressing and non-expressing cells. This part of the analysis could not be automated, since the selection of equivalent areas through the midline of each cell was required. Next, Macro 5 "TT_perturbation_process.ijm" was used to extract the amplitude data from the original, raw confocal files using the ROI sets. Similarly, localisation of markers to the T-tubule domain was achieved using

Macros 6 "TT_GET_circle_ROIs_for_localisation_comparison.ijm" and 7 "GEN_Put_circle.ijm" to define each localisation domain, and Macro 8 "TT_localisation_comparison_process.ijm" to extract the intensity data using the ROI sets. Macros 5 "TT_perturbation_process.ijm" and 8 "TT_localisation_comparison_process.ijm", apply a median filter with a pixel radius of 1 to the raw images before data extraction, to prevent potential noise generated by pixel spikes.

ROI sets were able to be exported to vector based .svg files using the python script "Export ROIs in RoiManager to a SVG file" from the "Python + ImageJ, Fiji Cookbook" at the BioImage Analysis Wiki http://wiki.cmci.info/start.

**Transmission electron microscopy**. Zebrafish were anaesthetised with tricaine (Sigma) in E3 and fixed with 2.5% glutaraldehyde in a Pelco BioWave microwave at 80 W for 6 min under vacuum in a 2 min on, 2 min off, 2 min on, sequence. The head and tail-tips of the zebrafish embryos were removed and the remaining trunk was re-fixed in 2.5% glutaraldehyde in the microwave using the same sequence. Zebrafish were then processed as follows[47]. Samples were washed in PBS and postfixed in 2% $OsO_4$ with 1.5% potassium ferricyanide. Samples were washed in water and incubated in a 1% (w/v) thiocarbohydrazide solution for 20 min. Tissue was postfixed again in 2% $OsO_4$, washed in water and stained with 1% uranyl acetate. Tissue was further stained with a lead aspirate solution (20 mM lead nitrate, 30 mM asparatic acid, pH 5.5), serially dehydrated in acetone, infiltrated with resin and polymerised. Ultrathin sections were cut on a Leica UC6 ultramicrotome and imaged at 80 kV on a JEOL-1011 transmission electron microscope fitted with an Morada 2k × 2k CCD camera under the control of Olympus iTEM software. Embryos injected with HRP (Supplementary Fig. 5b, c) were not stained with potassium ferricyanide. Instead, they were washed with DAB in cacodylate buffer for 1 min and subsequently treated with DAB in cacodylate buffer containing H2O2 for 30 min at room temperature. Embryos were postfixed in 1% $OsO_4$ for 15 min to provide contrast before dehydration and embedding as described above.

**Large-area tiled TEM**. Large-area imaging was performed on a Tecnai T12 G2 (FEI) transmission electron microscope at 120 kV fitted with a Direct Electron LC-1100 4k × 4k CCD camera using the Navigator program in SerialEM[48]. Images were captured with a pixel size of 8 nm and montages were aligned using the blendmont program in IMOD[49]. Images for all time points were captured using the same settings.

**3View Serial blockface scanning electron microscopy**. Samples for serial blockface sectioning scanning electron microscopy were processed as for TEM. Imaging was performed on a Zeiss Gemini FE-SEM fitted with a Gatan 3view, at 2.25 kV, with magnification 7.5k and a pixel size of 5.7 nm. Images were captured at a resolution of 5000 × 5000 (dwell 2), under low vacuum (8 Pa) with 50 nm sections. Image stacks were aligned using the xfalign program in IMOD and automated segmentation was performed by density-based thresholding in the Isosurface render program in IMOD.

**Focussed ion beam electron microscopy**. Following glutaraldehyde fixation and washing samples were placed in 2% osmium tetroxide in 0.1M cacodylate buffer and incubated in a Pelco Biowave at 80 W for 6 min under vacuum in a 2 min on, 2 min off, 2 min on, sequence. The osmium solution was removed without rinsing and a solution of 2.5% potassium ferricyanide was added and placed into the Pelco BioWave for a further 6 min, at 80 W, under vacuum on a 2 min on, 2 min off 2 min on, sequence. Samples were then removed and washed in distilled water 6 times for 2 min. Samples were further incubated in an aqueous filtered solution of 1% thiocarbohydrazide, (6 min under vacuum) in a 2 min off 2 min on sequence. After rinsing in distilled water in a 6 × 2 min sequence, samples were placed in an aqueous 2% osmium tetroxide for 6 min under vacuum in a 2 min on, 2 min off, 2 min on, sequence. After rinsing in distilled water in a 6 × 2 min sequence, samples were placed in 1% uranyl acetate solution and again irradiated at 80 W for 3 min. Samples were removed and washed in distilled water in a 6 × 2 min sequence, and dehydrated through an ethanol series (30, 50, 70, 90 and 100% ethanol for 40 s at 250 W) in the Pelco BioWave, prior to embedding in Durcupan resin. Samples were imaged on an FEI Scios FIB Dual Beam SEM at 1.5 kV with a 0.8 nA electron beam at a resolution of 3072 × 2048, with a chamber vacuum of $(1-2) \times 10^{-4}$ Pa. Slicing was carried out at 30 kV with a 1 nA Ga$^+$ ion beam. Voxel size and slice thickness was 8 nm.

**Statistics and reproducibility**. Linear, second and third degree polynomial regression equations for mathematical modelling (Fig. 3d–f) were generated using the LINEST function in Excel (Microsoft). A one-tailed T-test was used to compare inter T-tubule distance between fast and slow fibres from *smyhc1*$^{-/-}$ mutants (Fig. 4g; Microsoft Excel, T.TEST function for heteroscedastic data). One-tailed T-tests were also used to compare average pixel intensity at the T-tubule domain compared to the inter T-tubule domain (localisation data, Figs. 6j, m, 7h and 8i and Supplementary Fig. 8i; Microsoft Excel, T.TEST function for heteroscedastic data). Significance was tested on all markers showing ≥10% increase in intensity at the T-tubule compared to the inter-tubule domain. Two-tailed T-tests were used to

compare average pixel intensity in overexpressing cells compared to non-expressing cells (perturbation data, Fig. 8i and Supplementary Fig. 8i) and the extent of perturbation of wildtype vs dominant negative markers (Fig.6m; Microsoft Excel, T.TEST function for heteroscedastic data). One-way ANOVA followed by a Tukey multiple comparison test was used to compare tubule diameters between $ttn^{-/-}$ and wildtype fish in the presence or absence of SR association (Fig. 4k; GraphPad Prism 8).

For the non-quantitative data in Figs. 1a–f, 4a–f, 5a–c and 7e–g and Supplementary Figs. 4a–h and 7a–c, and for the quantitative data in Figures Figs. 6–8 and Supplementary Figs. 6 and 8, pilot studies were carried out to assess which compartment markers localised to and the extent of variation between samples from the same condition. All subsequent quantitative and non-quantitative observations were consistent with these initial pilot studies. As such, these experiments were performed independently twice (although direct quantitation was only performed once). The dextran/Alexa-647 injection experiments in Fig. 5d–g were replicated independently three times. Electron microscopy was was performed on a minimum of two animals per experimental condition but independent replication was not carried out due to the technically demanding and labour intensive nature of the experiments.

**Mathematical model**. The mathematical model was built according to the following assumptions.

i.  Each muscle fibre is assumed to be a perfect cylinder, where fibre cross-sectional area is given by the equation for the area of a circle $\pi r^2$, where $r =$ radius. Note that fibre diameter and cross-sectional area are both used in the existing literature as valid measurements of fibre dimensions in muscle cellularity studies. Our rationale in choosing fibre diameter as an input parameter rather than cross-sectional area was to provide the simplest and most applicable model. In our model, fibre cross-sectional area is calculated from fibre diameter ($D$) where

$$\pi\left(\frac{D}{2}\right)^2. \tag{1}$$

ii. Myofibril cross-sectional area is a function of fibre cross-sectional area and can be calculated from the linear regression equation shown in Fig. 3d ($y = 0.95x - 13.55$), where $x$ is muscle fibre cross-sectional area. Combining equations for fibre cross-sectional area and myofibril cross-sectional area allows calculation of myofibril cross-sectional area from fibre diameter ($D$):

$$0.95\pi\left(\frac{D}{2}\right)^2 - 13.55. \tag{2}$$

iii. The T-tubules track through the splits or furrows in the myofibrils, such that at the level of each and every Z-line within a fibre, the total combined length of the myofibrillar furrows in transverse section (illustrated by the red lines in Fig. 2k) will be equal to the total combined length of the associated T-tubules. Moreover, total linear furrow distance can be predicted from the cross-sectional area of the myofibrils within a fibre by the equation $y = 1.26x + 6.06$ (Fig. 3e). Combining equations for myofibril cross-sectional area and total linear furrow distance allows calculation of the length of T-tubule associated with each single Z-line within a muscle fibre, using the diameter area of the fibre ($D$):

$$1.26\left(0.95\pi\left(\frac{D}{2}\right)^2 - 13.55\right) + 6.06. \tag{3}$$

iv. The number of units given by this equation is equal to the number of Z-lines within a muscle fibre. This value can be calculated from fibre length ($L$) and sarcomere length ($s$) using $L/s$. Therefore the total length of Z-line associated T-tubules within a single muscle fibre of length ($L$) and diameter ($D$) is given by

$$\frac{L}{s}\cdot 1.26\left(0.95\pi\left(\frac{D}{2}\right)^2 - 13.55\right) + 6.06. \tag{4}$$

v.  T-tubules are an elliptical tube with a circumference given by Ramanujan's approximation for the perimeter of an ellipse ($p$) where $a$ and $b$ define the major and minor axes, measured from the centre to the perimeter (e.g. in the case of a circle, both $a$ and $b$ would be equal to the radius):

$$p \approx \pi\left(3(a + b) - \sqrt{(3a + b)(a + 3b)}\,\right). \tag{5}$$

In our model Ramanujan's approximation becomes the following, where $d1$ and $d2$ are the major and minor axes of T- diameters:

$$\pi\left(3\left(\frac{d1}{2} + \frac{d2}{2}\right) - \sqrt{\left(3\cdot\frac{d1}{2} + \frac{d2}{2}\right)\left(\frac{d1}{2} + 3\cdot\frac{d2}{2}\right)}\,\right). \tag{6}$$

Dividing by 1000, to convert the output units from nm to µm gives

$$\frac{1}{1000}\pi\left(3\left(\frac{d1}{2} + \frac{d2}{2}\right) - \sqrt{\left(3\cdot\frac{d1}{2} + \frac{d2}{2}\right)\left(\frac{d1}{2} + 3\cdot\frac{d2}{2}\right)}\,\right). \tag{7}$$

vi. The total surface area of the T-tubules within a single muscle fibre of length ($L$) and diameter ($D$) is given by combining the equations in (iv) and (v), using the major and minor axes of the cross section of an elliptical T-tubule ($d1$ and $d2$) and sarcomere length ($s$):

$$\frac{1}{1000}\cdot\pi\left(3\left(\frac{d1}{2} + \frac{d2}{2}\right) - \sqrt{\left(3\frac{d1}{2} + \frac{d2}{2}\right)\left(\frac{d1}{2} + 3\frac{d2}{2}\right)}\,\right)\cdot\frac{L}{s}$$
$$\cdot 1.26\left(0.95\pi\left(\frac{D}{2}\right)^2 - 13.55\right) + 6.06. \tag{8}$$

Simplifying:

$$\frac{1}{1000}\pi\left(3\left(\frac{d1}{2} + \frac{d2}{2}\right) - \sqrt{\left(3\frac{d1}{2} + \frac{d2}{2}\right)\left(\frac{d1}{2} + 3\frac{d2}{2}\right)}\,\right)$$
$$\frac{L}{s}\left(1.26\left(0.95\pi\left(\frac{D}{2}\right)^2 - 13.55\right) + 6.06\right), \tag{9}$$

$$\frac{1}{1000}\pi\left(3\left(\frac{d1}{2} + \frac{d2}{2}\right) - \sqrt{\left(3\frac{d1}{2} + \frac{d2}{2}\right)\left(\frac{d1}{2} + 3\frac{d2}{2}\right)}\,\right)$$
$$\frac{L}{s}\left(\frac{126}{100}\left(\frac{95}{100}\pi\cdot\left(\frac{D}{2}\right)^2 - \frac{1355}{100}\right) + \frac{606}{100}\right), \tag{10}$$

$$\frac{1}{2^3\cdot 5^3}\pi\left(3\left(\frac{d1 + d2}{2}\right) - \sqrt{\left(\frac{3d1}{2} + \frac{d2}{2}\right)\left(\frac{d1}{2} + \frac{3d2}{2}\right)}\,\right)$$
$$\frac{L}{s}\left(\frac{2\cdot 3^2\cdot 7}{2^2\cdot 5^2}\left(\frac{5\cdot 19}{2^2\cdot 5^2}\pi\cdot\frac{D^2}{2^2} - \frac{5\cdot 271}{2^2\cdot 5^2}\right) + \frac{2\cdot 3\cdot 101}{2^2\cdot 5^2}\right), \tag{11}$$

$$\frac{1}{2^3\cdot 5^3}\pi\left(3\cdot\frac{d1 + d2}{2} - \sqrt{\left(\frac{(3d1) + d2}{2}\right)\left(\frac{d1 + (3d2)}{2}\right)}\,\right)$$
$$\frac{L}{s}\left(\frac{3^2\cdot 7}{2^{2-1}\cdot 5^2}\left(\frac{19}{2^2\cdot 5^{2-1}}\pi\cdot\frac{D^2}{2^2} - \frac{271}{2^2\cdot 5^{2-1}}\right) + \frac{3\cdot 101}{2^{2-1}\cdot 5^2}\right), \tag{12}$$

$$\frac{1}{2^3\cdot 5^3}\pi\left(\frac{3(d1 + d2)}{2} - \sqrt{\left(\frac{3d1 + d2}{2}\right)\left(\frac{d1 + 3d2}{2}\right)}\,\right)$$
$$\frac{L}{s}\left(\frac{3^2\cdot 7}{2\cdot 5^2}\left(\frac{19}{2^2\cdot 5}\pi\cdot\frac{D^2}{2^2} - \frac{271}{2^2\cdot 5}\right) + \frac{3\cdot 101}{2\cdot 5^2}\right), \tag{13}$$

$$\frac{1}{2^3\cdot 5^3}\pi\left(\frac{3(d1 + d2)}{2} - \sqrt{\frac{3d1 + d2}{2}\cdot\frac{d1 + 3d2}{2}}\,\right)$$
$$\frac{L}{s}\left(\frac{3^2\cdot 7}{2\cdot 5^2}\left(\frac{19\pi D^2}{2^2\cdot 5\cdot 2^2} - \frac{271}{2^2\cdot 5}\right) + \frac{3\cdot 101}{2\cdot 5^2}\right), \tag{14}$$

$$\frac{1}{2^3\cdot 5^3}\pi\left(\frac{3(d1 + d2)}{2} - \sqrt{\frac{(3d1 + d2)(d1 + 3d2)}{2\cdot 2}}\,\right)$$
$$\frac{L}{s}\left(\frac{3^2\cdot 7}{2\cdot 5^2}\left(\frac{19\pi D^2}{2^2\cdot 2^2\cdot 5} - \frac{271}{2^2\cdot 5}\right) + \frac{3\cdot 101}{2\cdot 5^2}\right), \tag{15}$$

$$\frac{1}{2^3\cdot 5^3}\pi\left(\frac{3(d1 + d2)}{2} - \sqrt{\frac{(3d1 + d2)(d1 + 3d2)}{2^{1+1}}}\,\right)$$
$$\frac{L}{s}\left(\frac{3^2\cdot 7}{2\cdot 5^2}\left(\frac{19\pi D^2}{2^{2+2}\cdot 5} - \frac{271}{2^2\cdot 5}\right) + \frac{3\cdot 101}{2\cdot 5^2}\right), \tag{16}$$

$$\frac{1}{2^3\cdot 5^3}\pi\left(\frac{3(d1 + d2)}{2} - \sqrt{\frac{(3d1 + d2)(d1 + 3d2)}{2^2}}\,\right)$$
$$\frac{L}{s}\left(\frac{3^2\cdot 7}{2\cdot 5^2}\left(\frac{19\pi D^2}{2^4\cdot 5} - \frac{271}{2^2\cdot 5}\right) + \frac{3\cdot 101}{2\cdot 5^2}\right), \tag{17}$$

$$\frac{1}{2^3\cdot 5^3}\pi\left(\frac{3(d1 + d2)}{2} - \frac{\sqrt{(3d1 + d2)(d1 + 3d2)}}{\sqrt{2^2}}\right)$$
$$\frac{L}{s}\left(\frac{3^2\cdot 7}{2\cdot 5^2}\left(\frac{(19\pi D^2) + 4(-271)}{2^4\cdot 5}\right) + \frac{3\cdot 101}{2\cdot 5^2}\right), \tag{18}$$

$$\frac{1}{2^3\cdot 5^3}\pi\left(\frac{3(c + d)}{2} - \frac{\sqrt{(3c + d)(c + 3d)}}{2^{\frac{2}{2}}}\right)$$
$$\frac{L}{s}\left(\frac{3^2\cdot 7}{2\cdot 5^2}\left(\frac{19\pi e^2 - 4\cdot 271}{2^4\cdot 5}\right) + \frac{3\cdot 101}{2\cdot 5^2}\right), \tag{19}$$

$$\frac{1}{2^3 \cdot 5^3} \pi \left( \frac{3(d1+d2)}{2} - \frac{\sqrt{(3d1+d2)(d1+3d2)}}{2} \right)$$
$$\frac{L}{s} \left( \frac{3^2 \cdot 7}{2 \cdot 5^2} \left( \frac{19\pi D^2 - 1084}{2^4 \cdot 5} \right) + \frac{3 \cdot 101}{2 \cdot 5^2} \right), \quad (20)$$

$$\frac{1}{2^3 \cdot 5^3} \pi \left( \frac{(3(d1+d2)) + (-\sqrt{(3d1+d2)(d1+3d2)})}{2} \right)$$
$$\frac{L}{s} \left( \frac{3^2 \cdot 7}{2 \cdot 5^2} \cdot \frac{19\pi D^2 - 1084}{2^4 \cdot 5} + \frac{3 \cdot 101}{2 \cdot 5^2} \right), \quad (21)$$

$$\frac{1}{2^3 \cdot 5^3} \pi \left( \frac{3(d1+d2) + -\sqrt{(3d1+d2)(d1+3d2)}}{2} \right)$$
$$\frac{L}{s} \left( \frac{3^2 \cdot 7(19\pi D^2 - 1084)}{2 \cdot 5^2 \cdot 2^4 \cdot 5} + \frac{3 \cdot 101}{2 \cdot 5^2} \right), \quad (22)$$

$$\frac{1}{2^3 \cdot 5^3} \pi \left( \frac{(3d1+3d2) - \sqrt{(3d1+d2)(d1+3d2)}}{2} \right)$$
$$\frac{L}{s} \left( \frac{3^2 \cdot 7(19\pi D^2 - 1084)}{2 \cdot 2^4 \cdot 5^2 \cdot 5} + \frac{3 \cdot 101}{2 \cdot 5^2} \right), \quad (23)$$

$$\frac{1}{2^3 \cdot 5^3} \pi \left( \frac{3d1+3d2 - \sqrt{(3d1+d2)(d1+3d2)}}{2} \right)$$
$$\frac{L}{s} \left( \frac{3^2 \cdot 7(19\pi D^2 - 1084)}{2^{1+4} \cdot 5^{2+1}} + \frac{3 \cdot 101}{2 \cdot 5^2} \right), \quad (24)$$

$$\frac{1}{2^3 \cdot 5^3} \pi \cdot \frac{3d1+3d2 - \sqrt{(3d1+d2)(d1+3d2)}}{2}$$
$$\cdot \frac{L}{s} \left( \frac{3^2 \cdot 7(19\pi D^2 - 1084)}{2^5 \cdot 5^3} + \frac{3 \cdot 101}{2 \cdot 5^2} \right), \quad (25)$$

$$\frac{1}{2^3 \cdot 5^3} \pi \cdot \frac{3d1+3d2 - \sqrt{(3d1+d2)(d1+3d2)}}{2}$$
$$\cdot \frac{L}{s} \left( \frac{3^2 \cdot 7(19\pi D^2 - 1084) + 80(3 \cdot 101)}{2^5 \cdot 5^3} \right), \quad (26)$$

$$\frac{1}{2^3 \cdot 5^3} \pi \cdot \frac{3d1+3d2 - \sqrt{(3d1+d2)(d1+3d2)}}{2}$$
$$\cdot \frac{L}{s} \left( \frac{(3^2 \cdot 7(19\pi D^2 - 1084)) + 80(3 \cdot 101)}{2^5 \cdot 5^3} \right), \quad (27)$$

$$\frac{1}{2^3 \cdot 5^3} \pi \cdot \frac{3d1+3d2 - \sqrt{(3d1+d2)(d1+3d2)}}{2}$$
$$\cdot \frac{L}{s} \left( \frac{(63(19\pi D^2 - 1084)) + 80 \cdot 303}{2^5 \cdot 5^3} \right), \quad (28)$$

$$\frac{1}{2^3 \cdot 5^3} \pi \cdot \frac{3d1+3d2 - \sqrt{(3d1+d2)(d1+3d2)}}{2}$$
$$\cdot \frac{L}{s} \left( \frac{63(19\pi D^2 - 1084) + 24240}{2^5 \cdot 5^3} \right), \quad (29)$$

$$\frac{1}{2^3 \cdot 5^3} \pi \cdot \frac{3d1+3d2 - \sqrt{(3d1+d2)(d1+3d2)}}{2}$$
$$\cdot \frac{L}{s} \left( \frac{(63 \cdot 19\pi D^2 - 63 \cdot 1084) + 24240}{2^5 \cdot 5^3} \right), \quad (30)$$

$$\frac{1}{2^3 \cdot 5^3} \pi \cdot \frac{3d1+3d2 - \sqrt{(3d1+d2)(d1+3d2)}}{2}$$
$$\cdot \frac{L}{s} \left( \frac{(1197\pi D^2 - 68292) + 24240}{2^5 \cdot 5^3} \right), \quad (31)$$

$$\frac{1}{2^3 \cdot 5^3} \pi \cdot \frac{3d1+3d2 - \sqrt{(3d1+d2)(d1+3d2)}}{2}$$
$$\cdot \frac{L}{s} \left( \frac{1197\pi D^2 - 68292 + 24240}{2^5 \cdot 5^3} \right), \quad (32)$$

$$\frac{1}{2^3 \cdot 5^3} \cdot \frac{3d1+3d2 - \sqrt{(3d1+d2)(d1+3d2)}}{2}$$
$$\cdot \frac{L}{s} \left( \frac{1197\pi D^2 - 44052}{2^5 \cdot 5^3} \right), \quad (33)$$

$$\frac{1}{2^3 \cdot 5^3} \cdot \frac{3d1+3d2 - \sqrt{(3d1+d2)(d1+3d2)}}{2}$$
$$\cdot \frac{L}{s} \left( \frac{(3^2 \cdot 7 \cdot 19)\pi D^2 - 2^2 \cdot 3 \cdot 3671}{2^5 \cdot 5^3} \right), \quad (34)$$

$$\frac{1}{2^3 \cdot 5^3} \pi \cdot \frac{3d1+3d2 - \sqrt{(3d1+d2)(d1+3d2)}}{2}$$
$$\cdot \frac{L}{s} \left( \frac{\left( \frac{3^2 \cdot 7 \cdot 19\pi D^2}{3} - \frac{2^2 \cdot 3 \cdot 3671}{3} \right)}{2^5 \cdot 5^3} \right), \quad (35)$$

$$\frac{1}{2^3 \cdot 5^3} \pi \cdot \frac{3d1+3d2 - \sqrt{(3d1+d2)(d1+3d2)}}{2}$$
$$\cdot \frac{L}{s} \left( \frac{3(3^{2-1} \cdot 7 \cdot 19\pi D^2 - (2^2 \cdot 3671))}{2^5 \cdot 5^3} \right), \quad (36)$$

$$\frac{1}{2^3 \cdot 5^3} \pi \cdot \frac{3d1+3d2 - \sqrt{(3d1+d2)(d1+3d2)}}{2}$$
$$\cdot \frac{L}{s} \left( \frac{3(3 \cdot 7 \cdot 19\pi D^2 - (4 \cdot 3671))}{2^5 \cdot 5^3} \right), \quad (37)$$

$$\frac{1}{2^3 \cdot 5^3} \pi \cdot \frac{3d1+3d2 - \sqrt{(3d1+d2)(d1+3d2)}}{2}$$
$$\cdot \frac{L}{s} \left( \frac{3(399\pi D^2 - 14684)}{2^5 \cdot 5^3} \right), \quad (38)$$

$$\frac{\pi(3d1+3d2 - \sqrt{(3d1+d2)(d1+3d2)})L \cdot 3(399\pi D^2 - 14684)}{2^3 \cdot 5^3 \cdot 2s \cdot 2^5 \cdot 5^3}, \quad (39)$$

$$\frac{3\pi(3d1+3d2 - \sqrt{(3d1+d2)(d1+3d2)})L(399\pi D^2 - 14684)}{2^3 \cdot 2 \cdot 2^5 \cdot 5^3 \cdot 5^3 \cdot s}, \quad (40)$$

$$\frac{3\pi(3d1+3d2 - \sqrt{(3d1+d2)(d1+3d2)})L(399\pi D^2 - 14684)}{2^{3+1+5} \cdot 5^{3+3} \cdot s}, \quad (41)$$

$$\frac{3\pi(3d1+3d2 - \sqrt{(3d1+d2)(d1+3d2)})L(399\pi D^2 - 14684)}{2^9 \cdot 5^6 \cdot s}, \quad (42)$$

$$\frac{3\pi(3d1+3d2 - \sqrt{(3d1+d2)(d1+3d2)})L(399\pi D^2 - 14684)}{512 \cdot 15625s}, \quad (43)$$

$$\frac{3\pi(3d1+3d2 - \sqrt{(3d1+d2)(d1+3d2)})L(399\pi D^2 - 14684)}{8000000s}. \quad (44)$$

Using the values calculated from our experimental data of d1 = 59.24 nm, d2 = 96.47 nm, s = 1.9 μm, this simplifies to

$$\frac{157.939995L\pi(1.197D^2 - 44.052)}{15200}. \quad (45)$$

If cross-sectional area (C) is preferred as an input parameter, which may be of use to some investigators:

$$\frac{3\pi(3d1+3d2 - \sqrt{(3d1+d2)(d1+3d2)})L(399C - 3671)}{2000000s}. \quad (46)$$

Using the values calculated from our experimental data of d1 = 59.24 nm, d2 = 96.47 nm, s = 1.9 μm, this simplifies to

$$\frac{157.939995L\pi(1.197C - 11.013)}{3800} \quad (47)$$

a. The surface area of plasma membrane (sarcolemma) is calculated using equations for the area of a circle $\pi r^2$, where D is the fibre diameter (and D/2 is the radius, r):

$$\pi \left( \frac{D}{2} \right)^2 \quad (48)$$

and has a circumference given by $2\pi r$:

$$2\pi \frac{D}{2}. \quad (49)$$

Multiplying the cross-sectional area by 2 gives the surface area of each of the fibre ends, and multiplying the circumference by the fibre length ($L$), gives the remainder of the surface area. The total surface area, excluding the T-tubules, is therefore given by

$$2\pi\left(\frac{D}{2}\right)^2 + 2\pi L\left(\frac{D}{2}\right). \quad (50)$$

**Application of the mathematical model to 48 hpf, 5 dpf 10 dpf and adult zebrafish.** In order to calculate the surface areas of the T-tubule and sarcolemma at 48 hpf, 5 dpf and 10 dpf, values for fibre length ($L$) were 90.23 μm, 107.77 μm and 158.83 μm respectively, and were calculated using the regression equation for somite width ($y = 0.2x^3 - 2.6x^2 + 17x + 65$; Fig. 3f) for values of $x = 2$, $x = 5$ and $x = 10$. Values for maximum fibre diameter ($D$) were 15.64 μm, 19.23 μm and 26.72 μm respectively (directly measured cross-sectional areas of 192.04 μm², 290.42 μm² and 560.86 μm²) taken from the empirically derived data shown in Fig. 3a–c. Note that these calculations were based on maximum fibre size (rather than the mean size or animal age) since new fibres are continuously added to the myotome and existing fibres grow at different rates (see cellularity profile histograms in Supplementary Fig. 3a–c). To apply this model to the largest fibres found in adult zebrafish we took approximate measurements from the zebrafish atlas images at (http://bio-atlas.psu.edu/zf/view.php?s=220&atlas=18; http://bio-atlas.psu.edu/zf/view.php?s=275&atlas=17). The largest fibres we found in this data set were approximately 580 μm in length and 40 μm in diameter. Applying values of $L = 580$ μm and $D = 40$ μm, gave values of 113083 : 75398 μm²; (T-system : sarcolemma) which corresponds to approximately 1.5x. The graphs in Fig. 3h, i show the consequences of scaling fibre length and diameter up to the maximal values measured from 1 year old fish of $L = 580$ μm and $D = 40$ μm (cross-sectional area = 1256.64 μm²).

In order to find the point at which the surface area of the sarcolemma and the surface area of the T-tubules are equal (the intercept, shown as a dotted line in Fig. 3h, i), we equated the formulae as follows:

$$2\pi\left(\frac{D}{2}\right)^2 + 2\pi L\left(\frac{D}{2}\right) = \frac{157.939995L\pi(1.197D^2 - 44.052)}{15200}. \quad (51)$$

Solving for $L$:

$$\frac{\pi D}{2^2} + 2\pi L\left(\frac{D}{2}\right) = \frac{157.939995L\pi(1.197D^2 - 44.052)}{15200}, \quad (52)$$

$$\frac{\pi D}{2^2} + \pi LD = \frac{157.939995L\pi(1.197D^2 - 44.052)}{15200}, \quad (53)$$

$$\frac{\pi D}{2^2} \cdot 15200 + \pi LD \cdot 15200 = \frac{157.939995L\pi(1.197D^2 - 44.052)}{15200} \cdot 15200, \quad (54)$$

$$7600\pi D^2 + 15200\pi LD = 157.939995\pi L(1.197D^2 - 44.052), \quad (55)$$

$$7600\pi D^2 + 15200\pi LD = 157.939995L \cdot 1.197\pi D^2 \\ - 157.939995L \cdot 44.052, \quad (56)$$

$$7600\pi D^2 + 15200\pi LD = 1.197 \cdot 157.939995\pi\pi LD^2 \\ - 44.052 \cdot 157.939995\pi L, \quad (57)$$

$$7600\pi D^2 + 15200\pi LD = 189.05417\pi^2 LD^2 - 6957.57265\pi L, \quad (58)$$

$$15200\pi LD = 189.05417\pi^2 LD^2 - 6957.57265\pi L - 7600\pi D^2, \quad (59)$$

$$15200\pi LD - 189.05417\pi^2 LD^2 + 6957.57265\pi L = -7600\pi D^2, \quad (60)$$

$$\pi L(15200D - 189.05417\pi D^2 + 6957.57265) = -7600\pi D^2, \quad (61)$$

$$L = \frac{-7600\pi D^2}{\pi(15200D - 189.05417\pi D^2 + 6957.57265)}, \quad (62)$$

$$L = -\frac{7600D^2}{15200D - 189.05417\pi D^2 + 6957.57265}. \quad (63)$$

Solving for D (beginning from Eq. (58)):

$$7600\pi D^2 + 15200\pi LD + 6957.57265\pi L = 189.05417\pi^2 LD^2, \quad (64)$$

$$7600\pi D^2 + 15200\pi LD + 6957.57265\pi L - 189.05417\pi^2 LD^2 = 0, \quad (65)$$

$$(23876.10416 - 1865.88986L)D^2 + 47752.20833LD + 21857.85912L = 0. \quad (66)$$

Applying the quadratic formula gives

$$D = \frac{-47752.20833L \pm \sqrt{47752.20833^2 L^2 - 87431.43649L(23876.10416 - 1865.88986L)}}{2(23876.10416 - 1865.88986L)}. \quad (67)$$

**CRISPR/Cas9.** Guide RNAs were designed using CRISPRscan software[50]. To generate vectors for sgRNA production we modified the pT7-gRNA vector[51] to include a *ccdb* cassette vector (pT7-gRNA-*ccdb*) for negative selection to improve cloning efficiency, which in hands, was low with the original. sgRNA sequences were incorporated by one-step oligo annealing and ligation using BsmBI[51]. DNA templates were generated from these plasmids using a 100 μl PCR (98° for 5 s followed by 35 cycles of 98° for 5 s, 55° for 5 s, 72° for 20 s) and the generic primers sgRNA_F1 (5′-GCAGCTGGCACGACAGGTTTCC-3′) sgRNA_R1 (5′-AAAAAAAAAGCACCGACTCGGTGCCACTTTTTCAAGTT-3′). sgRNAs were generated from the purified amplicons using the Ambion MegaShortScript kit (quarter size reactions). sgRNAs were complexed with EnGen Spy Cas9 NLS protein (New England Biolabs M0646T) for 5 min at 37 °C before injection of approximately 4 nl into the yolk at the one cell stage. Gene-specific sgRNAs were co-injected with an sgRNA targeting tyrosinase[51], to allow selection of founders possessing the highest cutting frequencies. The molar ratio of Cas9 to total sgRNA was 1:1.5. Founders were outcrossed and F1 animals were genotyped by sequencing to identify null mutations and mutant lines were maintained as heterozygotes. sgRNA sequences used were tyr_80[51] (5′-GGACTGGAGGACTTCTGGGG-3′), smyhc1_977 (5′-CAGTGGCAATCAACTCTTCGG-3′) and ttn.1_C3[52] (5′-TACTGGGGAATTCAGTGAAGG-3′).

**Knockout fish lines.** The slow myosin heavy chain mutant generated in this study contains a 2 bp deletion in the slow myosin heavy chain open reading frame (AACGTCAGTGGCAATCAACT[CTdel]TCGGCATCATCAATCGATGC). The titin mutant *ttn.1* generated in this study contains a 2bp deletion/7bp insertion in the titin open reading frame (TGGGTACTGGGGAATTCAGT[GAdel]/[AGTATGGin]AGGCAGTATGGAAACTGCAC).

**Reporting summary.** Further information on research design is available in the Nature Research Reporting Summary linked to this article.

## Data availability

The authors declare that all data supporting the findings of this study are available within the article and its supplementary information files or from the corresponding author upon reasonable request. Data from the zebrafish Bio-Atlas was used in this study (http://bio-atlas.psu.edu/). All plasmids created and used in this study have been made available from a not for profit repository (Addgene, https://www.addgene.org/Rob_Parton/). A full list along with unique identifiers are given in Supplementary Data 1. Source data are provided with this paper.

## Code availability

Custom Fiji (ImageJ) macros are listed in Supplementary Table 2, and full code is given in Supplementary Software.

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

## Acknowledgements

The authors are grateful to Nick Hamilton for advice on image analysis, ImageJ macro programming and equation simplification and Markus Kerr and Alpha Yap for contributing ideas and comments on the manuscript. This work was supported by fellowships and grants from the National Health and Medical Research Council of Australia (to R.G. P., grant numbers 569542 and 1045092; to R.G.P., grant number APP1044041; and to T.E. H. and R.G.P., grant number APP1099251), the Australian Research Council (to T.E.H. and R.G.P., grant number DP200102559) as well as by the Australian Research Council Centre of Excellence in Convergent Bio-Nano Science and Technology (to R.G.P., grant number CE140100036). Confocal microscopy was performed at the Australian Cancer Research Foundation (ACRF)/Institute for Molecular Bioscience (IMB) Dynamic Imaging Facility for Cancer Biology, established with funding from the ACRF. The authors acknowledge the facilities, and the scientific and technical assistance, of the Australian Microscopy & Microanalysis Research Facility at the Centre for Microscopy and Microanalysis, The University of Queensland, in particular, Robyn Webb, Rick Webb, and Hui Diao for serial blockface and focussed ion beam electron microscopy.

## Author contributions

Conceptualisation, T.E.H. and R.G.P.; Methodology, T.E.H., R.G.P, Z.X., N.A., C.F.; Y.L. Validation, T.E.H.; Formal Analysis, T.E.H.; Investigation, T.E.H., N.M., A.X., N.A., J.R. and C.F.; Resources, T.E.H., N.M., H.P.L., Z.X.; Writing, T.E.H.; Visualisation, T.E.H.; Supervision T.E.H. and R.G.P.; Project Administration T.E.H. and R.G.P.; Funding Acquisition T.E.H. and R.G.P.

## Competing interests

The authors declare no competing interests.
