## [Peer Review File · Nature Communications]

Reviewers' Comments:

Reviewer #1:

Remarks to the Author:

This is an interesting manuscript that reports novel findings on the formation and organization of T-tubules in skeletal muscle based on the use of novel technical approaches and knowledge on vesicle trafficking. The authors use a very comprehensive approach to address their questions. The overall experimental procedure is clear and well designed. The results provide significant information to advance our understanding of this process. There are however some points in the manuscript that need to be improved.

Comments

1) The authors have used zebra fish as a model to visualize and analyze formation of T-tubules during muscle fiber development (Fig. 1). They report that most T-tubules extend radially from the surface membrane to the interior of the cells. There is limited evidence of longitudinally-oriented T-tubules in the early stages of myofiber formation. This is in contrast to what demonstrated by other authors, where at initial stages of development T tubules are mostly longitudinal and acquire a transverse orientation only in the first weeks after birth. Do the authors imply that in fishes the orientation of the differentiating T-tubule system differs from that of chicks and mice? This point needs to be clarified.

2) In experiments reported in Fig. 2a, the authors identify "stabilized" and "unstabilized" T-tubules depending on the presence or absence of myofibrils. In these images, there is limited evidence of sarcoplasmic reticulum vesicles next to the T tubules (i.e. so-called peripheral couplings, diads or triads) or to myofibrils, although connections between these compartments have been described both at the morphological levels and supported by molecular identifications of proteins involved (i.e. junctophilins, sAnk1, etc). On the other hand, it is not obvious how can the authors clearly distinguish T-tubules from the sarcoplasmic reticulum as they do in Fig 2a and 2b. I suggest that authors can extend and discuss these points in the context of current knowledge.

3) In Fig. 2e, the authors state that the arrowheads point to sites of contact between T-tubules and myofibrils. It is difficult to accept this explanation as the areas indicated by the arrows might be interpreted also in different ways.

4) Fig 3 reports results obtained following overexpression of Rab proteins where these proteins are ranked based on their localization on T-tubules or by their ability to perturb the organization of the T-tubules. While the reader can appreciate that Rab proteins appear to perturb T-tubules organization as from altered CaaX staining, it cannot be excluded that the Rab proteins that colocalize with the CaaX signal are not positioned on the triadic part of the sarcoplasmic reticulum. Confocal microscope cannot distinguish these two compartments.

5) The data in Supplemental Fig3 f-k would be better appreciated if images for individual protein, and not only merged figures, are presented.

6) The graphic representation with heatmap in figure 4b, Supplemental Fig 4d-e are difficult to follow, figure legends should be more descriptive to guide readers through them.

7) I can agree that myosin is not required for tubules formation, but the effects of latrunculin A appear not so conclusive. Cells in Fig 7j appear swollen/deformed, making it difficult to establish whether the effects of the drug are direct or, as I suspect, a consequence of the rough damage induced to the cells (lines 324-327). Please specify for a more general audience what Lifeact is.

8) While it has been reported that the sarcoplasmic reticulum is connected to the myofibrils, I am not sure that there is evidence to support that tubules are stabilized by interaction with the

sarcoplasmic reticulum, as stated in lines 317-319.

9) Lines 330-334. Can the authors better explain what they are referring to here. Tubules do not extend from the sarcomere to the extracellular milieu, but rather from the sarcolemma to the interior of the cell. The speculation here is perhaps too bold. I think it would be better to remove or change it.

Minor concerns:

- Line 74: Please replace "Fig. & b" with "Fig. a & b"
- Legend to Figure 1: please indicate what symbols mean. In general, figure legends are a bit too simple and do not always help understanding the complexity of all the data reported or the organization of the graphical representations.
- Line 98: Please correct "Fig 2d and d"
- Line 100: please correct "Supplimentary"
- Line 115: please replace Fig. 2K with Fig. 2k.
- Line 172: the term is paralogs and not orthologs
- Line 204: please replace "EHD1-dEH" with "EHD1b-dEH"
- Line 583: please check if "(f)" is correct
- Line 584: please check if "depolarizes" is correct

Reviewer #2:

Remarks to the Author:

In the article, "Muscle T-tubules Form by Endocytic Capture" Robert Parton's team analyzes the early formation of T-tubules in zebrafish embryos. The authors took advantage of a transgenic zebrafish line expressing the general membrane marker EGFP-CaaX to follow the development of T-tubules using timelaps imaging in living animals, parallel electron microscopy, and quantitative image analysis approaches. The methods are state-of-the-art and in part innovative, they are thoroughly described to ensure reproducibility and the ImageJ macros provided in the Supplements represent potentially useful analysis tools provided to the community. The representative figures are of high quality and clearly presented. The supplementary videos further attest to the quality and high potential of the newly introduced experimental paradigm and provide striking example videos of T-tubule development.

Both, a screen of Rab protein T-tubule localization and perturbation, and a protein localization screen indicate the importance of components of the endocytic machinery for T-tubule formation. Labeling analysis of a range of phospholipids revealed the specific lipid composition of early T-tubules compared to the sarcolemma. The concomitant appearance of T-tubules with myofibrils, as well as their dependence on the integrity of the myofibrils suggests the importance of the myofibrils in the regular organization and transverse orientation of the T-tubules. Thus, the authors suggest a three step model for T-tubule development, involving i) formation of dynamic endocytic tubules at transient nucleation sites on the sarcolemma ii) stabilization by myofibrils/sarcoplasmic reticulum and iii) delivery of membrane from the recycling endosome and Golgi complex.

This technically elegant and sound study contains a wealth of valuable experimental data in support of this intriguing model. Nevertheless, the specific and distinct roles of the endocytic machinery in the two proposed phases of T-tubule formation have not been stringently tested and alternative possibilities have not been explored. In particular, the presented data does not allow to differentiate between the possible mechanisms that the early T-tubules are formed by the addition of membranes from recycling endosomes vs. incomplete endocytosis, nor do they compellingly demonstrate the necessary involvement of recycling endosomes and the GOLGI at the third stage of their model.

A major weakness is the presentation of the data. Observations are described like facts, but are frequently not supported by quantitative information or statistical analysis. A handful of experts may see the described phenomena in the example pictures, but to the majority of the reader these descriptions may remain obscure.

Fig. 2 Although I trust the interpretation of the authors, from the text it is not clear how they distinguished T-tubules and SR in those cases where they were not continuous with the sarcolemma or the rough ER and nuclear envelope, respectively. Is it possible to unambiguously identify them without the use of molecular markers or extracellular tracers (e.g. ferritine, UTP, dextran)? If so, this needs to be explained in the manuscript. Also in Fig 7c, how can the authors be sure that these are "surface connected" tubules?

Experiments shown in Fig. 6 using of fluorescent tracers suggest that the earliest T-tubules are already open to the extracellular space. The evidence in support of "subsequent interactions with independent endosomal structures" is less clear. Also the definition of "transient and dynamic nucleation sites in the sarcolemma" is not clear. Figure legends and text fail to point out what to look for in the images and videos. If the authors want to derive mechanistic conclusions from these observations, the narratives need to be substantiated by quantitative or at least semi-quantitative image/video analyses. Also consider what alternative explanations for the observed phenomena might be possible and how these possibilities can be experimentally excluded?

Considering the importance of the presented video data, separate legends explaining the videos should be added to the Supplements.

Fig. 7 I,h Although the example figure is convincing, demonstration of different T-tubule spacing requires quantitative analysis or at least some information on how often the observation has been made. Furthermore, the lower magnification of the control (7i) does not allow assessment of the T-tubule spacing.

Minor

41 „neuronal“ APs end at the neuro-muscular junction. T-tubules propagate muscle APs

117 what was the contribution of tubule morphometrics from focused ion beam EM to development of the quantitative model?

132-143 It is unclear what the comparison of the rate of membrane synthesis in the GOLGI (single cell) compares to in multi-nucleated myofibers. Does the combined increase in surface area per minute refer to a single fiber, the analyzed fiber segment (what volume?) or to the entire fish? Explain! The prediction that this requires upregulation of the membrane synthesis machinery at the onset of T-tubule formation could easily be answered by analyzing the mass of GOLGI in thin sections of the relevant stages.

583 "...control shown in (f)" does not seem correct – supplementary figure 6f ?

651 avoid the use of the term "correlates" if correlation has not been assessed quantitatively

Fig 1i,j,k quantitative analysis from only one or two fish is rather thin.

Introduction and Discussion: the authors bemoan that T-tubule formation has been understudied. They should still cite some of the relevant old literature and discuss it in light of their new findings.

Reviewer #3:

Remarks to the Author:

I have been asked to look at the use of electron microscopy within this paper. I do not pretend to be an expert in zebrafish models, or muscle ultra structure, but I hope that I can give some feedback that you will find useful.

This paper examines the formation of t-tubules in a novel zebrafish model. The paper shows an impressive amount of work that has been collated in order to examine the role of endocytic recycling in the development of the muscle t-tubule system. The paper uses a combination fluorescent imaging with electron microscopic studies to examine the role of endocytosis associated Rab proteins in the tubule formation. The paper concludes that there is a significant increase in membrane needed to generate the tubules and that this comes from the endocytic pathway.

In the paper's current form it is difficult to work out how valid these conclusions are as many of the details of your experiments have been left out. I do hope that you will find my comments useful. By necessity they are negative as I am trying to point out the parts where I have struggled, but I hope that you will be able to re-shape this work in order to clarify the work being presented.

My main concern is with the mathematical model of the membrane surface area and the conclusions drawn from this. The model is based on your electron microscope data, but you do not tell us what the pixel size/imaging conditions (that can affect ultimate resolution in the block face imaging techniques) of this data are.

I would like to draw your attention to the coastline paradox

(https://en.wikipedia.org/wiki/Coastline_paradox), which states that the higher the resolution that you use to measure the outline the longer the apparent outline will be. In your case you have used a mixture of traces from TEM images of the 'furrows' to get fibre length with FIB-SEM images of the cross sections to give the perimeter of the ellipse (It would have been nice to see the FIB-SEM data to prove that the cross-section was an ellipse and not circular, but sectioned obliquely). These two data sets are measured at quite different resolutions (I assume). I'm not sure if they can even be used together like this, but I'm not a mathematician so can't really comment here. Another issue is that it is unclear how you then got the sarcolemma surface area, as a simple cylinder (as implied in sup fig 2d) seems to over simplify data especially as you have traced the sarcolemma already (though a 3D trace would be more accurate). The information in supplementary figure 2 d-f [sic] is not complete enough for me to judge accurately.

It would be helpful for you to explain the long formula in e): From my understanding you are working out how many sarcolemma to model (L/s), a correction for the micron vs nanometer (1/1000) the perimeter of the ellipse ($\pi(3... \text{etc})$) and the complicated part to calculate the furrow length based on the proportion of the cross section that is myofibrillar and the length of furrow that this would surround (Why is this based on plots for all the data, not age related plots?).

In the end you have a plot showing the increase in surface area with cross-sectional area, while I would have thought that it would be more useful to plot the increase with age, particularly as you seem to have traces for 16hpf to 10dpf. One of your conclusions is that the membrane grows at a rate of $1.70\mu\text{m}^2/\text{min}$ (line 133), and maximal rate is $2.7\mu\text{m}^2/\text{min}$ in transformed cells (line 136). However I do not know how you came up with this number. You mention isometric scaling (all dimensions growing at the same rate), yet I'm not sure if this is relevant (do the cells grow like this, or do they elongate with age?). I'm also not sure if this is what you are plotting in fig2I (ie when you double the length in your model did you double the diameter, or double the cross sectional area ($=1.41\times$ diameter (square root of 2))?)

The paper that defines the maximal growth rate (NB this is referenced in your ref 14: Griffiths G, Fuller SD, Back R, Hollinshead M, Pfeiffer S, Simons K. The dynamic nature of the Golgi complex. J Cell Biol. 1989 Feb;108(2):277-97.) uses stereology to generate the numbers, which is quite an involved process using random sections. You may well be able to use your SBFSEM data to generate similar numbers, but I have to admit that I find stereology quite daunting (I have seen

John Lucocq talk about using block face data for this sort of thing. Eg doi: 10.1007/s00418-017-1564-6). I don't think that it is appropriate to compare your model of membrane formation with data from a stereological approach, though part of me wonders whether you are in fact under-estimating the sarcolemma fraction? Hopefully you can adapt your model to better fit the data as the trend is almost certainly important.

Aside from this major issue I have some smaller comments/questions that may be useful for you. As an electron microscopist you will spot a natural bias:

- You use the term 'pseudo camera lucida', which I had to look up. I feel that 'tracing' is perfectly acceptable.
- The SBFSEM and FIBSEM methods could do with information on imaging conditions, pixel size, voltage, slice thickness vacuum etc.
- The TEM method is slightly unusual. My main question is why you used TCH without a second osmium. I'd always thought that the TCH was a mordant that bound the osmium from the first round of staining and allowed more osmium to bind there, thus increasing the stain density (doi: 10.1038/nprot.2011.439) Again there is no mention of the magnification (or pixel size) of the images. Especially whether they were standardised across the time points.
- How are you differentiating sarcoplasmic reticulum from tubules in the TEM images? Also when tracing the 'furrows' how are you able to differentiate SR from Tubules? (Perhaps I have missed the point here?)
- The image and movie of the 3view data (fig 2 f-h) is not clear that you can see the ruffles mentioned in the main text. The movie is far too fast, but you can just about see them at the end. Have you tried modelling a smaller portion in order to highlight a few sarcomeres?
- Figure 3 (and supplemental figure). I presume that the images are from mosaic fish as only the central cell shows over expression. This is not stated in the figure legend.
- Figure 6 d+e it does not state that the images are from 48hpf tissue and it is not clear from the text that the images are there to show the increased cytoplasm in 6e. In the serial section panels D1-D6 are you sure that the order is correct as the dark features better fit sections in the vertical order (D1,D3,D5,D2,D4,D6). The same is true for E1-E8 (also note that you only describe E1-6 in the figure legend).
- Have you quantified the number of tubules found near premyofibrils as opposed to found at random (fig 7d), or are these just examples that have been found?

As a final, potentially naïve, question. Have you looked at the expression pattern of other proteins/organelles in the Rab over expressing cells? Ie do they still possess sarcomeres or has the over expression had a wider affect on cellular organisation? I'd be interested to see TEM images of these cells to look at the changes in the endosomes, but I realise that this would be nearly impossible due to the use of mosaic expression.

Reviewer #4:

Remarks to the Author:

Overall it is a interesting and potentially significant study that is impressive in sheer number of fish lines and high-tech approaches/analyses. However, manuscript is very descriptive and simply screens (by overexpression) through large number of membrane trafficking proteins and lipids without making an effort in actually performing deeper analysis of some of "hits" from the screen. For example, identification of several Rabs that specifically localize to T-tubules are by far most interesting result (Rab33bb, Rab32b, Rab40b). However, there is no follow-up on these Rabs. Instead, authors propose "endosome capture" model without much direct data to support it. Authors then continue to investigate lipids as well as other semi-randomly associated proteins for their localization and possible effect on T-tubule formation. I think manuscript would be much stronger and more suitable for publication if authors would use their excellent tools and techniques to focus on dissecting the effects of Rabs identified in first screen. As is stands, in my opinion, the manuscript is too descriptive and not novel enough to be published in Nature Communications.

- 1) Figure 3. While overexpression of specific Rabs is good approach to analyze their localization, the "perturbation" as a measurement of Rab involvement in T-tubule formation is much less useful. For one, overexpression of Rabs often does not directly affect the function of interest. Even if one does see an effect, it is not immediately clear whether effect is direct. That is especially true for over-expressing endocytic Rabs, such as Rab11, Rab4 or Rab6. That may also explain why the Rabs that best localized with T-tubules, did not give highest "perturbation". Authors at the very least should express the dominant-negative forms of Rab33bb, Rab32b and Rab40b since those appear to be most likely regulators of T-tubules. Ideally, one should make a CRISPR-KO lines.
- 2) Authors state that "selective enrichment of Bin1b, Cavin4a, Cav3, EHD1a, dynamin2b and CD44b ... suggest an endocytic foirmation mechanism of T-tubules". Not clear why they reach this conclusion. From listed proteins only EHD1a and to some extend dynamin 2b would indicate that. Rest of listed proteins actually argue against this conclusion.
- 3) Figure 5. The lipid analysis in T-tubules does not really fit well with the story and is either not very conclusive or novel. Many findings, such as enrichment in PS has been previously reported. Enrichment in PIP2 and PIP3 would be expected since T-tubules appear to be extensions of plasma membrane. Finally, it is not clear what mechanistic or developmental conclusions are reached in this analysis.
- 4) "Endocytic capture" model is largely hypothetical and is not really directly supported by any shown data. Authors present an endosome model in Figure 8c, yet no direct evidence/testing of that model presented. Overexpression of Rab5c, Rab4a, Rab11a or Rab22a dominant negative mutants or creation a CRISPR-KO lines is needed. Furthermore, what happened to Rabs that colocalize with T-tubules (Rab33bb, Rab32b, Rab40b)? Why they are not part of the model? They also need to be tested using DN-mutants or CRISPR-KO before any model can be proposed.

Other minor comments:

- 1) Very small font and color-coded labels in Figure 3C-J, Figure 4C-F are very hard to see. Authors should increase the font and change to "black-and-white" color scheme.

Reviewers' comments

Reviewer #1 (Remarks to the Author):

This is an interesting manuscript that reports novel findings on the formation and organization of T-tubules in skeletal muscle based on the use of novel technical approaches and knowledge on vesicle trafficking. The authors use a very comprehensive approach to address their questions. The overall experimental procedure is clear and well designed. The results provide significant information to advance our understanding of this process. There are however some points in the manuscript that need to be improved.

Comments

1) The authors have used zebra fish as a model to visualize and analyze formation of T-tubules during muscle fiber development (Fig. 1). They report that most T-tubules extend radially from the surface membrane to the interior of the cells. There is limited evidence of longitudinally-oriented T-tubules in the early stages of myofiber formation. This is in contrast to what demonstrated by other authors, where at initial stages of development T tubules are mostly longitudinal and acquire a transverse orientation only in the first weeks after birth. Do the authors imply that in fishes the orientation of the differentiating T-tubule system differs from that of chicks and mice? This point needs to be clarified.

Response:

This is a fascinating point very pertinent to our study which we alluded to in the discussion of our original manuscript and have now expanded upon and clarified. In mice the initial tubules are longitudinally arranged and start to become transverse around birth (a process which is complete around post-natal day 10). As such the association of T-tubules with the sarcomere is delayed in amniote systems compared with the zebrafish. We know from work in mice (Franzini-Armstrong 1991) and in rat myoblasts (Cusimano et al. 2009) the T-tubules form concomitantly with the junctional SR later in development than the sarcomere. In fish, junctional SR specification, T-tubule formation and sarcomere formation occur simultaneously. This section of our discussion now reads:

“Genetic perturbation of titin however, results in dysregulation of T-tubule development such that tubules lose completely any ordered orientation. They are narrower, and form only stochastic junctions with SR, where they are also significantly wider in diameter. This raises the intriguing possibility that T-tubule recruitment might actually be mediated through maturation of SR, and only indirectly through scaffolding from the sarcomere. In differentiating mouse fibres, the junctional SR appears to form absolutely concomitantly with the transversely oriented tubules, and non-transverse tubules, such as longitudinal elements, are not associated with SR.”

2) In experiments reported in Fig. 2a, the authors identify “stabilized” and “unstabilized” T-tubules depending on the presence or absence of myofibrils. In these images, there is limited evidence of sarcoplasmic reticulum vesicles next to the T tubules (i.e. so-called peripheral couplings, diads or triads) or to myofibrils, although connections between these compartments have been described both at the morphological levels and supported by molecular identifications of proteins involved (i.e. junctophilins, sAnk1, etc). On the other hand, it is not obvious how can the authors clearly distinguish T-tubules from the sarcoplasmic reticulum as they do in Fig 2a and 2b. I suggest that authors can extend and discuss these points in the context of current knowledge.

Response:

We agree, and this point was in fact raised by all three reviewers. We have addressed this point in several ways. Firstly, we explain the EM technique we use to specifically distinguish between T-tubules and SR, and provide further examples (see panels a to f in Figure 2, and specifically compare the insets from panels d [SR] and f [tubule]). The appropriate section in the main text now reads:

“We used a modified version of a potassium ferricyanide stain on thin transmission electron microscopy (TEM) sections to specifically label the T-tubules and distinguish them from sarcoplasmic reticulum (SR). Sections taken at 16hpf showed electron dense, transverse oriented tubular membranes associated with the forming Z-lines (arrows Fig. 2a). By contrast, transverse sections taken towards the centre of the sarcomere some distance away from the Z-lines showed association of longitudinal sarcoplasmic reticulum (identifiable

by its beaded appearance and lower electron density) with the early myofibrillar material (arrowheads, Fig. 2c)."

Secondly, we provide further analysis of "unstabilised" vs "stabilised" tubules using new knockout lines of sarcomeric components, demonstrating that in the absence of sarcomere/SR association, tubules possess a distinct, narrower morphology (detailed in Fig 4). Thirdly, we have developed an entirely new in vitro system based on BHK cell expression of the tubulating protein bin1. We show that even in the absence of sarcomere or junctional SR in non-muscle cells in culture, we still see association of six out of 10 rab proteins with Bin1 induced, surface connected tubules. These new data are presented in Fig. 9.

3) In Fig. 2e, the authors state that the arrowheads point to sites of contact between T-tubules and myofibrils. It is difficult to accept this explanation as the areas indicated by the arrows might be interpreted also in different ways.

Response:

This was poorly worded in the original manuscript. It was not our intention to suggest that these are points of contact between the sarcomere and the T-tubules; indeed such direct connections may not exist. We merely intended to convey the concept that the plane of sectioning intersects the Z-line, and this is where we find the (ferricyanide stained, electron dense) T-tubules. The tubules are adjacent to the SR which occupies the rest of the space between the z-lines. This has been clarified in the text and the reader is now also referred to the 3D renderings which make the visualisation of the structure clearer. The figure legend now reads:

"(a-f) Thin transverse transmission electron microscope (TEM) sections showing early tubules and sarcoplasmic reticulum. (a, b) Transverse sections through the Z line (Z) at 16 and 48hpf. Electron dense, ferricyanide stained tubules are denoted by arrows. (c, d) Equivalent transverse sections taken through the centre of the sarcomere. Arrowheads denote sarcoplasmic reticulum, which has an unstained lumen and "beaded" appearance. (e, f) Oblique sections at 5 and 10dpf show both ferricyanide stained, electron dense tubules (arrows) and sarcoplasmic reticulum (arrowheads). M, M-line, Z, Z-line."

4) Fig 3 reports results obtained following overexpression of Rab proteins where these proteins are ranked based on their localization on T-tubules or by their ability to perturb the organization of the T-tubules. While the reader can appreciate that Rab proteins appear to perturb T-tubules organization as from altered CaaX staining, it cannot be excluded that the Rab proteins that colocalize with the CaaX signal are not positioned on the triadic part of the sarcoplasmic reticulum. Confocal microscope cannot distinguish these two compartments.

Response:

This is of course entirely true, and we have modified the text throughout the manuscript to make this point clear; the localisation screen pertains to tubule-junctional SR complex rather than just the T-tubule. Most importantly, this comment prompted us to provide independent validation of T-tubule association using a model system in which we assessed whether specific Rab proteins co-localise with Bin1b induced tubules in BHK cells (Fig 9). This provides independent validation of association with Bin-positive tubules. As a result of these analyses we believe that two of our hits from this screen, rab32b and rab38a are likely to be localised to the junctional SR rather than the T-tubule.

5) The data in Supplemental Fig3 f-k would be better appreciated if images for individual protein, and not only merged figures, are presented.

Response:

We have changed these panels in accordance with the reviewer's suggestion.

6) The graphic representation with heatmap in figure 4b, Supplemental Fig 4d-e are difficult to follow, figure legends should be more descriptive to guide readers through them.

Response:

We have changed these panels and legends in accordance with the reviewer's suggestion.

7) I can agree that myosin is not required for tubules formation, but the effects of latrunculin A appear not so conclusive. Cells in Fig 7j appear swollen/deformed, making it difficult to establish whether the effects of the

drug are direct or, as I suspect, a consequence of the rough damage induced to the cells (lines 324-327).
Please specify for a more general audience what Lifeact is.

Response:

We agree. Although we believe this to be the case, our further analyses of the effects of Latrunculin on sarcomere depolymerisation could not rule out the possibility that this was a non-specific effect. However, we now include a far more comprehensive investigation of the effects of knock-out of sarcomere components in the form of stable titin and slow myosin mutants. These new data are presented in Figure 4.

8) While it has been reported that the sarcoplasmic reticulum is connected to the myofibrils, I am not sure that there is evidence to support that tubules are stabilized by interaction with the sarcoplasmic reticulum, as stated in lines 317-319.

Response:

This is true, and we have made significant experimental efforts to probe this question more deeply and obtain direct evidence for this. Our new EM and light microscope data on the Titin mutant (Figure 4) shows us that in the absence of junctional SR, Tubules have a narrower morphology and lose their regular, transverse orientation.

9) Lines 330-334. Can the authors better explain what they are referring to here. Tubules do not extend from the sarcomere to the extracellular milieu, but rather from the sarcolemma to the interior of the cell. The speculation here is perhaps too bold. I think it would be better to remove or change it.

Response:

This sentence was poorly worded in the original manuscript. Here, we were specifically referring to the subsarcolemmal tubules that connect the plasma membrane to the sarcomere associated tubules. We have changed this sentence to read:

“Our data also shows that the tubules extending from the sarcomere to the sarcolemma, remain narrow and tortuous, even after the mature architecture is fully developed...”

Minor concerns:

- Line 74: Please replace “Fig. & b” with “Fig. a & b”
- Legend to Figure 1: please indicate what symbols mean. In general, figure legends are a bit too simple and do not always help understanding the complexity of all the data reported or the organization of the graphical representations.
- Line 98: Please correct “Fig 2d and d”
- Line 100: please correct “Supplimentary”
- Line 115: please replace Fig. 2K with Fig. 2k.
- Line 172: the term is paralogs and not orthologs
- Line 204: please replace “EHD1-dEH” with “EHD1b-dEH”
- Line 583: please check if “(f)” is correct
- Line 584: please check if “depolarizes” is correct

Response:

All minor concerns have been changed in accordance with the reviewer’s suggestion. The terms “orthologues” and “paralogues” have been checked throughout the manuscript and changed as appropriate.

--

Reviewer #2 (Remarks to the Author):

In the article, “Muscle T-tubules Form by Endocytic Capture” Robert Parton’s team analyzes the early formation of T-tubules in zebrafish embryos. The authors took advantage of a transgenic zebrafish line expressing the general membrane marker EGFP-CaaX to follow the development of T-tubules using timelaps imaging in living animals, parallel electron microscopy, and quantitative image analysis approaches. The methods are state-of-the-art and in part innovative, they are thoroughly described to ensure reproducibility and the ImageJ macros provided in the Supplements represent potentially useful analysis tools provided to the community. The representative figures are of high quality and clearly presented. The supplementary videos further attest to the quality and high potential of the newly introduced

experimental paradigm and provide striking example videos of T-tubule development.

Both, a screen of Rab protein T-tubule localization and perturbation, and a protein localization screen indicate the importance of components of the endocytic machinery for T-tubule formation. Labeling analysis of a range of phospholipids revealed the specific lipid composition of early T-tubules compared to the sarcolemma. The concomitant appearance of T-tubules with myofibrils, as well as their dependence on the integrity of the myofibrils suggests the importance of the myofibrils in the regular organization and transverse orientation of the T-tubules. Thus, the authors suggest a three step model for T-tubule development, involving i) formation of dynamic endocytic tubules at transient nucleation sites on the sarcolemma ii) stabilization by myofibrils/sarcoplasmic reticulum and iii) delivery of membrane from the recycling endosome and Golgi complex.

This technically elegant and sound study contains a wealth of valuable experimental data in support of this intriguing model. Nevertheless, the specific and distinct roles of the endocytic machinery in the two proposed phases of T-tubule formation have not been stringently tested and alternative possibilities have not been explored. In particular, the presented data does not allow to differentiate between the possible mechanisms that the early T-tubules are formed by the addition of membranes from recycling endosomes vs. incomplete endocytosis, nor do they compellingly demonstrate the necessary involvement of recycling endosomes and the GOLGI at the third stage of their model.

Response:

We agree. The intention of this study was to provide sound experimental evidence for an endocytic capture mechanism for T-tubule formation. Further, we provide a robust experimental framework in which the specifics of different endocytic mechanisms can now be tested. We have refocused our discussion to allow the reader to appreciate this, and our future studies will specifically focus on these questions.

A major weakness is the presentation of the data. Observations are described like facts, but are frequently not supported by quantitative information or statistical analysis. A handful of experts may see the described phenomena in the example pictures, but to the majority of the reader these descriptions may remain obscure.

Response:

We have tightened the data presentation in all figures considerably, and provide quantitative information for the majority of panels in accordance with the reviewer's suggestion.

Fig. 2 Although I trust the interpretation of the authors, from the text it is not clear how they distinguished T-tubules and SR in those cases where they were not continuous with the sarcolemma or the rough ER and nuclear envelope, respectively. Is it possible to unambiguously identify them without the use of molecular markers or extracellular tracers (e.g. ferritine, UTP, dextran)? If so, this needs to be explained in the manuscript. Also in Fig 7c, how can the authors be sure that these are "surface connected" tubules?

Response:

We have addressed the first point with extensive revisions. Please see the response to comment 2 from Reviewer #1. The specific CaaX-labelled tubules in figure 7c (now Figure 5c) we cannot be sure are surface connected. However in this revised figure we now show direct evidence of surface connected tubules by fluid phase marker infiltration (light microscopy Fig. 5d-f; TEM Fig S5b & c). We also show direct connections by renderings of three dimensional electron microscope volumes (Fig 5g, and S5a).

Experiments shown in Fig. 6 using of fluorescent tracers suggest that the earliest T-tubules are already open to the extracellular space. The evidence in support of "subsequent interactions with independent endosomal structures" is less clear. Also the definition of "transient and dynamic nucleation sites in the sarcolemma" is not clear. Figure legends and text fail to point out what to look for in the images and videos. If the authors want to derive mechanistic conclusions from these observations, the narratives need to be substantiated by quantitative or at least semi-quantitative image/video analyses. Also consider what alternative explanations for the observed phenomena might be possible and how these possibilities can be experimentally excluded?

Response:

We agree, and we have made substantial changes to this aspect of the manuscript. We have removed the description "transient and dynamic" as quantitative analyses of these specific characteristics was not

possible. As described above, we have also characterised morphologically distinct sarcomere/SR associated tubules vs narrower, non-sarcomere associated tubules in sarcomeric mutants and wildtype fish. We have expanded the figure legends, and removed the reference to “interactions with independent endosomal structures” since this remains hypothetical.

Considering the importance of the presented video data, separate legends explaining the videos should be added to the Supplements.

Response:

We have added separate legends for the videos to the supplements in accordance with the reviewers suggestion.

Fig. 7 I,h Although the example figure is convincing, demonstration of different T-tubule spacing requires quantitative analysis or at least some information on how often the observation has been made. Furthermore, the lower magnification of the control (7i) does not allow assessment of the T-tubule spacing.

Response:

We have tightened the data presentation in all figures considerably, and provide quantitative information for these panels (and others) in accordance with the reviewer’s suggestion. See Figure 4, panel e.

Minor

41 „neuronal“ APs end at the neuro-muscular junction. T-tubules propagate muscle APs

117 what was the contribution of tubule morphometrics from focused ion beam EM to development of the quantitative model?

132-143 It is unclear what the comparison of the rate of membrane synthesis in the GOLGI (single cell) compares to in multi-nucleated myofibers. Does the combined increase in surface area per minute refer to a single fiber, the analyzed fiber segment (what volume?) or to the entire fish? Explain! The prediction that this requires upregulation of the membrane synthesis machinery at the onset of T-tubule formation could easily be answered by analyzing the mass of GOLGI in thin sections of the relevant stages.

583 “...control shown in (f)” does not seem correct – supplementary figure 6f ?

651 avoid the use of the term “correlates” if correlation has not been assessed quantitatively

Fig 1i,j,k quantitative analysis from only one or two fish is rather thin.

Introduction and Discussion: the authors bemoan that T-tubule formation has been understudied. They should still cite some of the relevant old literature and discuss it in light of their new findings.

Response:

All minor comments have been addressed as requested. Note that the contribution of the focussed ion beam data to the mathematical model was to provide direct measurements of T-tubule width. This part of the manuscript has been extensively revised and examples shown. Similarly, the details of the mathematical model have been expanded and clarified.

--

Reviewer #3 (Remarks to the Author):

I have been asked to look at the use of electron microscopy within this paper. I do not pretend to be an expert in zebrafish models, or muscle ultra structure, but I hope that I can give some feedback that you will find useful.

This paper examines the formation of t-tubules in a novel zebrafish model. The paper shows an impressive amount of work that has been collated in order to examine the role of endocytic recycling in the development of the muscle t-tubule system. The paper uses a combination fluorescent imaging with electron microscopic

studies to examine the role of endocytosis associated Rab proteins in the tubule formation. The paper concludes that there is a significant increase in membrane needed to generate the tubules and that this comes from the endocytic pathway.

In the paper's current form it is difficult to work out how valid these conclusions are as many of the details of your experiments have been left out. I do hope that you will find my comments useful. By necessity they are negative as I am trying to point out the parts where I have struggled, but I hope that you will be able to re-shape this work in order to clarify the work being presented.

My main concern is with the mathematical model of the membrane surface area and the conclusions drawn from this. The model is based on your electron microscope data, but you do not tell us what the pixel size/imaging conditions (that can affect ultimate resolution in the block face imaging techniques) of this data are.

I would like to draw your attention to the coastline paradox (https://en.wikipedia.org/wiki/Coastline_paradox), which states that the higher the resolution that you use to measure the outline the longer the apparent outline will be. In your case you have used a mixture of traces from TEM images of the 'furrows' to get fibre length with FIB-SEM images of the cross sections to give the perimeter of the ellipse (It would have been nice to see the FIB-SEM data to prove that the cross-section was an ellipse and not circular, but sectioned obliquely). These two data sets are measured at quite different resolutions (I assume). I'm not sure if they can even be used together like this, but I'm not a mathematician so can't really comment here. Another issue is that it is unclear how you then got the sarcolemma surface area, as a simple cylinder (as implied in sup fig 2d) seems to over simplify data especially as you have traced the sarcolemma already (though a 3D trace would be more accurate). The information in supplementary figure 2 d-F [sic] is not complete enough for me to judge accurately.

It would be helpful for you to explain the long formula in e): From my understanding you are working out how many sarcolemma to model (L/s), a correction for the micron vs nanometer (1/1000) the perimeter of the ellipse ($\pi(3... \text{etc})$) and the complicated part to calculate the furrow length based on the proportion of the cross section that is myofibrillar and the length of furrow that this would surround (Why is this based on plots for all the data, not age related plots?).

In the end you have a plot showing the increase in surface area with cross-sectional area, while I would have thought that it would be more useful to plot the increase with age, particularly as you seem to have traces for 16hpf to 10dpf. One of your conclusions is that the membrane grows at a rate of $1.70\mu\text{m}^2/\text{min}$ (line 133), and maximal rate is $2.7\mu\text{m}^2/\text{min}$ in transformed cells (line 136). However I do not know how you came up with this number. You mention isometric scaling (all dimensions growing at the same rate), yet I'm not sure if this is relevant (do the cells grow like this, or do they elongate with age?). I'm also not sure if this is what you are plotting in fig2l (ie when you double the length in your model did you double the diameter, or double the cross sectional area ($=1.41 \times \text{diameter}$ (square root of 2))?)

The paper that defines the maximal growth rate (NB this is referenced in your ref 14: Griffiths G, Fuller SD, Back R, Hollinshead M, Pfeiffer S, Simons K. The dynamic nature of the Golgi complex. J Cell Biol. 1989 Feb;108(2):277-97.) uses stereology to generate the numbers, which is quite an involved process using random sections. You may well be able to use your SBFSEM data to generate similar numbers, but I have to admit that I find stereology quite daunting (I have seen John Lucocq talk about using block face data for this sort of thing. Eg doi: 10.1007/s00418-017-1564-6). I don't think that it is appropriate to compare your model of membrane formation with data from a stereological approach, though part of me wonders whether you are in fact under-estimating the sarcolemma fraction? Hopefully you can adapt your model to better fit the data as the trend is almost certainly important.

Response:

We have expanded and clarified this section of the manuscript substantially and consulted a mathematician, Dr Nick Hamilton to verify this portion of the manuscript. We give full details including the pixel size and imaging conditions and now include the focussed ion beam data showing elliptical tubules as requested. The coastline paradox in this instance, is not relevant, since all perimeter measurements were made on tiled TEM sections at the same resolution, using the same technology. Other measurements using focussed ion beam and indeed light microscopy were made using point-to-point straight line measurements only. In terms of scaling of individual muscle fibre growth, although it might seem intuitive to plot fibre growth with age it is not logical or straightforward. New fibres are added throughout development, meaning that at any time there is a particular frequency distribution of new and old fibres, and of fibre diameters. This point has been clarified and expanded upon in the text, and the new histograms added to figure S3 demonstrate this point.

Fibre length is more predictable since all fibres run from myoseptum to myoseptum, meaning that for a horizontal fibre, fibre length is equal to somite width (a plot of somite width during the first ten days is now shown in figure 3f). Thus, some fibres will indeed grow isometrically, while others will not. Our new analyses including some adult fibres has shown that actually fibres grow proportionally more lengthways than by hypertrophy. The revised output from the model is given in Fig. 3h.

Aside from this major issue I have some smaller comments/questions that may be useful for you. As an electron microscopist you will spot a natural bias:

- You use the term ‘pseudo camera lucida’, which I had to look up. I feel that ‘tracing’ is perfectly acceptable.

Response:

This has been changed in accordance with the reviewer’s suggestion.

- The SBFSEM and FIBSEM methods could do with information on imaging conditions, pixel size, voltage, slice thickness vacuum etc.

Response:

These data have been added in accordance with the reviewer’s suggestion

- The TEM method is slightly unusual. My main question is why you used TCH without a second osmium. I’d always thought that the TCH was a mordant that bound the osmium from the first round of staining and allowed more osmium to bind there, thus increasing the stain density (doi: 10.1038/nprot.2011.439) Again there is no mention of the magnification (or pixel size) of the images. Especially whether they were standardised across the time points.

Response:

The reviewer is correct and this was a mistake in the original manuscript; in fact there were two osmium steps. We have revised the methods accordingly and apologise for the error. The magnification for the tiled TEM was standardised between samples. The pixel size was 8nm and is given in the methods section.

- How are you differentiating sarcoplasmic reticulum from tubules in the TEM images? Also when tracing the ‘furrows’ how are you able to differentiate SR from Tubules? (Perhaps I have missed the point here?)

Response:

Please see point 2 from Reviewer #1

- The image and movie of the 3view data (fig 2 f-h) is not clear that you can see the ruffles mentioned in the main text. The movie is far too fast, but you can just about see them at the end. Have you tried modelling a smaller portion in order to highlight a few sarcomeres?

Response:

We have decreased the frame rate of this movie substantially. The reader should now be able to appreciate that the tubules form within “splits” or “furrows” within the myofibrils.

- Figure 3 (and supplemental figure). I presume that the images are from mosaic fish as only the central cell shows over expression. This is not stated in the figure legend.

Response:

Yes, we have clarified this and other examples in the figure legends.

- Figure 6 d+e it does not state that the images are from 48hpf tissue and it is not clear from the text that the images are there to show the increased cytoplasm in 6e. In the serial section panels D1-D6 are you sure that the order is correct as the dark features better fit sections in the vertical order (D1,D3,D5,D2,D4,D6). The same is true for E1-E8 (also note that you only describe E1-6 in the figure legend).

Response:

Yes! Many thanks to the reviewer for noticing this error. These labels have now been changed.

- Have you quantified the number of tubules found near premyofibrils as opposed to found at random (fig 7d), or are these just examples that have been found?

Response:

These early FIB images have now been removed from the revised manuscript as we now use the sarcomeric mutants to substantiate this point.

As a final, potentially naïve, question. Have you looked at the expression pattern of other proteins/organelles in the Rab over expressing cells? Do they still possess sarcomeres or has the over expression had a wider effect on cellular organisation? I'd be interested to see TEM images of these cells to look at the changes in the endosomes, but I realise that this would be nearly impossible due to the use of mosaic expression.

Response:

This is a very interesting point and is one we may address in the future. However, for this manuscript all Rabs were expressed under a muscle specific promoter, so are only expressed specifically in the skeletal muscle fibres.

--

Reviewer #4 (Remarks to the Author):

Overall it is an interesting and potentially significant study that is impressive in sheer number of fish lines and high-tech approaches/analyses. However, manuscript is very descriptive and simply screens (by overexpression) through large number of membrane trafficking proteins and lipids without making an effort in actually performing deeper analysis of some of "hits" from the screen. For example, identification of several Rabs that specifically localize to T-tubules are by far most interesting result (Rab33bb, Rab32b, Rab40b). However, there is no follow-up on these Rabs. Instead, authors propose "endosome capture" model without much direct data to support it. Authors then continue to investigate lipids as well as other semi-randomly associated proteins for their localization and possible effect on T-tubule formation. I think manuscript would be much stronger and more suitable for publication if authors would use their excellent tools and techniques to focus on

dissecting the effects of Rabs identified in first screen. As it stands, in my opinion, the manuscript is too descriptive and not novel enough to be published in Nature Communications.

1) Figure 3. While overexpression of specific Rabs is good approach to analyze their localization, the "perturbation" as a measurement of Rab involvement in T-tubule formation is much less useful. For one, overexpression of Rabs often does not directly affect the function of interest. Even if one does see an effect, it is not immediately clear whether effect is direct. That is especially true for over-expressing endocytic Rabs, such as Rab11, Rab4 or Rab6. That may also explain why the Rabs that best localized with T-tubules, did not give highest "perturbation". Authors at the very least should express the dominant-negative forms of Rab33bb, Rab32b and Rab40b since those appear to be most likely regulators of T-tubules. Ideally, one should make a CRISPR-KO lines.

Response:

We agree and in our original manuscript we included overexpression of dominant negative and constitutively active rab mutant proteins. These data are also in our revised manuscript in Figure S8. Furthermore, in our restructured narrative we now place the rab screen towards the end, as perhaps the most "speculative" of the data we show.

2) Authors state that "selective enrichment of Bin1b, Cavin4a, Cav3, EHD1a, dynamin2b and CD44b ... suggest an endocytic formation mechanism of T-tubules". Not clear why they reach this conclusion. From listed proteins only EHD1a and to some extent dynamin 2b would indicate that. Rest of listed proteins actually argue against this conclusion.

Response:

We agree and in order to focus this figure more on trafficking pathways, we have excluded Rho and Rac from these analyses, which are potentially mediating actin dynamics. Instead we include components of the exocyst complex which is involved in the tethering of secretory vesicles to the plasma membrane. We find

specific localisation of sec15 to the triad/T-tubule junction. Sec15 has been shown in other systems to bind rab GTPases on secretory vesicles before SNARE-mediated fusion.

3) Figure 5. The lipid analysis in T-tubules does not really fit well with the story and is either not very conclusive or novel. Many findings, such as enrichment in PS has been previously reported. Enrichment in PIP2 and PIP3 would be expected since T-tubules appear to be extensions of plasma membrane. Finally, it is not clear what mechanistic or developmental conclusions are reached in this analysis.

Response:

It is true that enrichment of PS has been shown by biochemical methods in mature muscle in other systems. However, our analyses is novel in the context of development, and feel that it is important within the context of our model to demonstrate a distinct phosphoinositide profile.

4) "Endocytic capture" model is largely hypothetical and is not really directly supported by any shown data. Authors present an endosome model in Figure 8c, yet no direct evidence/testing of that model presented. Overexpression of Rab5c, Rab4a, Rab11a or Rab22a dominant negative mutants or creation a CRISPR-KO lines is needed. Furthermore, what happened to Rabs that colocalize with T-tubules (Rab33bb, Rab32b, Rab40b)? Why they are not part of the model? They also need to be tested using DN-mutants or CRISPR-KO before any model can be proposed.

Response:

As detailed in our cover letter, we went on to make 32 different rab knockout lines (2 alleles for each of 16 rabs). None of these knockout lines showed any phenotype different to wildtype, which could be seen as unsurprising given recent reports of functional redundancy between rab proteins. However, our new *in vitro* BHK cell screen has demonstrated that even in a non-muscle cell line in the absence of SR or sarcomeric proteins, 6 out of the top ten hits for localisation are still recruited to Bin1-induced tubules. Furthermore two rabs which do not localise to Bin1-induced tubules, rab32b and rab38a are known ER-localised proteins in other systems, raising the intriguing possibility that these play a role in specification of the junctional SR.

Other minor comments:

1) Very small font and color-coded labels in Figure 3C-J, Figure 4C-F are very hard to see. Authors should increase the font and change to "black-and-white" color scheme.

Response:

This has been changed in accordance with the reviewer's suggestion

Reviewers' Comments:

Reviewer #1:

Remarks to the Author:

The manuscript "An endocytic capture model for skeletal muscle T-tubule formation" by Hall and collaborators reports a number of elegant EM- and confocal microscopy- based experiments aimed to understand the mechanisms that support the formation of the T tubule system using the zebrafish model system.

An interesting combination of experimental data and mathematical models allows the authors to propose a general mechanism of T tubule formation and stabilization based on a model of endocytic capture where endocytic tubules formed in the initial stages of development are subsequently stabilized by myofibrils /SR and by delivery of membranes from the recycling endosome and Golgi complex. The experiments described are based on an impressive number of different transgenic fish lines and plasmids in order to identify different membrane compartments in muscle cells.

After a first validation of the main markers used to label the sarcolemma and T-tubule membrane, the authors show that, after the T-tubule system starts to develop, it arrives in few days to have a surface area greater than that of the external sarcolemma.

1) Page 6, lines 12-20: ratio between T-tubule and surface area of the sarcolemma.

Are you calculating T-tubule surface as a fraction of total membrane ? or what do you mean by sarcolemma: total membrane ? is this to represent both T-tubule and external membrane, or only the external membrane on the fiber surface. Anyway, if at 10dpf T-tubules eclipse the surface area of sarcolemma (89082:28151), their ratio cannot be 49%; or am I missing something ? You probably may want to rephrase the text since here I find it a bit confusing.

Experiments reported in Figure 4 confirm that the T-tubules follow the organization of the sarcomeres, are aligned with the Z line and that the junctional SR also follows the same pattern. KO of the slow myosin heavy chain gene reduced the length of the sarcomere and hence the distance between T-tubules, while KO of titin completely dysregulated the internal membranes system. Data reported in Figure 5 indicate that the T-tubules are in contact with the extracellular space, as they were accessible to Alexa-647 UTP and HRP, as expected.

2) Figure 5h and video 5: what does dynamic interaction of dextran containing vesicle with forming tubules mean ? It is not really clear what can be deduced from these experiments. These data are cited in the Discussion at page 15, but are not described in the Results session.

Interestingly, results based on the overexpression of a number of trafficking and t-tubule proteins identify a set of proteins associated with the endocytic machinery capable to affect T-tubule development.

3) The role of EHD1 in T-tubule development is not novel and the relevant paper should be mentioned (doi:10.1016/j.ydbio.2014.01.004)

4) Figure 6: the figure present the IT compartment, which is only mentioned in the figure legend: how confident can we be that this is an inter-tubule domain ? How significant are the value reported from a statistical point of view ?

5) Figure 7: Lipid composition of T-tubules is a rather controversial topic. Indeed the only data which suggest a specific difference are those relative to PS, but again - and unfortunately - how significant are the value reported from a statistical point of view ? I find it to be a rather bold statement to say that "the data in Figure 7 demonstrate a specific lipid composition of T-tubules" and even more to make a comparison to the apical vs basolateral domains of the plasma membrane.

6) The statement at the end of page 12 anticipates data which are to be presented in the next paragraph. Thus it should be postponed or changed.

7) what is the correlation between the intensity profile of mKate2ING and the magenta line in figure 7d ?

As a general comments I would encourage the authors to be less assertive in presenting the data and drawing conclusions. To me they are reporting a large amount of work on the molecular characterization of proteins associated to T-tubules and the effect of heterologous expression of a number of proteins that can affect T-tubule's structure. Unfortunately, whether these proteins are physiologically relevant or directly involved in T-tubule formation is not directly proved. In other words one cannot be sure that the observed effects are the result of interfering with other compartments of the cells that are not normally involved in T-tubule development.

Reviewer #2:

Remarks to the Author:

In their revised article the authors went a long way to address the concerns of this and the other two reviewers. In particular additional experiments and analyses plus substantial rewording have substantially improved the scientific stringency of the study, which I now consider suitable for publication in Nature Communications.

Reviewer #3:

Remarks to the Author:

I first reviewed this paper in Jan 2018 and although I don't remember everything from that time I do remember having some concerns over the lack of detail/explanation of some aspects and I had a bit of an obsession with the long formula that you were using to t-tubule membrane area. I was pleased to receive this paper to review again as I impressed by the amount of work and thought that it had potential. Whilst reading your rebuttal I grew confident that the changes you have made would improve the readability and explain away many of my initial concerns. However, having tried to work through the updated version I have found that there are too many typographical errors for me to be able to review this with any confidence. I have also struggled to find some of the changes that you state were made to the manuscript. As such I have decided that it is not worth my time to try to second guess what you meant to write as I was finding myself building layer upon layer of assumptions most of which seemed obvious, but with each my confidence was diminished.

Rather than make a list of the issues that I have found I thought that I'd give you an example of my thought process looking at the long calculation shown in supplementary figure 3. Firstly, it now has color, which is helpful! It would be useful to mention this in the figure legend as my printed copy (I still read better off paper) did not have color, so the text underneath didn't mean anything as it is not mentioned in the figure legend.

Looking at the formula I wondered why you're measuring the diameter of the fibre to calculate the area, when the diagram (supplementary 3d) clearly shows that you can measure the area. I decided that it may be because you want to use the model to calculate surface areas in cells that you are not imaging by EM. So I decided to read the results section to learn more.

Starting at the bottom of page 4 there is a section explaining the methodology used to build the mathematical model, so I'll start there.

You mention a modified 'ferricyanide stain'? The reference on page 4 points to a TEM paper that does indeed mention a 'ferricyanide method' (although it looks to me to be a relatively standard reduced osmium with uranyl acetate en-bloc). However, in the 'transmission electron microscopy' section of the methods you reference a paper looking at lipid droplets using 3view. The method that they used is just the standard 3view method by Tom Deerink et al (<https://ncmir.ucsd.edu/sbem-protocol>). How have you modified either of these stains?

Continuing onto page 5 a video is mentioned and your rebuttal mentions reducing the frame rate

of the videos. I have no videos to review, so can't comment on these. The images in the pdf are very compressed compared to the previous version, but I've looked at the old images and they looked nice. A single image doesn't prove that the T-tubules are oval, but with the video (that I can't see) it'd help (currently the image is so compressed that it makes them look almost circular). So the modelling part begins. Here is where I start having some bigger issues. The mosaics look OK (I do wonder how you managed to section the entire fish precisely across a z-line, but again I may be miss understanding your work flow)

page 5, 3 lines from the bottom "Regression analyses of these data gave predictive functions for myofibril cross sectional area (Fig. 3d-h, Supplementary Fig. 3a-c)"

I don't think you can say that all of those plots are about regression for myofibril cross-sectional area, especially as the next line about splits only mentions Fig. 3e. Perhaps you meant that all the plots are used to generate the model, but this is not what you wrote?

Page 6 has a number of typos:

Page 6 line 1: "Together with measurements of fibre length from confocal microscopy (Fig. 3g)"

Do you mean 3f?

Page 6 line 2: "precise measurements of tubule morphometrics obtained from focused ion beam electron microscopy (Fig 3h)"

Do you mean 3g?

Page 6 line 4: "mathematical model for the surface areas of membranes present in both T-tubule and sarcolemmal compartments for fibres of given sizes (Fig. 3g, Supplementary Fig. 2a-c)."

Do you mean 3h and supplementary 3a-c?

near the middle of page 6: "after 10dpf will eclipse the surface area of the sarcolemma (89082 : 28151 μm^2 ; 49%)"

Comparing with your other calculations I make this 76%, which does indeed 'eclipse' the surface area, but doesn't fit well on your plot (fig 3h) as the 89082 T-tubule would tally with a cross-section of around 1100 μm^2 , but the 28151 sarcolemma is in the area is much closer to the cross over (where your 49% would be more acceptable, but not an 'eclipse').

While looking at the plot I note that the scale bar has changed since the previous version (which I still have for some odd reason). The plot in old figure 2L and new figure 3h look similar, but the numbers are different. Looking at the just the Sarcolemma plot. The old graph went up to 700 μm^2 fibre cross-section. The line at this point appears to reach approx. 25000 μm^2 Surface area.

However you new plot covers a larger range (as you explain in your rebuttal), but if I trace a line around 700 μm^2 I find that it is closer to 35000 μm^2 . The cross-over also seems to have moved from approx. 500 to 600 μm^2 fibre cross-sectional area.

So, I'm now wondering how the numbers have changed when the model looks to be identical to last time (the adult data mentioned in your rebuttal isn't shown that I can see). I have a number of other questions about the model, including why the simplified version (supp figure 3 f) still has measured diameters of the T-tubule (when you have a plot showing the mean d_1 d_2), yet you are using a circle and a constant to calculate the area of the fibre? I was hoping that by reading through your method all this would be explained, however the incorrect figure legends and wrong calculations just confused the matter.

I hope you can see that this is just one issue that I'm having when I start with the question "how does the formula work?". In trying to learn more I have uncovered a number of typographical errors where the correct answer sometimes appears obvious (eg figure legends), but each time I see an error I lose confidence. I can't tell if I'm following the logic of your work, or making up my own story based on miss-reading the figures. As such I cannot honestly review a paper if I do not trust my understanding of what you are trying to show. This seems a shame as the changes mentioned in your rebuttal sound promising.

I have seen other issues of varying severity whilst looking at the paper (including a few changes that you mention in your rebuttals that are not obvious in the paper eg beam settings etc). However, I do not want to spend my time proof reading.

I really hope that somehow I have been sent the wrong version of the file to review as otherwise it would appear that the person writing the rebuttal has not read the uploaded paper.

Reviewer #4:

Remarks to the Author:

For most part authors have answered comments from all reviewers and incorporated many of their suggestions. Thus, I think that the manuscript is now ready for publication at Nature Communications.

Reviewer #1 (Remarks to the Author):

The manuscript "An endocytic capture model for skeletal muscle T-tubule formation" by Hall and collaborators reports a number of elegant EM- and confocal microscopy- based experiments aimed to understand the mechanisms that support the formation of the T tubule system using the zebrafish model system.

An interesting combination of experimental data and mathematical models allows the authors to propose a general mechanism of T tubule formation and stabilization based on a model of endocytic capture where endocytic tubules formed in the initial stages of development are subsequently stabilized by myofibrils /SR and by delivery of membranes from the recycling endosome and Golgi complex. The experiments described are based on an impressive number of different transgenic fish lines and plasmids in order to identify different membrane compartments in muscle cells.

After a first validation of the main markers used to label the sarcolemma and T-tubule membrane, the authors show that, after the T-tubule system starts to develop, it arrives in few days to have a surface area greater than that of the external sarcolemma.

1) Page 6, lines 12-20: ratio between T-tubule and surface area of the sarcolemma.

Are you calculating T-tubule surface as a fraction of total membrane? or what do you mean by sarcolemma: total membrane ? is this to represent both T-tubule and external membrane, or only the external membrane on the fiber surface. Anyway, if at 10dpf T-tubules eclipse the surface area of sarcolemma (89082:28151), their ratio cannot be 49%; or am I missing something ? You probably may want to rephrase the text since here I find it a bit confusing.

Response:

This point has been clarified and changed according to the reviewers suggestion. The reviewer is also correct in noticing that the numbers for 10dpf fish were incorrectly pasted from the spreadsheet. These have now been corrected. This portion of the manuscript (p6) now reads:

"With this model we calculated that the T-tubule system (undetectable at 16h), has developed into a complex organelle with a surface area comprising 35% of the total surface area of the cell ($2579 : 4817 \mu\text{m}^2$; T-system : sarcolemma) by 48hpf (total surface area = T-system surface area + sarcolemma). By 5dpf the relative surface area of the T-tubule domain gradually increases to 40% ($4737 : 7092 \mu\text{m}^2$) and after 10dpf will eclipse the surface area of the sarcolemma ($13695 : 14456 \mu\text{m}^2$; 49%)."

Experiments reported in Figure 4 confirm that the T-tubules follow the organization of the sarcomeres, are aligned with the Z line and that the junctional SR also follows the same pattern. KO of the slow myosin heavy chain gene reduced the length of the sarcomere and hence the distance between T-tubules, while KO of titin completely dysregulated the internal membranes system. Data reported in Figure 5 indicate that the T-tubules are in contact with the extracellular space, as they were accessible to Alexa-647 UTP and HRP, as expected.

2) Figure 5h and video 5: what does dynamic interaction of dextran containing vesicle with forming tubules mean ? It is not really clear what can be deduced from these experiments. These data are cited in the Discussion at page 15, but are not described in the Results session.

Response:

Apologies, the reference to the figure was omitted in the revision leaving only the reference to the video. These data primarily demonstrate that there is a physical separation between the surface connected T-tubule domain and the endosomes. While we cannot currently quantify "dynamic interactions" robustly, or conclusively show fission/fusion events, these data also provide a tantalising glimpse of the possibilities for future study. We have added the reference to the figure to the results section and the passage now reads:

"Injection of the tracer into 16, 24 or 48 hpf GFP-CaaX zebrafish resulted in immediate infiltration into the tubules, confirming surface connectivity (Figs. 5d-f). Similarly, uptake of the fluid-phase marker Alexa-647-dextran ($10,000\text{MW}$), also resulted in immediate infiltration into the tubules, whereas uptake into putative endosomes only occurred after several minutes (Fig. 5g, Supplementary Video 5)."

Interestingly, results based on the overexpression of a number of trafficking and t-tubule proteins identify a set of proteins associated with the endocytic machinery capable to affect T-tubule development.

3) The role of EHD1 in T-tubule development is not novel and the relevant paper should be mentioned (doi:10.1016/j.ydbio.2014.01.004)

Response: Thank you for the suggestion. This has now been added (reference 31, shown in the text below): "Firstly, tubules derived from the sarcolemma nucleate on the inner leaflet of the plasma membrane. These early tubules possess markers of clathrin independent endocytosis, in particular CD44, EHD1³¹ and dynamin2^{32,33}"

4) Figure 6: the figure present the IT compartment, which is only mentioned in the figure legend: how confident can we be that this is an inter-tubule domain?

Response: We are 100% confident that the position in the centre of the sarcomere represents the inter-tubule domain. Our light and electron microscope observations and the published literature support the view that transverse tubules don't occur in the centre of the sarcomere.

How significant are the values reported from a statistical point of view?

Response: This is a nice idea. We had already included statistical analysis of our Rab screen perturbation data and we have now extended these analyses to the localisation data (see Figure 6j, 6m, 8i, and Supplementary Fig. 8i).

5) Figure 7: Lipid composition of T-tubules is a rather controversial topic. Indeed the only data which suggest a specific difference are those relative to PS, but again - and unfortunately - how significant are the values reported from a statistical point of view?

Response: As described above, we have calculated the significance levels and indicated them on Fig. 7h with the appropriate asterisks.

I find it to be a rather bold statement to say that “the data in Figure 7 demonstrate a specific lipid composition of T-tubules” and even more to make a comparison to the apical vs basolateral domains of the plasma membrane.

Response: We agree, this is too bold a statement. We have weakened this statement by using “suggest” rather than “demonstrate” and have removed the comparison to apical and basolateral domains of the plasma membrane. The sentence now reads:

“These data suggest that a defining feature of the early tubules is a distinct lipid composition, which may form a platform on which other proteins are recruited to establish the distinct protein composition.”

6) The statement at the end of page 12 anticipates data which are to be presented in the next paragraph. Thus it should be postponed or changed.

Response: We agree, this sentence has been moved to the concluding sentence of the results section.

7) what is the correlation between the intensity profile of mKate2/ING and the magenta line in figure 7d?

Response: This particular image was perhaps a poor choice to include in the figure. The reason for this is that in this image the T-tubules are not perfectly perpendicular to the horizontal axis of the image. This does not, of course, affect the quantitation or statistical analyses since the macros use multiple regions of interest on the different domains themselves rather than a simple average through the x and y of the entire image. The image in Fig. 7d has now been replaced with another example, which allows the peaks to be seen. What we did not appreciate until the reviewer asked this question (and requested the statistical analysis) is that there is a small but significant enrichment of the ING probe in the region of the triad. This is interesting and consistent with some observations from the literature showing that muscle specific Bin1 is able to bind PI5P (e.g. Fugier et al. Nat Med 2010), which in theory can be produced by myotubularin from endosomal PI(3,5)P2. However, as the reviewer points out, this is a somewhat controversial topic, the discussion of which is too speculative for the current manuscript.

As a general comments I would encourage the authors to be less assertive in presenting the data and drawing conclusions. To me they are reporting a large amount of work on the molecular characterization of proteins associated to T-tubules and the effect of heterologous expression of a number of proteins that can affect T-tubule's structure. Unfortunately, whether these proteins are physiologically relevant or directly involved in T-tubule formation is not directly proved. In other words one cannot be sure that the observed effects are the result of interfering with other compartments of the cells that are not normally involved in T-tubule development.

We appreciate the reviewer's comment and have tempered our conclusions throughout. Most importantly, we have changed the following sentence in the Abstract:

“We present a new endocytic capture model...”

to read:

“We propose a new endocytic capture model...”

and the following sentence in the Discussion:

“... we propose a three-step framework for skeletal muscle T-tubule formation.”

to read:

“...we put forward a testable model for skeletal muscle T-tubule formation comprising of three stages.”

Most significantly, note that we have also changed the title of the manuscript from first submission, from:

“Muscle T-tubules Form by Endocytic Capture”

to:

“An Endocytic Capture Model for Skeletal Muscle T-tubule Formation”

Reviewer #3 (Remarks to the Author):

I first reviewed this paper in Jan 2018 and although I don't remember everything from that time I do remember having some concerns over the lack of detail/explanation of some aspects and I had a bit of an obsession with the long formula that you were using to t-tubule membrane area. I was pleased to receive this paper to review again as I impressed by the amount of work and thought that it had potential. Whilst reading your rebuttal I grew confident that the changes you have made would improve the readability and explain away many of my initial concerns. However, having tried to work through the updated version I have found that there are too many typographical errors for me to be able to review this with any confidence. I have also struggled to find some of the changes that you state were made to the manuscript. As such I have decided that it is not worth my time to try to second guess what you meant to write as I was finding myself building layer upon layer of assumptions most of which seemed obvious, but with each my confidence was diminished.

Response: We appreciate that this section was difficult for the reviewer to follow. In part, this was due to space constraints within the text of the manuscript. In order to address to this we have added a new and extensive section to the Methods, which describes in detail, each step in the development of the model (please see the paragraph entitled “Mathematical model”, p33). We have also verified that in every case the text corresponds to the appropriate figure.

Rather than make a list of the issues that I have found I thought that I'd give you an example of my thought process looking at the long calculation shown in supplementary figure 3. Firstly, it now has color, which is helpful! It would be useful to mention this in the figure legend as my printed copy (I still read better off paper) did not have color, so the text underneath didn't mean anything as it is not mentioned in the figure legend.

Response: We have added the additional text requested to the figure legend which now reads:

“(e) Equation for T-tubule surface area based on the input parameters shown in d. Sections of the equation are colour coded to show source. (f) Simplified version of the formula shown in e, with the same input parameters. (g) Simplified version of the equations shown in e and f, with the constant parameters $d1$, $d2$ and s amalgamated, leaving only L and D . A third version of the equation where diameter (D) is replaced by cross sectional area is also given in the Methods.”

Looking at the formula I wondered why you're measuring the diameter of the fibre to calculate the area, when the diagram (supplementary 3d) clearly shows that you can measure the area. I decided that it may be because you want to use the model to calculate surface areas in cells that you are not imaging by EM. So I decided to read the results section to learn more.

Response: The reviewer is correct. Most muscle cellularity measurements are made on paraffin or frozen sections and fibre size measurements are often only reported as diameters. To elaborate further, the literature shows that different groups have different preferences for using fibre diameter or cross-sectional area, some believing that cross-sectional area is more accurate as it is a direct measurement, others believing that the minimum Feret diameter is the most accurate, arguing that some fibres might be cut somewhat obliquely and consequently give artificially large cross-sectional areas. Our rationale in choosing fibre diameter as the input parameter rather than cross-sectional area was to provide the simplest possible model. However, in response to the reviewer's very valid suggestion, and to accommodate for the current dichotomy in established methodology, we have now also provided the equivalent derivation of our equation which uses cross-sectional area as an input parameter rather than fibre diameter. This can be found in the new Methods paragraph entitled “Mathematical model” on p35.

Starting at the bottom of page 4 there is a section explaining the methodology used to build the mathematical model, so I'll start there.

You mention a modified 'ferricyanide stain'? The reference on page 4 points to a TEM paper that does indeed mention a 'ferricyanide method' (although it looks to me to be a relatively standard reduced osmium with uranyl acetate en-bloc). However, in the 'transmission electron microscopy' section of the methods you reference a paper looking at lipid droplets using 3view. The method that they used is just the standard 3view method by Tom Deerink et al (<https://ncmir.ucsd.edu/sbem-protocol>). How have you modified either of these stains?

Response: This point has been clarified and we understand why the reviewer was confused. On page 4 we refer to Franzin-Armstrong (1991), because it beautifully illustrates the specificity of a ferricyanide stain for the T-tubules in muscle cells. We have deleted the phrase "modified version of a" from this sentence (p4-5) which now reads:

"We used a potassium ferricyanide stain on thin transmission electron microscopy (TEM) sections to specifically label the T-tubules⁸ and distinguish them from sarcoplasmic reticulum (SR)."

The reference in the methods (Herms et al 2013) is the protocol we used to do the staining. The specific modifications that we used (microwave fixation and cut the heads and tails off the zebrafish) are now clearly described in the Methods section (p30) which reads:

"Zebrafish were anesthetized with tricaine (Sigma) in E3 and fixed with 2.5% glutaraldehyde in a Pelco BioWave microwave at 80W for 6 min under vacuum in a 2min on, 2min off, 2min on, sequence. The head and tail-tips of the zebrafish embryos were removed and the remaining trunk was re-fixed in 2.5% glutaraldehyde in the microwave using the same sequence. Zebrafish were then processed as described previously⁴⁷. Briefly, samples were washed in PBS and postfixed in 2% OsO₄ with 1.5% potassium ferricyanide. Samples were washed in water and incubated in a 1% (w/v) thiocarbonylhydrazide solution for 20min. Tissue was postfixed again in 2% OsO₄, washed in water and stained with 1% uranyl acetate. Tissues was further stained with a lead aspartate solution (20mM lead nitrate, 30mM aspartic acid, pH 5.5), serially dehydrated in acetone, infiltrated with resin and polymerized. Ultrathin sections were cut on a Leica UC6 ultramicrotome and imaged at 80kV on a JEOL-1011 transmission electron microscope fitted with an Morada 2k x 2k CCD camera under the control of Olympus software."

Continuing onto page 5 a video is mentioned and your rebuttal mentions reducing the frame rate of the videos. I have no videos to review, so can't comment on these. The images in the pdf are very compressed compared to the previous version, but I've looked at the old images and they looked nice. A single image doesn't prove that the T-tubules are oval, but with the video (that I can't see) it'd help (currently the image is so compressed that it makes them look almost circular).

Response: The FIB images in the original manuscript were a different series taken at a very early stage, and were taken to look at the earliest steps of T-tubule formation rather than for direct measurements of the tubules at the triads. The objective here was high magnification and resolution. These data were removed from the manuscript for the second submission in response to the reviewer's initial comments. For this second set of images, we had to trade off magnification and resolution in order to give us an adequate number of triads to measure to be able to calculate variance while still allowing precision of measurement. It is true that "A single image doesn't prove that the T-tubules are oval" and we draw the reviewer's attention to the plot in Fig. 3g where we provide the quantitative information derived from this set of images.

So the modelling part begins. Here is where I start having some bigger issues. The mosaics look OK (I do wonder how you managed to section the entire fish precisely across a z-line, but again I may be missing understanding your work flow)

Response: We were indeed unable to section an entire fish precisely through the Z-line. However, our SBFEM data shows that T-tubules track through the splits in the myofibrils at the level of the Z-lines. For this reason, we used measurement of the total linear distance of myofibril splits in a transverse section of a fibre as a proxy for the total linear distance of the Z-line associated T-tubule. We have specifically addressed this point in our new "Mathematical model" section of the Methods (p33-34) as follows:

"The T-tubules track through the splits or "furrows" in the myofibrils, such that at the level of each and every Z-line within a fibre, the total combined length of the myofibrillar furrows in transverse section (illustrated by the red lines in Fig. 2k) will be equal to the total combined length of the associated T-tubules. Moreover, total linear furrow distance can be predicted from the cross sectional area of the myofibrils within a fibre by the equation $y = 1.26x + 6.06$ (Fig. 3e). Combining equations for myofibril cross sectional area and total linear furrow distance allows calculation of the length of T-tubule associated with each single Z-line within a muscle fibre, using the diameter area of the fibre (D):"

$$1.26 \left[0.95\pi \left(\frac{D}{2} \right)^2 - 13.55 \right] + 6.06$$

page 5, 3 lines from the bottom "Regression analyses of these data gave predictive functions for myofibril cross sectional area (Fig. 3d-h, Supplementary Fig. 3a-c)"

I don't think you can say that all of those plots are about regression for myofibril cross-sectional area, especially as the next line about splits only mentions Fig. 3e. Perhaps you meant that all the plots are used to generate the model, but this is not what you wrote?

Response: The reviewer is correct. "3d-h" should read 3d. This has been corrected as below, in the new "Mathematical model" section (p33) of the Methods:

"Myofibril cross-sectional area is a function of fibre cross sectional area and can be calculated from the linear regression equation shown in Fig 3d ($y = 0.95x - 13.55$), where x is muscle fibre cross sectional area. Combining equations for fibre cross sectional area and myofibril cross sectional area allows calculation of myofibril cross sectional area from fibre diameter (D)."

Page 6 has a number of typos:

Page 6 line 1: "Together with measurements of fibre length from confocal microscopy (Fig. 3g)"

Do you mean 3f?

Page 6 line 2: "precise measurements of tubule morphometrics obtained from focused ion beam electron microscopy (Fig 3h)"

Do you mean 3g?

Page 6 line 4: "mathematical model for the surface areas of membranes present in both T-tubule and sarcolemmal compartments for fibres of given sizes (Fig. 3g, Supplementary Fig. 2a-c)."

Do you mean 3h and supplementary 3a-c?

Response: All figure labels now correspond correctly to the text.

near the middle of page 6: "after 10dpf will eclipse the surface area of the sarcolemma (89082 : 28151 μm^2 ; 49%)"

Comparing with your other calculations I make this 76%., which does indeed 'eclipse' the surface area, but doesn't fit well on your plot (fig 3h) as the 89082 T-tubule would tally with a cross-section of around 1100 μm^2 , but the 28151 sarcolemma is in the area is much closer to the cross over (where your 49% would be more acceptable, but not an 'eclipse').

The reviewer is correct. As reviewer 1 also pointed out, one of the numbers for 10dpf fish was incorrectly pasted from the spreadsheet. These have now been corrected. This portion of the manuscript (p6) now reads:

"With this model we calculated that the T-tubule system (undetectable at 16h), has developed into a complex organelle with a surface area comprising 35% of the total surface area of the cell (2579 : 4817 μm^2 ; T-system : sarcolemma) by 48hpf (total surface area = T-system surface area + sarcolemma). By 5dpf the relative surface area of the T-tubule domain gradually increases to 40% (4737 : 7092 μm^2) and after 10dpf will eclipse the surface area of the sarcolemma (13695 : 14456 μm^2 ; 49%)."

While looking at the plot I note that the scale bar has changed since the previous version (which I still have for some odd reason). The plot in old figure 2L and new figure 3h look similar, but the numbers are different. Looking at the just the Sarcolemma plot. The old graph went up to 700 μm^2 fibre cross-section. The line at this point appears to reach approx. 25000 μm^2 Surface area. However you new plot covers a larger range (as you explain in your rebuttal), but if I trace a line around 700 μm^2 I find that it is closer to 35000 μm^2 . The cross-over also seems to have moved from approx. 500 to 600 μm^2 fibre cross-sectional area.

So, I'm now wondering how the numbers have changed when the model looks to be identical to last time (the adult data mentioned in your rebuttal isn't shown that I can see).

Response: The reviewer refers to the current Fig 3h & I, where we use the model to calculate the surface area of the T-tubules and the surface area of the sarcolemma (not including the T-tubules) for fibres of given sizes. In fact, these numbers changed in the first revision as a direct response to the reviewer's original critique. Our original graph was based upon the stated assumption that fibres scale allometrically, i.e. they continue to grow by the same proportions in diameter as they do in length. The reviewer questioned whether this was, in fact, the case and we went on to investigate further. What we found was that while fibre length is relatively predictable and essentially equal to somite width (now shown in Fig. 3f), at any one time there will be a mixed distribution of fibre cross sectional areas since fibres hypertrophy at different rates and are added to the myotome at different times during growth and development (now shown in Fig. S3a-c). So in fact, cross-sectional area/diameter is only part of the story, as fibre length is changing at the same time. With this in mind, in our first revision, we took the stages for which we had accurate data on fibre cross sectional area/diameter derived from our EM measurements (48hpf, 5dpf and 10dpf) and made the calculations described in the results section for 48hpf, 5dpf and 10dpf fish. Full details of these calculations are now given in another new paragraph in the Methods section entitled "Application of the mathematical model to 48hpf, 5dpf 10dpf and adult zebrafish" on p36 and are also pasted below. The final graph (Fig. 3h in revision 1), thanks to the reviewer's suggestion, was not based upon

allometric scaling, but a linear scaling of fibre length and diameter between the largest fibres present at 48hpf, and the largest fibres we were able to find from a one year old adult from the zebrafish online atlas (<http://bio-atlas.psu.edu/>). However, in the current revision, thanks to the reviewers latest comments, we have realised there is a better way of illustrating the final model. We have further revised this graph into two separate panels (current Fig 3h & i). In these panels we now plot sarcolemma and T-tubule surface area on 3 axes (the independent variables being fibre length and fibre diameter). We believe that this is the most intuitive way of visualising how the surface area of the two compartments change as a fibre grows.

Application of the mathematical model to 48hpf, 5dpf 10dpf and adult zebrafish.

In order to calculate the surface areas of the T-tubule and sarcolemma at 48hpf, 5dpf and 10dpf, values for fibre length (L) were 90.23µm, 107.77µm and 158.83µm respectively, and were calculated using the regression equation for somite width ($y = 0.2x^3 - 2.6x^2 + 17x + 65$; Fig 3f) for values of $x = 2$, $x = 5$ and $x = 10$. Values for maximum fibre diameter (D) were 15.64µm, 19.23µm and 26.72µm respectively (directly measured cross-sectional areas of 192.04µm², 290.42µm² and 560.86µm²) taken from the empirically derived data shown in Fig 3a-c. Note that these calculations were based on maximum fibre size (rather than the mean size or animal age) since new fibres are continuously added to the myotome and existing fibres grow at different rates (see cellularity profile histograms in Supplementary Fig. 3a-c). To apply this model to the largest fibres found in adult zebrafish we took approximate measurements from the zebrafish atlas images at (<http://bio-atlas.psu.edu/zf/view.php?s=220&atlas=18>, <http://bio-atlas.psu.edu/zf/view.php?s=275&atlas=17>). The largest fibres we found in this data set were approximately 580µm in length and 40µm in diameter. Applying values of L = 580µm and D = 40µm, gave values of 113083 : 75398µm²; (T-system : sarcolemma) which corresponds to approximately 1.5x. The graphs in Fig. 3h & i show the consequences of scaling fibre length and diameter up to the maximal values measured from 1 year old fish of L = 580µm and D = 40µm (cross sectional area = 1256.64µm²).

I have a number of other questions about the model, including why the simplified version (supp figure 3 f) still has measured diameters of the T-tubule (when you have a plot showing the mean d1 d2), yet you are using a circle and a constant to calculate the area of the fibre? I was hoping that by reading through your method all this would be explained, however the incorrect figure legends and wrong calculations just confused the matter.

Response: We have "simplified out" the values for d1 and d2, according to the reviewer's preference. The new "simpler" equation is shown in Fig S3g.

I hope you can see that this is just one issue that I'm having when I start with the question "how does the formula work?". In trying to learn more I have uncovered a number of typographical errors where the correct answer sometimes appears obvious (eg figure legends), but each time I see an error I lose confidence. I can't tell if I'm following the logic of your work, or making up my own story based on miss-reading the figures. As such I cannot honestly review a paper if I do not trust my understanding of what you are trying to show. This seems a shame as the changes mentioned in your rebuttal sound promising.

I have seen other issues of varying severity whilst looking at the paper (including a few changes that you mention in your rebuttals that are not obvious in the paper eg beam settings etc). However, I do not want to spend my time proof reading.

I really hope that somehow I have been sent the wrong version of the file to review as otherwise it would appear that the person writing the rebuttal has not read the uploaded paper.

Toby Starborg

We note that Referee #3 provided further comments to us – for other points that should be addressed. These are the following:

The SBFSEM and FIBSEM methods could do with information on imaging conditions, pixel size, voltage, slice thickness vacuum etc.

Response: We have added the following imaging conditions to the Methods section (p31) for the 3View microscopy:

"Imaging was performed on a Zeiss Gemini FE-SEM fitted with a Gatan 3view, at 2.25kV, with magnification 7.5k and a pixel size of 5.7nm. Images were captured at a resolution of 5000x5000 (dwell 2), under low vacuum (8Pa) with 50nm sections."

We have added the following imaging conditions to the Methods section (p31) for the Focussed ion beam microscopy:

“Samples were imaged on an FEI Scios FIB Dual Beam SEM at 1.5kV with a 0.8nA electron beam at a resolution of 3072x2048, with a chamber vacuum of $1 - 2 \times 10^{-4}$ Pa. Slicing was carried out at 30kV with a 1nA Ga⁺ ion beam. Voxel size and slice thickness was 8nm.”

Ref #3: In their methods I can only see that they removed a temperature 50°C and added the missing osmium stains. There is no mention of the information I asked for.

Response: We have now written out the TEM method in full (p30) explaining what we modified (we used microwave fixation and cut the heads and tails off the fish). We have also added in pixel/voxel sizes for the serial blockface methods, as well as resolution, vacuum settings, kV and slice thicknesses. We hope that this information is sufficient.

Figure 3 (and supplementary figure). I presume that the images are from mosaic fish as only the central cell shows over expression. This is not stated in the figure legend.

Response: We presume the reviewer is referring to Figure 8, which was Figure 3 in the old manuscript (the current Fig. 3 and S3 contain no images of cells). We draw the reviewer's attention to the graphical legend on the bottom right of Figure 8, which reads "mosaic expression from DNA injection". Where appropriate all figures have been given graphical legends of this type.

We have also thoroughly checked all figure legends for accuracy.

Ref #3: Figure 3 is now Figure 8, but I can't see any mention of mosaic fish, unless I'm miss-reading the legends.

Response: We believe this point refers to the same figure as above, and has been addressed as detailed.

Other points: supplementary figure 8 has asterisks – but it is not defined what they mean/highlight – please clarify.

Response: We have now detailed levels of significance in this figure legend.

Supplementary figure 5 parts b and c are not mentioned in the figure legend

Response: We have added the following text (p23) to the figure legend:

(b, c) Injection of unconjugated horse radish peroxidase into the circulation at 48hpf results in infiltration into the T-tubules after 45 min.

Reviewer #4 (Remarks to the Author):

For most part authors have answered comments from all reviewers and incorporated many of their suggestions. Thus, I think that the manuscript is now ready for publication at Nature Communications.

Reviewers' Comments:

Reviewer #1:

None

Reviewer #3:

Remarks to the Author:

Firstly, I'd like to thank you for giving me something to tax my brain during lockdown. It has been nice to have something out of my comfort zone to work-through/understand. In particular I appreciate the time that you have taken to explain how you came up with the mathematical formula which you have used to model the cellular increase in membrane. I now find this whole section much clearer and can follow most of it.

As a non-muscle, non-zebrafish person who was asked to look at the electron microscopy I feel that the changes that you have made have made the paper easier to read overall. I still have a few minor questions which I hope that you will be able clarify in order to make the paper even easier to follow.

Key questions:

It is not clear what the distinction is between T-tubules and furrows in the EM analysis. I bring up this point as page 6, line 5 has a statement (in brackets) that the T-tubule system is "undetectable at 16h". The example analysis image in figure 2k shows 3 'furo' lines at 16h. This would suggest that you are not confident that these lines are measuring the T-tubule length, yet the "furo" lengths from the other time points are used in your model of t-tubule development (fig 3e and top of page 34). Perhaps modify the line to suggest that detection was difficult at 16h?

Supplementary figure 5: How does the contrast following HRP injection differ from the contrast shown in the other TEM images (eg figure 2F)? The method for HRP injection (page 26) is missing the DAB step. It is the DAB, precipitated by the HRP, that is osmophilic and so stains during sample preparation giving localised density.

----- The mathematical model -----

I would like to reiterate how much I appreciate the time that you have taken to break down the steps in forming the mathematical model of membrane surface area. I have found this far easier to follow and only have a few minor quibbles with the resultant text.

Firstly, feel free to move the text into the supplemental data if you are pushed for word count. It is very useful to keep, but your conclusions are well represented by the new plots in figure 3h+i.

My concerns now are only over the use of the term 'simplify'. I enjoy mathematics, but don't have a strong background there. I do like to follow how the formula was derived, but at a few points I had to trust you as the numbers that appeared didn't seem to make any sense.

As I write this I have a feeling that this is due to my own lack of understanding and so I'm happy for you to leave these as they are for others to judge.

In case you're interested the areas I got stuck are:

Page 18 line 6: I have no idea where the 3 came from at the front of the formula. I can see where the 14684 and 800000 are from, but don't have the time to work out how you came up with those exact numbers.

The formula on line 12 makes more sense:

- I can see that the 157.9399995 is the derived mean diameter of the tubule based on the d_1, d_2 figures.
- I can see that the 11.013 comes from the $1.26 \times 13.55 + 6.06$.
- The 3800 appears to be $2 \times 1.9 \times 1000$ which is sarcomere length with the conversion of the tubule diameter to microns. I'm not sure why this is multiplied by 2 as we're looking at the area here, but I'm willing to accept that I've probably missed a step.

As it is straightforward to convert from a diameter to a radius I do wonder whether the formula would look nicer (ie easier to read) if you use the radius instead of diameter at each point (ie r_1 and r_2 for the tubules and R for the cell instead of $d_1/2$, $D/2$).

Page 35 line 18. You use $2\pi D/2$ to find the circumference of a circle. Why not just πD ?

The simplified formulas on page 37 are beyond me. I don't understand why you've left π in one of them, but not the other.

I have already stated my concern with using a simplified cylinder to measure the sarcoplasmic area. However, this does not affect the overall conclusion: that a lot of membrane needs to be produced. In fact, a rippling sarcolemma would actually increase the amount of membrane needed as the cell grows.

A few minor points:

Figure 1 a and b both show caax. The rest of the manuscript uses CaaX. My undergraduate biochemistry swears that this should be CAAX for L-isomer amino acids. However, I know that genes names tend to follow different patterns, so long as it is consistent I don't mind.

Pg 4 line 25. I still don't agree that potassium ferricyanide is a 'specific' stain, however you have a reference that states that this is the case, so I can't argue. For the record my concern is two parts: Firstly, I don't feel that the stain is 'specific' to T-tubules, only that the contrast is increased somewhat. Secondly, I don't think of potassium ferricyanide as a stain on its own. I have always thought of $K_3[Fe(CN)_6]$ as a reducing agent altering osmium from 8+ to one of the lower oxidation states and thus altering the binding potential within the components in the sample. However, I will accept that I am not a chemist and that both Iron (atomic number 26) and K (19) will give some electron density above the resin.

Page 8 line 6: Did you quantify the diameters based on transverse TEM sections? The spread of diameters for WT (4k) suggests that the orientation is random. I presume that you measured the short axis of oblique cut tubes. This is not clear in the text (if only to differentiate from the diameters measured using FIBSEM data earlier in the paper).

Figure 5 h has arrows on it that aren't mentioned in the text, or figure legend.

Figure 5a-c: what is the contrast agent/stain.

Figure 5a-c and g: has the contrast been inverted?

It may be worth adding a note to the figure legends when you have altered the image contrast for printing. For example, figure 6h shows mKate2-ARF6. This is stated to have a very low TT:IT ratio, yet you can clearly see lines in the image. I pasted the image into Fiji and was able to get a clear line plot of these peaks if I only analysed the top half of the image. I did notice that analysing the entire image removed the peaks due to the displacement of the lower set of lines (tubules). This displacement is also seen in the EGFP plot above the merged image as double peaks. Contrast enhancement is mentioned in the methods, but not everybody will read the methods. The issue is particularly noticeable for figures 6 -8 as the TT:IT ratios don't always match what the image appears to show.

Another example of this is 6e, where the central cell appears to have a highly reduced CaaX stain compared to the neighbouring cells, yet the TT:IT ratio is shown as being high.

There are more examples, but I presume that these are due to print contrast as the analysis use the raw data.

Page 6 line 9. I was going to write that 49% does not "eclipse" 51%, but as I write this I wonder if

you meant that after this point the T-tubule surface area will overtake sarcolemma?

Page 31 line 7. The reference that you wanted (Mastronarde) is:

Mastronarde, D. N. Automated electron microscope tomography using robust prediction of specimen movements. *J. Struct. Biol.* 152, 36–51 (2005).

REVIEWERS' COMMENTS:

Reviewer #3 (Remarks to the Author):

Firstly, I'd like to thank you for giving me something to tax my brain during lockdown. It has been nice to have something out of my comfort zone to work-through/understand. In particular I appreciate the time that you have taken to explain how you came up with the mathematical formula which you have used to model the cellular increase in membrane. I now find this whole section much clearer and can follow most of it. As a non-muscle, non-zebrafish person who was asked to look at the electron microscopy I feel that the changes that you have made have made the paper easier to read overall. I still have a few minor questions which I hope that you will be able clarify in order to make the paper even easier to follow.

Key questions:

It is not clear what the distinction is between T-tubules and furrows in the EM analysis. I bring up this point as page 6, line 5 has a statement (in brackets) that the T-tubule system is "undetectable at 16h". The example analysis image in figure 2k shows 3 'furrow' lines at 16h. This would suggest that you are not confident that these lines are measuring the T-tubule length, yet the "furrow" lengths from the other time points are used in your model of t-tubule development (fig 3e and top of page 34). Perhaps modify the line to suggest that detection was difficult at 16h?

This line has been modified to read:

"With this model we calculated that the T-tubule system (only identifiable after 16hpf), has developed into a complex organelle with a surface area comprising 35% of the total surface area of the cell (2579 : 4817 μm^2 ; T-system : sarcolemma) by 48hpf (total surface area = T-system surface area + sarcolemma)."

Supplementary figure 5: How does the contrast following HRP injection differ from the contrast shown in the other TEM images (eg figure 2F)? The method for HRP injection (page 26) is missing the DAB step. It is the DAB, precipitated by the HRP, that is osmophilic and so stains during sample preparation giving localised density.

----- The mathematical model -----

I would like to reiterate how much I appreciate the time that you have taken to break down the steps in forming the mathematical model of membrane surface area. I have found this far easier to follow and only have a few minor quibbles with the resultant text.

Firstly, feel free to move the text into the supplemental data if you are pushed for word count. It is very useful to keep, but your conclusions are well represented by the new plots in figure 3h-i.

Response:

Thank you for your detailed critique. The explanation is currently in the Methods section, which has no word limit.

My concerns now are only over the use of the term 'simplify'. I enjoy mathematics, but don't have a strong background there. I do like to follow how the formula was derived, but at a few points I had to trust you as the numbers that appeared didn't seem to make any sense.

As I write this I have a feeling that this is due to my own lack of understanding and so I'm happy for you to leave these as they are for others to judge.

In case you're interested the areas I got stuck are:

Page 18 line 6: I have no idea where the 3 came from at the front of the formula. I can see where the 14684 and 800000 are from, but don't have the time to work out how you came up with those exact numbers.

Response:

We have now added all intermediate steps to the Methods section.

The formula on line 12 makes more sense:

- I can see that the 157.9399995 is the derived mean diameter of the tubule based on the d_1, d_2 figures.
- I can see that the 11.013 comes from the $1.26 \times 13.55 + 6.06$.
- The 3800 appears to be $2 \times 1.9 \times 1000$ which is sarcomere length with the conversion of the tubule diameter to microns. I'm not sure why this is multiplied by 2 as we're looking at the area here, but I'm willing to accept that I've probably missed a step.

Response:

We have now added all intermediate steps to the Methods section.

As it is straightforward to convert from a diameter to a radius I do wonder whether the formula would look nicer (ie easier to read) if you use the radius instead of diameter at each point (ie r_1 and r_2 for the tubules and R for the cell instead of $d_1/2, D/2$).

Response:

We did consider this approach. However as diameters are the directly measured quantity, and muscle fibre diameters in particular are the conventional unit reported in the literature we decided that the formula would be most appealing to potential users with diameters as input parameters.

Page 35 line 18. You use $2\pi D/2$ to find the circumference of a circle. Why not just πD ?

Response:

This is to provide clarity for the reader. We believe it is important to produce the full equation in its entirety, followed by a final simplification.

The simplified formulas on page 37 are beyond me. I don't understand why you've left π in one of them, but not the other.

Response:

We have included the full simplifications in the materials and methods. The reason π is left in one and not the other is simply a matter of in one case being able to produce a much shorter, simpler formula. Multiplying out π when solving for D enabled us to use the standard form quadratic formula in this simplification (see equations 65-67).

I have already stated my concern with using a simplified cylinder to measure the sarcoplasmic area. However, this does not affect the overall conclusion: that a lot of membrane needs to be produced. In fact, a rippling sarcolemma would actually increase the amount of membrane needed as the cell grows.

Response:

We agree that a rippling sarcolemma would increase the amount of membrane needed as the cell grows. However, the model we have produced is exactly that – a (relatively!) simple model which allows us to make some predictions about membrane area during muscle fibre growth. Since we provide all the derivations in their entirety, the potential is there for further refinement, as and when more detailed empirical observations are made.

A few minor points:

Figure 1 a and b both show caax. The rest of the manuscript uses CaaX. My undergraduate biochemistry swears that this should be CAAX for L-isomer amino acids. However, I know that genes names tend to follow different patterns, so long as it is consistent I don't mind.

Response:

We have changed Figure 1a and b to show "CaaX". Both "CaaX" and "CAAX" are used in the existing literature to denote a sequence where "C" is a Cysteine, "A" or "a" is any aliphatic amino acid, and "X" determines which enzyme acts on the protein. In standard IUPAC code an uppercase "A" specifically represents Alanine. Since there is no standard IUPAC code to denote "any aliphatic amino acid" we have chosen to use the lower case "a" to avoid confusion.

Pg 4 line 25. I still don't agree that potassium ferricyanide is a 'specific' stain, however you have a reference that states that this is the case, so I can't argue. For the record my concern is two parts: Firstly, I don't feel that the stain is 'specific' to T-tubules, only that the contrast is increased somewhat. Secondly, I don't think of potassium ferricyanide as a stain on its own. I have always thought of $K_3[Fe(CN)_6]$ as a reducing agent altering osmium from 8+ to one of the lower oxidation states and thus altering the binding potential within the components in the sample. However, I will accept that I am not a chemist and that both Iron (atomic number 26) and K (19) will give some electron density above the resin.

Response:

We accept the reviewer's point that potassium ferricyanide is not "specific" to the tubules in the true sense of the word. We have deleted the word "specifically" from this sentence, which now reads:

"We used a potassium ferricyanide stain on thin transmission electron microscopy (TEM) sections to label the T-tubules⁸ and distinguish them from sarcoplasmic reticulum (SR)."

Page 8 line 6: Did you quantify the diameters based on transverse TEM sections? The spread of diameters for WT (4k) suggests that the orientation is random. I presume that you measured the short axis of oblique cut tubes. This is not clear in the text (if only to differentiate from the diameters measured using FIBSEM data earlier in the paper).

Response:

This is correct. We have clarified the text, which now reads:
"Quantitation of the **minimum** diameters of SR-associated vs non-SR associated T-tubule (**oblique cut tubes**)"

Figure 5 h has arrows on it that aren't mentioned in the text, or figure legend.

Response:
These arrows have now been removed

Figure 5a-c: what is the contrast agent/stain.
Figure 5a-c and g: has the contrast been inverted?

Response:
This is simply the EGFP-CaaX line with inverted contrast. We have changed the figure legend to read:

"Different tubule morphologies by confocal microscopy (**EGFP-CaaX, inverted**)."

It may be worth adding a note to the figure legends when you have altered the image contrast for printing. For example, figure 6h shows mKate2-ARF6. This is stated to have a very low TT:IT ratio, yet you can clearly see lines in the image. I pasted the image into Fiji and was able to get a clear line plot of these peaks if I only analysed the top half of the image. I did notice that analysing the entire image removed the peaks due to the displacement of the lower set of lines (tubules). This displacement is also seen in the EGFP plot above the merged image as double peaks. Contrast enhancement is mentioned in the methods, but not everybody will read the methods. The issue is particularly noticeable for figures 6-8 as the TT:IT ratios don't always match what the image appears to show.

Another example of this is 6e, where the central cell appears to have a highly reduced CaaX stain compared to the neighbouring cells, yet the TT:IT ratio is shown as being high.

There are more examples, but I presume that these are due to print contrast as the analysis use the raw data.

Response:
The traces are not intended to reflect the ratios between the CaaX signal and the marker signal; this is not a result of adjustment in image contrast for printing. In fact, each data set is scaled independently between the maximum and minimum for the channel. As alluded to in our rebuttal #2 in response to a query from reviewer 2, these traces give an "at a glance" representation of the extent to which the two channels correlate, taking into account the limitation that they are averaged across the entire image. Importantly, this is not how our quantitative comparisons work. The macros use multiple regions of interest on the different domains themselves rather than a simple average through the x and y of the entire image. In order to clarify this point, we have added the following text to the Methods section:

"Intensity traces shown beside individual panels represent average pixel intensities across the entire panel in the x and y planes, with each channel scaled between the maximum and minimum for the channel. The traces are not intended to reflect ratios between channels, rather, these traces give an "at a glance" representation of the extent to which the two channels correlate, taking into account the limitation that they are averaged across the entire image. Methodology for quantitative comparisons is detailed below."

Page 6 line 9. I was going to write that 49% does not "eclipse" 51%, but as I write this I wonder if you meant that after this point the T-tubule surface area will overtake sarcolemma?

Response:
This is exactly what we mean. We have substituted the word "overtake" for "eclipse". The sentence now reads:

"By 5dpf the relative surface area of the T-tubule domain gradually increases to 40% (4737 : 7092 μm^2) and after 10dpf will **overtake** the surface area of the sarcolemma (13695 : 14456 μm^2 ; 49%)."

Page 31 line 7. The reference that you wanted (Mastronarde) is:
Mastronarde, D. N. Automated electron microscope tomography using robust prediction of specimen movements. J. Struct. Biol. 152, 36–51 (2005).

Response:
This has now been added.